# Seasonal and diurnal variation in CO fluxes from an agricultural bioenergy crop

M. Pihlatie[1,2], Ü. Rannik[1], S. Haapanala[1], O. Peltola[1], N. Shurpali[3], P. J. Martikainen[3], S. Lind[3], N. Hyvönen[3], P. Virkajärvi[4], M. Zahniser[5], I. Mammarella[1]

[1]Department of Physics, University of Helsinki, P.O. Box 48, FI-00014 University of Helsinki
[2]Department of Food and Environmental Sciences, P.O. Box 56, FI-00014 University of Helsinki
[3]Biogeochemistry research group, Department of Environmental and Biological Sciences, University of Eastern Finland, Yliopistoranta 1D-E, PO Box 1627, Kuopio campus, FI-70211 Finland
[4]Natural Resources Institute Finland, Green technology, Halolantie 31 A, FI-71750 Maaninka Finland
[5]Aerodyne Research, Inc. 45 Manning Road Billerica, MA 01821-3976, USA

*Correspondence to*: M. Pihlatie (mari.pihlatie@helsinki.fi)

**Abstract.** Carbon monoxide (CO) is an important reactive trace gas in the atmosphere, while its sources and sinks in the biosphere are only poorly understood. Soils are generally considered as a sink of CO due to microbial oxidation processes, while emissions of CO have been reported from a wide range of soil-plant systems. We measured CO fluxes by the micrometeorological eddy covariance method from a bioenergy crop (reed canary grass) in Eastern Finland over April to November 2011. Continuous flux measurements allowed us to assess the seasonal and diurnal variability, and to compare the CO fluxes to simultaneously measured net ecosystem exchange of $CO_2$, $N_2O$ and heat fluxes as well as to relevant meteorological, soil and plant variables in order to investigate factors driving the CO exchange.

The reed canary grass crop was a net source of CO from mid-April to mid-June, and a net sink throughout the rest of the measurement period from mid-June to November 2011, excluding a measurement break in July. CO fluxes had a distinct diurnal pattern with a net CO uptake in the night and a net CO emission during the daytime with a maximum emission at noon. This pattern was most pronounced during the spring and early summer. During this period the most significant relationships were found between CO fluxes and global radiation, net radiation, sensible heat flux, soil heat flux, relative humidity, $N_2O$ flux and net ecosystem exchange. The strong positive correlation between CO fluxes and radiation suggests towards abiotic CO production processes, whereas, the relationship between CO fluxes and net ecosystem exchange of $CO_2$, and night-time CO fluxes and $N_2O$ emissions indicate towards biotic CO formation and microbial CO uptake, respectively.

The study shows a clear need for detailed process-studies accompanied with continuous flux measurements of CO exchange to improve the understanding of the processes associated with CO exchange.

## 1 Introduction

Carbon monoxide (CO) is an important reactive trace gas in the atmosphere where it participates in the chemical reactions with hydroxyl radicals (OH), potentially leading to the production of the strong greenhouse gas ozone ($O_3$). The reactions of CO and OH decrease the atmospheric capacity to oxidize atmospheric methane ($CH_4$), hence indirectly affecting the lifetime of this important greenhouse gas. Although CO itself absorbs only little infrared radiation from the Earth, the cumulative indirect radiative forcing of CO may even be larger than that of a third powerful greenhouse gas nitrous oxide ($N_2O$) (Myhre et al., 2013). Anthropogenic activities related to burning of fossil fuel and biomass (e.g. forest fires) and photochemical oxidation of $CH_4$ and non-methane hydrocarbons are the main sources of CO (Duncan et al., 2007), while the reaction with OH is the major sink of CO in the atmosphere (Duncan and Logan, 2008). Soils are globally considered as a sink for CO due to microbial oxidation processes in the soil (Conrad and Seiler, 1982; Potter et al., 1996; Whalen and Reeburgh, 2001; King and Weber, 2007). According to Conrad and Seiler (1980) the soil consumption of CO is a microbial process, it follows first-order kinetics and can take place in both aerobic and anaerobic conditions. A diverse group of soil microbes are capable of oxidizing CO. They include carboxydotrophs, methanotrophs, and nitrifiers (Ferenci et al., 1975; Jones and Morita, 1983; Bender and Conrad, 1994; King and Weber, 2007), hence potentially linking CO fluxes to the exchange of $CH_4$ and $N_2O$. In addition to CO consumption, production of CO has been found from a wide range of soils (Moxley and Smith, 1998; Gödde et al., 2000; King, 2000; Varella et al., 2004; Galbally et al., 2010; Bruhn et al., 2013; van Asperen et al., 2015), plant roots (King and Crosby, 2002; King and Hungria, 2002), living and degrading plant material (Tarr et al., 1995; Schade et al., 1999; Derendorp et al., 2011; Lee et al., 2012) and degrading organic matter (Wilks, 1959; Conrad and Seiler 1985b). Although microbial CO formation may occur in anaerobic conditions (Funk et al., 1994; Rich and King, 1999), most often the CO production has been related to abiotic processes such as thermal, UV- or visible light-induced degradation of organic matter or plant material (Conrad and Seiler, 1985b; Tarr et al., 1995; Schade et al., 1999; Derendorp et al., 2011; Lee et al., 2012; van Asperen et al., 2015; Fraser et al., 2015). Photodegradation involves direct and indirect photodegradation of e.g. litter or organic material (King et al., 2012). In the direct photodegradation, a molecule (e.g. lignin) has absorbed radiation and undergoes direct changes such as fragmentation, intramolecular rearrangement or electron transfer from or to the molecular (King et al., 2012). In the indirect photodegradation, certain photosensitizers absorb the incoming radiation and

transfer the energy to other molecules such as triplet oxygen, forming reactive intermediates such as singlet oxygen, hydroxyl radical or hydrogen peroxide, which further can change the chemistry of another non-light-absorbing molecule (e.g. cellulose) or part of the same molecule where the photosensitizer resided (King et al., 2012). Indirect photodegradation may also refer to radiation induced stimulation of microbial degradation through breaking down organic compounds making

them easily available for microbial degradation (see King et al. 2012). Thermal degradation is identified as the temperature-dependent degradation of carbon in the absence of radiation and possibly oxygen (Derendorp et al., 2011; Lee et al., 2012; van Asperen et al., 2015).  The separation between CO formation through thermal degradation and photodegradation is very challenging because they both can take place simultaneously and the indirect photodegradation may occur even in the absence of solar radiation if adequate thermal energy is present (Lee et al., 2012).

Understanding of the biological processes leading to CO release and the importance of these sources in terrestrial ecosystems are poorly understood (Moxley and Smith, 1998; King and Crosby, 2002; Vreman et al., 2011; He and He, 2014). Formation of CO from living green plants under illumination and the presence of oxygen was found already in the late 1950's by Wilks (1959) and Siegel et al. (1962). More recently, CO has been found to be formed e.g. in plant roots (King and Crosby, 2002), in stressed plants (He and He, 2014), during heme oxidation (Engel et al., 1972; Vreman et al., 2011), in aromatic amino acid

degradation processes (Hino and Tauchi, 1987), and in lipid peroxidation reactions (Wolff and Bidlack, 1976). However, the importance of these biological CO forming processes in the net CO exchange and, in general, to the global CO budget still remain largely unknown (King and Crosby, 2002).

Most of the reported CO flux measurements are either short-term field experiments (e.g. Conrad and Seiler 1985a; Funk et al, 1994; Zepp et al., 1997; Kuhlbusch et al., 1998; Moxley and Smith 1998; Schade et al., 1999; Varella et al., 2004; Bruhn

et al., 2013; van Asperen et al., 2015), or laboratory incubations with specific treatments of the soil or plant material (Tarr et al., 1995; King and Crosby 2002; Lee et al., 2012). Both CO uptake and emissions are reported from soil-plant systems in different climatic regions, and mostly the CO fluxes range between -2 and 2 nmol $m^{-2}$ $s^{-1}$ (Conrad et al., 1988; Khalil et al., 1990; Funk et al., 1994; Zepp et al., 1997; Moxley and Smith, 1998; Schade et al., 1999; King, 2000; King and Hungria, 2002; Varella et al., 2004; Galbally et al., 2010). Based on the available literature, there is a tendency of south to north

gradient with higher CO emissions from tropical and Mediterranean environments compared to boreal and temperate ecosystems (e.g. Zepp et al., 1997; Kuhlbusch et al., 1998; King, 2000; Varella et al., 2004; Galbally et al., 2010; Constant et al., 2008; Bruhn et al., 2013; van Asperen et al., 2015). However, the high variation between CO uptake and emission rates does not allow yet to classify the ecosystem types or climatic regions. Tall tower (Andreae et al., 2015) and airborne

measurements have indicated source areas of CO both in the Amazon basin (Harriss et al., 1990) and in the North American tundra (Ritter et al., 1992; 1994) suggesting a connection between high plant biomass and biological CO forming processes. To our understanding this is the first study to report long-term and continuous field measurements of CO fluxes ($F_{CO}$) using the micrometeorological eddy covariance (EC) method. We measured $F_{CO}$ above a boreal perennial grassland ecosystem, reed canary grass, over a 7-month snow-free period in 2011 by two parallel laser absorption spectrometers. We compared the $F_{CO}$ with simultaneously measured fluxes of carbon dioxide ($CO_2$), net ecosystem exchange of $CO_2$ (NEE), nitrous oxide ($N_2O$), heat and energy as well as with relevant soil, plant and meteorological variables. Based on previous studies, we expect that the diurnal and seasonal variations in $F_{CO}$ are strongly dependent on radiation and temperature. On the other hand, we do not expect strong relationships between $F_{CO}$ and NEE, or $F_{CO}$ and $N_2O$ fluxes due to the limited information available on the involvement of biological processes in $F_{CO}$, and challenges in separating between parallel abiotic and biotic drivers of $F_{CO}$. We hypothesize that a negative correlation between $F_{CO}$ and NEE can indicate an involvement of a biological component in CO production, and that a positive correlation between night-time $F_{CO}$ and $N_2O$ flux may indicate an involvement of nitrifiers in CO consumption.

## 2    Materials and methods

### 2.1    Measurement site

The measurements were conducted on a mineral agricultural field located in Eastern Finland (63°9'48.69" N, 27°14'3.29" E), cultivated with a perennial reed canary grass (RCG, *Phalaris arundinaceae*, L. cv. Palaton). The measurements covered a period from snow-melt to the new snowfall, from April to November 2011. Long-term (reference period 1981-2010) annual mean air temperature in the region is 3.2°C and the annual precipitation is 612 mm (Pirinen et al., 2012). The crop was cultivated in the beginning of June 2009. In 2011 in the beginning of the growing season (23 May, day 143), the crop was fertilized with an N-P-K-S fertilizer containing 76 kg N ha$^{-1}$ ($NO_3$-N : $NH_4$-N = 47:53). The crop from the previous season was kept at the site over the winter (Burvall, 1997), and was harvested on 28 April (day 118) (Lind et al., 2016). The spring and early summer (days 118-160) was characterized by fast growing crop with the crop height increasing from about 10 cm in mid-May to 1.7 m in late June (day 180), reaching the maximum height of 1.9 m in early July. The field was 6.3 ha in size and from the sampling location of the EC measurement system the footprint was homogenous in all directions, extending 162, 137, 135 and 178 m to N, E, S and W, respectively. There is a slight south to north slope in the field and the wettest area lies in the northern corner of the footprint, which has often standing water during the period of snow-melt (April).

The soil at the site is classified as a Haplic Cambisol/Regosol (Hypereutric, Siltic) (IUSS Working Group WRB, 2007) and the texture of the topsoil (0–28 cm) varied from clay loam to loam based on the US Department of Agriculture (USDA) textural classification system. Within the ploughing layer from the surface to about 30 cm, soil pH varies from 5.4 to 6.1, and soil organic matter content varied between 3 and 11%, respectively. The average C/N ratio in the ploughing layer was 14.9 (ranging from 14.1 to 15.7).

We performed footprint analysis in order to identify the source area of the flux measurements. Two limiting cases were analysed: first, a low crop representing the beginning of the campaign, and second, canopy with 1.9 m height representing the RCG canopy after mid-summer. The measurement heights 2.2 and 2.4 m were used in the analysis, respectively. In the first case, we represented the low canopy as the surface with aerodynamic roughness 0.04 m (determined from measurements), in the second case, a canopy with leaf area distribution characteristic to RCG crops was represented by a beta distribution. In both cases the sources were assumed at the soil surface. Such an assumption was made due to limited information on source-sink behaviour (see Sect. 3 below), and also in order to obtain more conservative footprint estimates. Three stability classes representing unstable (the Obukhov length L = -10 m), near-neutral (L = -100 m) and stable (L = +10 m) conditions were considered. The footprint evaluation was performed by using the Lagrangian stochastic trajectory simulations (e.g. Rannik et al., 2003). The upwind distance contributing 80% of the flux was identified for low/high canopy as follows: 53/23 m, 83/34 m, and 166/60 m for unstable, near-neutral, and stable stratifications, respectively. The conducted footprint analysis reveals that the presence of a canopy significantly reduces the footprint extent. Note that the conservative footprint scenario with no canopy is applicable only for a short period of time due to fast canopy growth in the beginning of the campaign (see Fig. 1d). Considering that prevailing wind direction during the measurement period was from SE and SSW directions, and the wind direction interval 110-315° contributed 90% of the half-hour periods used in the analysis, the footprint analysis hence confirms that the footprint was sufficient and the measurements well represent the RCG canopy.

## 2.2    CO flux measurements

The EC measurements were made as a part of the ICOS (Integrated Carbon Observation System) Finland program during April to November 2011. Here we report the results of $F_{CO}$ calculated from the concentration measurements by two continuous-wave quantum cascade lasers: AR-CW-QCL (model CW-TILDAS-CS Aerodyne Research Inc., see e.g. Zahniser et al., 2009) and LGR-CW-QCL (model N2O/CO-23d, Los Gatos Research Inc., see e.g. Provencal et al., 2005). The measurements by AR-CW-QCL extended the whole measurement period from April to November 2011 (days 110-325),

whereas for LGR-CQ-QCL data is available from later summer to the end of the measurement period (days 206-330). Fluxes by the two analyzers are compared, however, due to the longer data coverage, the diurnal and seasonal variation in $F_{CO}$ is assessed using data from AR-CW-QCL only. The AR-CW-QCL and LGR-CQ-QCL were the same as used in the study of Rannik et al. (2015) wherein four laser-based fast-response gas analyzers to measure nitrous oxide ($N_2O$) fluxes were compared.

The measurement height was 2.2 m until 30 June 2011 (day 181) when the height was raised to 2.4 m due to the growth of RCG. The gas inlets of the closed-path analyzers were located 10 cm below a sonic anemometer (USA-1, Metek Germany GMBH, respectively) used for measuring turbulent wind components. In addition, $CO_2$ and $H_2O$ fluxes were measured at the site by an infrared gas analyzer (LI7000 – Li-Cor Inc., Lincoln, NE, USA) connected to a sonic anemometer (R3-50, Gill Solent Ltd., UK). The closed-path gas analyzers were located in an air conditioned cabin at about 15 m east from the air inlet and the anemometers. This wind direction (50-110° sector) was therefore discarded from further analysis due to possible disturbances to flux measurements. Sample lines (PTFE) were shielded and heated slightly above ambient air temperature. Sample lines were 16 meters in length, their inner diameters were 4 and 8 mm, the sample air flow rates were 13.2 and 11.6 LPM (Rannik et al., 2015). Based on material testing with LGR-CW-QCL, the PTFE tubing was found inert with respect to CO in a constant-flow setup and flow rate of 2.5 LPM (unpublished data). The EC measurements were sampled at 10 Hz frequency. Further details on the EC set-up, instrument specifications and data acquisition, can be found in Rannik et al. (2015) and Lind et al. (2016).

## 2.3 Supporting measurements

A weather station located at the site monitored continuously several meteorological and soil parameters such as air temperature ($T_{air}$) and relative humidity (RH) (model: HMP45C, Vaisala Inc.), precipitation ($P_r$) (model: 52203, R.M. Young Company), global ($R_{glob}$) and net radiation ($R_{net}$) (model: CNR1, Kipp&Zonen B.V.), photosynthetically active radiation (PAR, model: SKP215, Skye instruments Ltd.), soil heat flux at 7.5 cm depth (G) (model: HPF01SC, Hukseflux), soil temperatures at 2.5, 5, 10, 20 and 30 cm depths ($T_{soil}$) (model: 107, Campbell Scientific Inc.), and soil water content at 2.5, 5, 10 and 30 cm depths (SWC) (model: CS616, Campbell Scientific Inc.). All meteorological data were recorded as 30 min mean values and stored using a datalogger (model: CR 3000, Campbell Scientific Inc.).

Leaf area index (LAI) was measured at approximately weekly intervals during the main crop growth period using a plant canopy analyser (model: LAI-2000, LiCor). Green area index (GAI) was estimated on weekly basis from plots adjacent to the LAI measurements according to Wilson et al. (2007) and Lind et al. (2016). The GAI measurements were conducted from three locations (1 x 1 m$^2$) and within each from three spots (8 x 8 cm$^2$) by counting a number of green stems ($S_n$) and green leaves ($L_n$) per unit area and measuring the green area of leaves ($L_a$) and stems ($S_a$). The GAI was calculated as

$$GAI = (S_n S_a) + (L_n L_a) \ .$$

## 2.4    Data processing and analysis

The EC data processing was performed with post-processing software EddyUH (Mammarella et al., 2016). Filtering to eliminate spikes (Vickers and Mahrt, 1997) was performed according to an approach, where the high frequency EC data were despiked by comparing two adjacent measurements. If the difference between two adjacent concentration measurements of CO was greater than 20 ppb, the following point was replaced with the same value as the previous point.

The spectroscopic correction due to water vapour impact on the absorption line shape was accounted for along with the dilution correction. LGR-CW-QCL automatically corrected the water vapour effect by a built-in module in the LGR data acquisition software. The same spectroscopic correction was applied to AR-CW-QCL after a software update in July 2011. Prior to this software update, the respective dilution and spectroscopic corrections to AR-CW-QCL high-frequency CO mole fraction data were performed during the post-processing phase according to Rannik et al. (2015) with the instrument specific CO spectroscopic coefficient (b=0.28) determined in the field.

Prior to calculating the turbulent fluxes, a 2-D rotation (mean lateral and vertical wind equal to zero) of sonic anemometer wind components was done according to Kaimal and Finnigan (1994) and all variables were linearly detrended. The EC fluxes were calculated as 30 min co-variances between the scalars and vertical wind velocity following commonly accepted procedures (e.g. Aubinet et al., 2000). Time lag between the concentration and vertical wind speed measurements induced by the sampling lines was determined by maximizing the covariance. Due to the larger inner diameter (8 mm) of the sampling line in LGR-CW-QCL, the resulting lag time was 4.2 sec compared to that of 0.91 sec for AR-CW-QCL with the sampling line inner diameter of 4 mm. The final processing was, however, done by fixing the time lag to avoid unphysical variation of lag occurring due to random flux errors. Spectral corrections were applied to account for the low and high frequency attenuation of the covariance. The first order response times of the EC systems were determined to be 0.07 and 0.26 sec for

the AR-CW-QCL and LGR-CW-QCL systems, respectively, following the method by Mammarella et al. (2009). This resulted in different flux correction factors mainly due to tube damping: For AR-CW-QCL the 5 and 95 percentile values of flux underestimation were 2.1 and 12.2% and for LGR-CW-QCL 5.7 and 21.4%, respectively. Data quality screening was performed according to Vickers and Mahrt (1997) to ensure exclusion of the system malfunctioning as well as unphysical and/or unusual occasions in measurements. We chose to perform tests on single time series to ensure quality of measurements used in the analysis and did not use the flux stationarity test (Foken and Wichura, 1996) because the CO fluxes are frequently small and respectively with large relative random errors. In such cases the tests based on relative errors are not expected to perform well (e.g. Rannik et al., 2003). After quality screening, 66.0% of the $F_{CO}$ data (AR-CW-QCL) was available, with data coverage of 59.2% during the daytime and 75.9% during the night-time. For details of the data processing and quality screening see Rannik et al. (2015).

To evaluate in detail the seasonal changes in $F_{CO}$ and factors affecting the fluxes, the data was divided into six periods (days 110-145 (20 April – 25 May) = spring (S), days 146-160 (25 May – 9 June) = early summer (ES), days 161-181 (10 June – 30 June) = mid-summer (MS), days 205-240 (24 July – 28 August) = late summer (LS), days 241-295 (29 August – 23 October) = autumn (A), and days 296-325 (24 October – 21 November) = late autumn (LA)). The division into these periods was based on seasonal changes in crop growth and development, or changes in $F_{CO}$ and temperature, while the lengths of the periods were kept as similar in length as possible. Also, $F_{CO}$ were not measured during an instrumental break between days 181 and 204. To compare diurnal changes in the $F_{CO}$, the data was further divided into daytime ($F_{CO\_day}$) and night-time ($F_{CO\_night}$) data. We used sun elevation angle $h<0$ for night-time and $h>0$ for daytime. Pearson correlations between daytime and night-time half-hour average fluxes and other measured parameters were determined. Data processing was performed with Matlab version R2014a (The MathWorks, Inc., United States) and the statistical testing with IBM SPSS statistics 23 (IBM Corporation, United States).

To evaluate the gross CO emission during daytime (gross daytime CO emission), we calculated the gross daytime CO emission in two ways 1) by assuming an equivalent CO uptake for daytime and night-time (constant uptake), and 2) by taking into account temperature dependency ($Q_{10}$ of 1.8) in CO uptake according to Whalen and Reeburgh (2001). Based on a constant CO uptake, the gross daytime CO emission was calculated by subtracting the night-time $F_{CO}$ ($F_{CO\_night}$) from the daytime $F_{CO}$ ($F_{CO\_day}$), presented in Table 1. The uptake CO fluxes refers to the estimated CO uptake taking place during the day, based on measured CO uptake values at night. The temperature corrected daytime CO uptake (Daytime CO uptake, ($Q_{10}$

1.8)) is calculated by extrapolating the measured night-time CO fluxes ($F_{CO\_night}$) (table 1) to using the difference between day and night soil temperatures (2.5 cm depth) ($\Delta t_{soil}$) and the $Q_{10}$-value of 1.8 (Whalen and Reeburgh, 2001). The temperature dependent daytime CO uptake ($R2$) was solved from the equation

$$Q_{10} = \frac{\left(\frac{R2}{R1}\right)^{10}}{(T2-T1)} \qquad ,$$

where $Q_{10}$ is 1.8 (Whalen and Reeburgh, 2001), $R1$ is the night-time $F_{CO}$ (net $F_{CO\_night}$) (nmol m$^{-2}$ s$^{-1}$), and $T2$-$T1$ is the temperature difference between daytime ($T2$) and night-time ($T1$) soil temperature at 2.5 cm depth (ºC), respectively. The temperature corrected gross daytime CO emissions (Gross daytime CO emission ($Q_{10}$ 1.8)) was estimated by subtracting the temperature corrected daytime CO uptake (Daytime CO uptake, ($Q_{10}$ 1.8)) from the daytime $F_{CO}$ ($F_{CO\_day}$). These gross CO emission and uptake rates were estimated for each of the six measurement periods and are presented in Table 2.

## 3      Results

### 3.1      Seasonal variation

The RCG field was a net source of CO from mid-April in the spring to mid-June (days 110-160), after which the site turned to a net sink until the end of the measurement period in November 2011 (days 161-325) (Fig. 1f). Cumulative CO flux (cum $F_{CO}$) curves, calculated by cumulating the half-hourly fluxes, show that the site was a net sink of CO over the 7-month

measurement period (Fig. 1f). During daytime, the net CO fluxes ($F_{CO\_day}$) were positive during the spring and early summer (days 110-160) and again during late summer (days 205-240). These daytime emissions were highest during the spring (Table 1). Night-time CO fluxes ($F_{CO\_night}$) were negative (CO uptake) throughout the whole measurement period with a trend of increasing CO consumption towards late autumn (Table 1).

The spring emission period (days 110-145) covered a time (days 110-118) with a standing dry crop from the previous year.

The old crop was harvested on 28 of April (day 118), after which the ground consisted mainly of short dead plant material and litter, and a slowly sprouting new RCG. The second emission period in early summer (days 146-160) was characterized by fast growing RCG crop, high and fertilizer-induced $N_2O$ emissions (Shurpali et al., 2016), increasing air and soil temperatures, growing leaf area and increasing NEE (Fig. 1). After the crop had reached its maximum height of 1.9 m in mid-June (around day 160), the site started to act as a net sink of CO, followed by a period of net daytime emissions during

late summer in July-August (days 205-240). The autumn (A, LA) was characterized by decreasing daytime $F_{CO}$ ($F_{CO\_day}$) and

slowly dropping air and soil temperatures, decreasing radiation intensity, and decreasing photosynthetic activity of the crop (less negative NEE) (Fig. 1).

Comparison of the two gas analyzers, AR-CW-QCL and LGR-CW-QCL, during the period when both were operational (days 205-325), shows that the measured $F_{CO}$ agree reasonably well (Fig. 1f). A correlation scatter plot of the $F_{CO}$ from LGR-CW-QCL against $F_{CO}$ of AR-CW-QCL results a correlation coefficient of 0.95 and a slope of 0.96 (data not shown). According to this comparison, LGR-CW-QCL shows slightly (4%) smaller fluxes compared to AR-CW-QCL, however, the difference between the two analyzers is very small, giving us confidence in the use of either of the analyzer in further analysis.

## 3.2 Diurnal variation

The $F_{CO}$ had a distinct diurnal pattern with an uptake in the night-time and an emission during the daytime with maximum emissions at noon (Fig. 2). This pattern was most pronounced during the spring, days 110-145, when the maximum daytime CO emissions reached 2.7 nmol m$^{-2}$ s$^{-1}$ (Fig. 2). The net $F_{CO}$ was positive (emission) during the spring and early summer, after which the night-time uptake dominated making the site as a net sink of CO (Fig. 2, Table 1.). Night-time $F_{CO}$ show a near constant uptake of CO over the whole measurement period with a mean of -0.77 nmol m$^{-2}$ s$^{-1}$ over the whole measurement period (Fig. 2, Table 1.).

The diurnal $F_{CO}$ over the six measurement periods followed closely the daily pattern of $R_{glob}$ with a maximum $F_{CO}$ (emission) at around noon and minimum $F_{CO}$ (highest uptake) at midnight (Figs. 2 and 3). The highest radiation intensity was reached during the early summer (days 146-160), while the maximum $F_{CO}$ were observed during the spring (days 110-145) (Figs. 2 and 3). Diurnal variation in soil temperature was highest during the spring and early summer, and always peaked during the afternoon (Fig. 3).

Compared to the $F_{CO}$, the diurnal variation in $CO_2$ exchange, expressed here as NEE, was very small during the spring (days 110-145) (Fig. 4). A rapid increase in LAI and GAI at around day 150 (Fig. 1d) lead to an increase in $CO_2$ uptake during daytime, which is seen in a distinct diurnal pattern with high $CO_2$ uptake (negative NEE) during daytime and a small positive NEE during night-time (Fig. 4). Maximum NEE values were reached during mid-June (days 161-181) after which the NEE slowly decreased and the $CO_2$ uptake disappeared by mid-October (day 290) (Figs. 1 and 4).

During early summer, the fluxes of $N_2O$ followed a similar daily pattern as that of $F_{CO}$ with higher daytime $N_2O$ emissions compared to night-time fluxes (Shurpali et al., 2016). This period of high $N_2O$ emissions (days 143-158) was a direct

response to the N-P-K-S fertilizer application on 23 May, and it lasted for about 15 days. After this, an opposite diurnal pattern was observed during which the $N_2O$ emissions were on average 50% higher during the night than during the day (Shurpali et al., 2016).

The gross daytime CO emissions were estimated in two ways: 1) assuming an equal CO uptake during day and night
(constant uptake), and 2) accounting for temperature dependent CO uptake according to Whalen and Reeburgh (2001). The gross CO emissions calculated in either way, show that in the daytime the site emitted CO throughout the whole measurement period with the highest emissions during the spring and late summer (Table 2). During mid-summer and autumn the daytime emissions were markedly smaller, and less than half of the emissions during the spring. The smallest gross CO emissions were measured in late autumn (Table 2). When the temperature dependency in the CO uptake was taken
into account, using a $Q_{10}$ value of 1.8 (Whalen and Reeburgh, 2001), both the daytime CO uptake (Daytime CO uptake ($Q_{10}$, 1.8)), and the daytime emission (Daytime CO emission ($Q_{10}$, 1.8)) were almost twice as high as the rates without the temperature correction (Table 2).

### 3.3    Driving factors for CO fluxes

The most pronounced relationships between $F_{CO}$ and other measured scalars were found for the daytime data (sun elevation h>0) during the two emission periods in the spring and early summer (Table 3, Figure 5). Furthermore, the strongest correlations were found during the spring between $F_{CO\_day}$ and $R_{glob}$ (r=0.760, p<0.01), $R_{net}$ (r=0.760, p<0.01), H (r=0.729, p<0.01) and G (r=0.575, p<0.01). These positive correlations remained significant but became weaker towards the end of the measurement period (Table 3, Figure 5). Strong negative correlations were found during the spring between $F_{CO\_day}$ and RH
(r=-0.537, p<0.01), and during the early summer with NEE (r=-0.469, p<0.01), while the correlation between daytime $F_{CO}$ and $M_{CO}$, $F_{N2O}$ or ecosystem respiration (RESP) were very weak throughout the 7-month measurement period (Table 3). Night-time (h<0) $F_{CO}$ ($F_{CO\_night}$) correlated weakly with $F_{N2O}$ (r=-0.336, p<0.01), H (r=0.315, p<0.01), and LE (r=-0.241, p<0.05) in the spring and with $M_{soil}$ (r=0.308, p<0.01) during early summer (Table 4). A strong negative correlation was found between $F_{CO\_night}$ and $F_{N2O}$ during mid-summer (r=-0.607, p<0.01) and late autumn (r=-0.514, p<0.01), and a positive
correlation between $F_{CO\_night}$ and LE (r=0.459, p<0.05) during mid-summer (Table 4).

# 4       Discussion

Based on the 7-month EC flux measurements at the RCG crop, we demonstrate that the EC method is suitable for measuring CO fluxes ($F_{CO}$) from a perennial agricultural crop. We show that the soil-plant system acted as a net source of CO during the spring and early summer and a net sink of CO over the late summer and autumn, and that the $F_{CO}$ had a clear diurnal pattern with net CO emissions during daytime and net CO uptake during the night. This source-sink pattern existed over the whole measurement period with decreasing net emissions towards the end of the autumn. To our knowledge, similar long-term and continuous $F_{CO}$ data series measured by the EC method over any ecosystem type does not exist, and hence this study is unique in bringing new insight to the understanding of short-term diurnal and long-term seasonal $F_{CO}$ dynamics at ecosystem-level. Combining the continuous $F_{CO}$ data with simultaneously measured $CO_2$, $N_2O$ and energy fluxes as well as meteorological and soil variables allowed us to distinguish driving variables of the $F_{CO}$, and demonstrate the suitability of the EC method to analyze ecosystem-level CO exchange dynamics. Due to the fact that the EC method measures net fluxes, we cannot directly separate between different processes, such as CO production and consumption. However, based on process understanding and our data, we made an assumption that most of the CO production takes place during daytime and that the night-time CO uptake is due to microbial activity. After these assumptions, we divided the data into daytime and night-time periods in order to analyse seasonal changes in dependencies between CO emissions and uptake and their driving variables.

Cumulative CO fluxes (cum $F_{CO}$) over the whole 7-month measurement period showed that the RCG crop was a net sink of CO. This cum $F_{CO}$ estimation may be biased due to the instrumental break during July (days 181-205), during which we do not have an estimate of the CO fluxes. Also, due to the fact that the data processing removed more daytime values (40.8% removed) compared to night-time data (24.1% removed), the night-time CO uptake is weighing more in the cumulative flux estimation, potentially leading to smaller and more negative net fluxes than estimated based on an equal number of flux data from daytime and night-time. We tested a simple statistical gap-filling method to obtain a balanced number of daytime and night-time data, however, as this gap-filling did not change the interpretation of the results, and as we do not have an appropriate process model to account for uptake and emission processes, we decided not to present these results.

Based on the seasonal variation, we could divide the $F_{CO}$ to a distinct emission period and an uptake period. During the "emission" period (days 110-160), the soil-plant system was a strong source of CO during daytime and a small sink during night-time. Furthermore, the emission period was divided into a spring emission period (days 110-145) and an early summer emission period (days 146-160), which differed from each other based on the daytime CO emission rates and relationships with other measured variables such as radiation and NEE. The highest CO emissions were observed soon after the snow melt during the spring in April to early May when the air and soil temperatures were rather low and the crop was not yet actively

photosynthesizing (low LAI, low NEE), while the radiation intensity was already rather high. As suggested by King (2000), the elevated spring-time CO emissions probably resulted from the degradation of the readily available last year's crop and litter, which has been shown to be a significant source of CO (King, 2000; King et al., 2012; Lee et al., 2012). Decreasing amounts of this readily degradable litter also partly explains the decreasing trend in CO emissions during spring and early summer (King, 2000).

In general, the $F_{CO}$ rates from the RCG crop in this study fall into the same range as those reported from different natural and managed ecosystems across the different climatic regions (Table 5). There is a tendency of higher CO emissions from tropical and Mediterranean ecosystems compared to northern and boreal ecosystems. The data comparison also indicates net CO uptake from forest ecosystems (Zepp et al., 1997; King, 2000; Kuhlbusch et al., 1998), CO emissions from savanna and croplands ecosystems (King, 2000; Kisselle et al., 2002; Varella et al., 2004; Galbally et al., 2010), and variation between CO uptake and emission from grassland ecosystems (Constant et al., 2008; Bruhn et al., 2013; van Asperen et al., 2015; Table 5). When comparing daytime fluxes, the mean daytime $F_{CO}$ at the RCG of 0.21 nmol m$^{-2}$ s$^{-1}$ is at the lower end of the emissions reported in grasslands or croplands (King, 2000; Bruhn et al., 2013; van Asperen et al., 2015), however, the strong seasonality and higher CO emissions during the spring (0.91 nmol m$^{-2}$ s$^{-1}$) are very similar to the fluxes measured in tropical pastures and croplands (King, 2000; Varella et al., 2004; Galbally et al., 2010). The comparison of reported CO fluxes to our results is challenged by the differences in temporal resolution of the flux measurements. As most of the reported studies are conducted during daytime only and with biweekly to monthly intervals, possible diurnal and seasonal variation in the fluxes are neglected (e.g. King, 2000; Varella et al., 2004; Galbally et al., 2010; van Asperen et al., 2015).

To calculate an annual CO balance of the RCG site, we used a mean $F_{CO}$ over the whole measurement campaign of -0.25 nmol m$^{-2}$ s$^{-1}$ (Table 1) to apply for the missing period from day 326 to day 109 (22 November 2011 - 18 April 2012). This annual cumulative $F_{CO}$ of -111 mg CO m$^{-2}$ yr$^{-1}$ naturally has a high uncertainty due to the missing measurements. However, we expect that the $F_{CO}$ are minimal during the snow-cover period in December-February. Whereas, for the spring period during the snow-melt in March-April, the assumption of small $F_{CO}$ does not necessarily hold as the amount of radiation and temperature increase and the soil surface is freed from the snow allowing the old previous year's crop residues to decompose. Hence, we expect that the use of the mean $F_{CO}$ from the measurement period probably underestimates the $F_{CO}$ during the early spring period.

Similar to our findings from the emission period, soils from boreal to tropical regions have been found to have a clear diurnal pattern with emissions in the noon and uptake during the night (Conrad and Seiler, 1985a; Schade et al., 1999; Kisselle et al., 2002; Constant et al., 2008; van Asperen et al., 2015). The existing literature suggests that the net CO exchange involves

simultaneous production and consumption processes occurring in a variety of soil-plant systems. While the consumption is suggested to be a microbial process in the soil (Conrad and Seiler, 1980), the production of CO has been mostly linked with abiotic photodegradation or thermal degradation of soils, organic matter and vegetation (Conrad and Seiler 1985a; 1985b; Moxley and Smith 1998; Lee et al., 2012; Bruhn et al., 2013; Fraser et al., 2015) or to a minor extent to anaerobic microbial

activity in wet soils (Funk et al., 1994; Bender and Conrad, 1994). In our study, the net CO uptake during night-time indicates that there is a microbial sink of atmospheric CO. We expect that this CO consumption also exists during daytime, and it may be increased due to temperature dependency of the consumption (King, 2000; Whalen and Reeburgh, 2001). We did not find correlation between daytime or night-time CO concentration ($M_{CO}$) and $F_{CO}$ (Tables 3 and 4), indicating that $M_{CO}$ is not limiting CO consumption at our site. In our site the estimated daytime CO consumption is overruled by a

simultaneous strong CO production, creating the observed diurnal pattern in the spring and early summer. Assuming a temperature dependent CO uptake (Whalen and Reeburgh, 2001), we estimated that the daytime CO uptake (mean of -1.79 nmol $m^{-2}$ $s^{-1}$) is over two times that in the night (mean -0.77 nmol $m^{-2}$ $s^{-1}$) (Tables 1 and 2). When this was taken into account in gross daytime CO emissions, also daytime CO emission was estimated markedly higher compared to the daytime CO emission without the temperature corrected CO uptake. These gross rate calculations result slightly higher CO uptake and

smaller emission compared to what van Asperen et al. (2015) reported from a Mediterranean grassland. van Asperen et al. (2015) reported night-time CO uptake up to -1.0 nmol $m^{-2}$ $s^{-1}$ and daytime emissions of around 10 nmol $m^{-2}$ $s^{-1}$ by a flux gradient method. They also reported night-time minimum chamber fluxes of -0.8 nmol $m^{-2}$ $s^{-1}$ and daytime maximum chamber fluxes of up to 3 nmol $m^{-2}$ $s^{-1}$, both measured over about one month period. Other reported diurnal CO fluxes are mostly over 24-hours only, hence mainly demonstrating the potential variation in the CO exchange over one day (Zepp et al.,

1997; Kisselle et al., 2002; Constant et al., 2008).

Strong correlations between daytime $F_{CO}$ and $R_{glob}$ (and other radiation components) especially in the spring and early summer indicate that the direct or indirect effects of radiation drive the CO emissions. During the spring period, the strongest correlations were observed between daytime $F_{CO}$ and solar radiation ($R_{glob}$, $R_n$), sensible heat flux and soil heat flux, all indicating a close connection between $F_{CO}$ and radiation and heat transfer. Factors supporting the CO production through

abiotic photodegradation and thermal degradation processes include high C to N ratio of the plant material (King et al., 2012), presence of oxygen (Tarr et al., 1995; Lee et al., 2012), greater solar radiation exposure (no shading) (King et al., 2012), and litter area to mass ratio (King et al., 2012; Lee et al., 2012). As the dead plant material in our measurement site has a high C to N ratio (mean ±stdev: 66±6.3), and as this dry plant material was well exposed to radiation in the spring, we expect that the conditions were suitable for CO formation through abiotic degradation processes. Correlation between $F_{CO}$

and soil heat flux (G), and that between $F_{CO}$ and $T_{air}$ indicate that also thermal degradation plays an important role in daytime CO formation. As the correlation between $F_{CO}$ and $T_{soil}$ was poor (at maximum r=0.355), the $T_{soil}$ at the depth of 2.5 cm does not seem to reflect the location of CO formation via thermal degradation. However, a better correlation between $F_{CO}$ and $T_{air}$ indicates that most likely majority of thermal degradation or indirect photodegradation takes place on the soil surface or in

(dead) plant material on top of the soil where temperature and degradation processes are directly influenced by radiation. A close look at the diurnal pattern of $F_{CO}$ during the autumn and summer days in Figure 2 during the time of sunrise or sunset reveals that the $F_{CO}$ starts to increase before the sun rise at around 9 am (late autumn, days 296-325), and the $F_{CO}$ in the afternoon continues to decrease after the sun set at around 20 pm (late summer, days 205-240). These phenomena could be explained by temperature driven CO consumption, which according to soil temperatures should have a minimum soon after

sunrise, hence affecting to the diurnal variation of the net $F_{CO}$ (Figure 3). As the abiotic thermal degradation is temperature dependent, we do not expect thermal degradation to be responsible for increased CO production during early morning hours before the sunrise, however, this process may have contributed to the prolonged CO formation after the sunset during late summer. Our data does not allow for deeper process-level interpretation, however, these findings also indicate that direct photodegradation is probably not the sole source of CO at the site, and that also indirect photodegradation, thermal

degradation or biological processes may play roles in the CO formation.

Although we cannot separate between biotic and abiotic CO formation at the RCG field site, our findings of the negative correlation between daytime $F_{CO}$ and NEE (r=-0.469) during early summer (days 146-160), the period of maximum NEE, indicate that some CO may also be formed via plant physiological processes. This early summer CO emission period (days 146-160) coincides with the steepest slope in $CO_2$ uptake (more negative NEE), supporting the findings of Wilks (1959),

Bruhn et al. (2013) and Fraser et al. (2015) that CO can be emitted not only from dead plant matter but also from living green leaves. The observed daytime CO emissions during early summer can have also been formed through abiotic processes, which also occur in living plants (Tarr et al., 1995; Erickson et al., 2015). King et al. (2012) suggested that the CO emissions from photodegradation generally decrease with increasing leaf area index, and Tarr et al. (1995) and Erickson et al. (2015) found that the CO photoproduction efficiency is lower for living plants compared to senescent or dead vegetation.

These studies support our findings of lower daytime CO emissions from fully developed crop during the summer (days 205-240) compared to CO emissions during the spring (days 110-145), when the ground was covered by the dead plant litter. Still the role of biological CO formation in living green plants and the forming processes remain unresolved and call for further process-studies.

Based on our data, we suggest that a poor correlations between $F_{CO}$ and ecosystem respiration (RESP) throughout the measurement campaign indicates that microbial and plant respiratory activity does not play an important role in the CO formation. With respect to $F_{N2O}$ and $F_{CO}$, we do not expect a strong relationship due to the difficulties in separating between overlapping abiotic CO production, microbial CO consumption (Conrad and Seiler, 1980; Moxley and Smith 1998), and microbial $N_2O$ production/uptake in the soil. As nitrifiers are among the diverse microbial community oxidizing CO in soils (Jones and Morita, 1983; Bender and Conrad, 1994; King and Weber, 2007), a high nitrification activity may be reflected in higher CO consumption in the soil. In the field, this could be visible during night-time when the CO consumption is expected to dominate the net CO fluxes, while in most of the year during daytime the CO production overrides the consumption. If a large fraction of the CO uptake was due to nitrification activity, we should be able to see this in negative correlation between night-time $F_{N2O}$ and $F_{CO\_night}$. In fact, we found significant negative correlations between $F_{N2O}$ and $F_{CO\_night}$ in the spring (r=-0.336), mid-summer (r=-0.607) and late autumn (r=-0.514). These correlations were significant but much weaker during the daytime (Table 3). These findings hint towards the role of nitrifiers in CO consumption at the reed canary grass site. However, we have no process data from the site showing the link between nitrifiers and CO consumption.

This is the first study to apply EC based techniques to measure long-term variation in $F_{CO}$ at any ecosystem type in the world. In addition to the long-term seasonal variability in the $F_{CO}$, we were able to identify the driving variables and processes at ecosystem level, findings that have previously been shown with plot scale chamber measurements or in the laboratory. The high diurnal and seasonal variability over the 7-month measurement period shows that there is an urgent need for continuous and long-term assessment of $F_{CO}$. The limitations of the EC method, such as inability to separate between CO production and consumption processes, naturally increase uncertainties in the interpretation of the results. However, despite these limitations, the data allowed us to distinguish between the daytime and night-time processes involved and to link the diurnal and seasonal variability to abiotic and biotic processes. Also, the EC method has clear advantages over the traditional enclosure methods such as measuring non-disturbed ecosystem fluxes and avoiding surface reactions with measurement material, both supporting the application of the EC method to measure $F_{CO}$ in different ecosystems.

## 5      Conclusions

Long-term and continuous EC based measurements of $F_{CO}$ over an arable reed canary grass showed clear seasonal variation with net emissions during the spring and early summer, and net uptake of CO during the late summer and autumn. Daytime emissions of CO and night-time uptake of CO demonstrate the dynamic nature of parallel consumption and production

processes. Based on daytime and night-time separation of $F_{CO}$, and correlation analysis between $F_{CO}$ and radiation, $T_{soil}$, $T_{air}$, heat fluxes (H, LE), NEE and ecosystem respiration, and $F_{N2O}$ the daytime CO emissions were suggested to be driven mainly by direct and indirect effects of radiation such as heat fluxes and temperature, while the night-time CO uptake was found to be connected to $N_2O$ emissions. Although, the measurement approach does not allow to separate between different CO forming and consuming processes, CO emissions are suggested to mainly result from abiotic photo- and thermal degradation of plant material and soil organic matter, whereas the night-time CO uptake was expected to be microbial. This study demonstrates the applicability of the EC method in CO flux measurements at ecosystem scale, and shows the potential in linking the short-term $F_{CO}$ dynamics to its environmental drivers. In order to fully understand the source-sink dynamics and processes of CO exchange, continuous and long-term $F_{CO}$ measurements in combination with process-based studies are urgently needed.

**Acknowledgements**

This work was supported by the Academy of Finland (project nos. 118780, 127456, 1118615 and 263858, 294088, 288494). ICOS (271878), ICOS-Finland (281255) and ICOS-ERIC (281250), DEFROST Nordic Centre of Excellence and InGOS EU are gratefully acknowledged for funding this work. This work was also supported by institutional research funding (IUT20-11) of the Estonian Ministry of Education and Research. The UEF part of the research work was supported by the funding from the UEF infrastructure funding, Academy of Finland FidiPro programme (PIs – Profs Pertti Martikainen and Seppo Kellomäki) and the Ministry of Agriculture and Forestry, Finland and MTT Agrifood Research Finland (MTT) strategic funding (project no. 21030028).

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

| Period, days | $F_{CO\_day}$ | | | | $F_{CO\_night}$ | | | | net $F_{CO}$ | | | |
| --- | --- | --- | --- | --- | --- | --- | --- | --- | --- | --- | --- | --- |
| | mean | median | 25$^{th}$-75$^{th}$ percentile | | mean | median | 25$^{th}$-75$^{th}$ percentile | | mean | median | 25$^{th}$-75$^{th}$ percentile | |
| S, 110-145 | 0.97 | 0.68 | -0.15 | 2.00 | -0.64 | -0.56 | -0.97 | -0.20 | 0.41 | 0.09 | -0.57 | 1.28 |
| ES, 146-160 | 0.24 | 0.08 | -0.29 | 0.57 | -0.67 | -0.49 | -0.72 | -0.33 | 0.03 | -0.10 | -0.45 | 0.43 |
| MS, 161-181 | -0.07 | -0.08 | -0.40 | 0.24 | -0.67 | -0.52 | -0.86 | -0.22 | -0.22 | -0.18 | -0.55 | 0.16 |
| LS, 205-240 | 0.36 | 0.30 | -0.07 | 0.87 | -0.76 | -0.49 | -0.96 | -0.19 | -0.09 | -0.04 | -0.53 | 0.49 |
| A, 241-295 | -0.12 | -0.18 | -0.48 | 0.13 | -0.66 | -0.61 | -0.90 | -0.32 | -0.44 | -0.44 | -0.77 | -0.10 |
| LA, 296-325 | -0.62 | -0.59 | -0.94 | -0.26 | -1.05 | -1.01 | -1.37 | -0.65 | -0.92 | -0.89 | -1.25 | -0.49 |
| All, 110-325 | 0.21 | 0.01 | -0.41 | 0.55 | -0.77 | -0.66 | -1.06 | -0.33 | -0.25 | -0.34 | -0.79 | 0.17 |

6
7
8
9
10
11
12
13
14
15
16
17
18
19
20
21

Table 2. Mean, median and $25-75^{th}$ percentiles of the estimated gross daytime CO emission (Gross daytime CO emission, nmol $m^{-2}$ $s^{-1}$), temperature corrected daytime CO uptake (Daytime CO uptake, ($Q_{10}$ 1.8)), and temperature corrected gross daytime CO emission (Gross daytime CO emission ($Q_{10}$ 1.8)) calculated for the read canary grass (RCG) crop at Maaninka. The CO emission and uptake rates are calculated for six measurement periods (S = spring, ES = early summer, MS = mid-summer, LS = late summer, A = autumn, LA = late autumn), and over the full measurement period (All) from April to November 2011. The estimated gross daytime CO emission is calculated in two ways: 1) assuming a constant CO uptake, and 2) assuming temperature dependent CO uptake. Gross daytime CO emission based on a constant CO uptake (way 1, Chapter 2.4) refers to the difference between daytime fluxes ($F_{CO\_day}$) and night-time fluxes ($F_{CO\_night}$) presented in Table 1. The temperature corrected gross daytime CO emission (Gross daytime CO emission ($Q_{10}$, 1.8)) refers to the difference between daytime fluxes ($F_{CO\_day}$) (Table 1.) and daytime CO uptake ($Q_{10}$, 1.8). The daytime CO uptake (Daytime CO uptake ($Q_{10}$, 1.8)) is calculated by extrapolating the night-time CO fluxes ($F_{CO\_night}$) to daytime using the difference between day and night soil temperatures (2.5 cm depth) ($\Delta t_{soil}$) and the $Q_{10}$-value of 1.8 (Whalen and Reeburgh, 2001), as described in Chapter 2.4.

| Period, DOY | Gross daytime CO emission | | | | $\Delta t_{soil}$ | Daytime CO uptake ($Q_{10}$, 1.8) | | | | Gross daytime CO emission ($Q_{10}$, 1.8) | | | |
|---|---|---|---|---|---|---|---|---|---|---|---|---|---|
| | mean | median | $25^{th}$-$75^{th}$ percentile | | $T_{day}$-$T_{night}$ | mean | median | $25^{th}$-$75^{th}$ percentile | | mean | median | $25^{th}$-$75^{th}$ percentile | |
| S, 110-145 | 1.61 | 1.24 | 0.83 | 2.20 | 2.1 | -1.24 | -1.09 | -1.89 | -0.39 | 2.22 | 1.76 | 1.74 | 2.39 |
| ES, 145-160 | 0.91 | 0.57 | 0.43 | 0.91 | 1.2 | -1.27 | -0.92 | -1.36 | -0.63 | 1.51 | 1.00 | 1.06 | 1.20 |
| MS, 160-181 | 0.59 | 0.45 | 0.46 | 0.46 | 0.7 | -1.23 | -0.96 | -1.58 | -0.41 | 1.15 | 0.89 | 1.18 | 0.65 |
| LS, 205-240 | 1.12 | 0.79 | 0.89 | 1.07 | 0.9 | -1.42 | -0.91 | -1.78 | -0.36 | 1.77 | 1.21 | 1.71 | 1.24 |
| A, 240-295 | 0.54 | 0.42 | 0.41 | 0.45 | 1.0 | -1.24 | -1.13 | -1.68 | -0.59 | 1.11 | 0.95 | 1.19 | 0.72 |
| LA, 295-325 | 0.42 | 0.42 | 0.43 | 0.39 | 0.3 | -1.90 | -1.84 | -2.49 | -1.18 | 1.28 | 1.25 | 1.56 | 0.92 |
| ALL, 110-325 | 0.98 | 0.68 | 0.65 | 0.88 | 3.5 | -1.58 | -1.37 | -2.19 | -0.68 | 1.79 | 1.38 | 1.78 | 1.23 |

Table 3. Pearson correlation matrix for half-hour daytime CO fluxes ($F_{CO\_day}$) during six periods (S = spring, ES = early summer, MS = mid-summer, LS = late summer, A = autumn, LA = late autumn) at the reed canary grass crop in Maaninka. $M_{CO}$ = CO mixing ratio, NEE = net ecosystem exchange, RESP = ecosystem respiration, $F_{N2O}$ = N$_2$O flux, H = sensible heat flux, LE = latent heat flux, $T_{air}$ = air temperature, $R_{glob}$ = global radiation, $R_{net}$ = net radiation, G = soil heat flux, $T_{soil}$ = soil temperature at 2.5 cm, SWC = soil water content at 2.5 cm.

| | $F_{CO\_day}$ S, 110-145 | | n | $F_{CO\_day}$ ES, 146-160 | | n | $F_{CO\_day}$ MS, 161-180 | | n | $F_{CO\_day}$ LS, 205-240 | | n | $F_{CO\_day}$ A, 241-295 | | n | $F_{CO\_day}$ LA, 296-325 | | n |
|---|---|---|---|---|---|---|---|---|---|---|---|---|---|---|---|---|---|---|
| $M_{CO}$ | 0.080 | * | 711 | 0.128 | ** | 510 | -0.116 | * | 436 | -0.074 | | 488 | 0.038 | | 851 | -0.284 | ** | 288 |
| NEE | -0.188 | ** | 711 | -0.469 | ** | 510 | -0.308 | ** | 436 | -0.488 | ** | 488 | -0.237 | ** | 850 | -0.25 | ** | 288 |
| RESP | 0.015 | | 711 | 0.274 | ** | 510 | 0.272 | ** | 436 | 0.257 | ** | 488 | 0.198 | ** | 850 | 0.077 | | 288 |
| $F_{N2O}$ | -0.219 | ** | 669 | 0.000 | | 453 | -0.293 | ** | 426 | -0.026 | | 478 | -0.085 | * | 850 | -0.172 | ** | 287 |
| H | 0.729 | ** | 711 | 0.329 | ** | 510 | 0.234 | ** | 436 | 0.427 | ** | 488 | 0.132 | ** | 851 | -0.076 | | 288 |
| LE | 0.402 | ** | 418 | 0.398 | ** | 401 | 0.514 | ** | 224 | 0.625 | ** | 307 | 0.317 | ** | 573 | 0.289 | ** | 185 |
| RH | -0.537 | ** | 711 | -0.176 | ** | 510 | -0.303 | ** | 436 | -0.434 | ** | 488 | -0.081 | * | 851 | -0.179 | ** | 288 |
| $T_{air}$ | 0.425 | ** | 711 | 0.344 | ** | 510 | 0.36 | ** | 436 | 0.433 | ** | 488 | 0.241 | ** | 851 | 0.073 | | 288 |
| $R_{glob}$ | 0.760 | ** | 711 | 0.498 | ** | 510 | 0.373 | ** | 436 | 0.549 | ** | 488 | 0.265 | ** | 851 | 0.256 | ** | 288 |
| $R_{net}$ | 0.760 | ** | 711 | 0.515 | ** | 510 | 0.376 | ** | 436 | 0.558 | ** | 488 | 0.277 | ** | 851 | 0.218 | ** | 288 |
| G | 0.575 | ** | 711 | 0.473 | ** | 510 | 0.406 | ** | 436 | 0.485 | ** | 488 | 0.247 | ** | 851 | 0.033 | | 288 |
| $T_{soil}$ | 0.191 | ** | 711 | 0.282 | ** | 510 | 0.318 | ** | 436 | 0.358 | ** | 488 | 0.206 | ** | 851 | 0.071 | | 288 |
| $M_{soil}$ | -0.099 | ** | 711 | 0.033 | | 510 | 0.095 | * | 436 | 0.086 | | 488 | -0.105 | ** | 851 | 0.095 | | 288 |

**. Correlation is significant at the 0.01 level (2-tailed).

*. Correlation is significant at the 0.05 level (2-tailed).

Table 4. Pearson correlation matrix for half-hour night-time CO fluxes ($F_{CO\_night}$) during six periods (S = spring, ES = early summer, MS = mid-summer, LS = late summer, A = autumn, LA = late autumn) at the reed canary grass crop in Maaninka. $M_{CO}$ = CO mixing ratio, NEE = net ecosystem exchange, RESP = ecosystem respiration, $F_{N2O}$ = N$_2$O flux, H = sensible heat flux, LE = latent heat flux, $T_{air}$ = air temperature, $R_{glob}$ = global radiation, $R_{net}$ = net radiation, G = soil heat flux, $T_{soil}$ = soil temperature at 2.5 cm, SWC = soil water content at 2.5 cm.

| | $F_{CO\_night}$ S, 110-145 | n | $F_{CO\_night}$ ES, 146-160 | n | $F_{CO\_night}$ MS, 161-180 | n | $F_{CO\_night}$ LS, 205-240 | n | $F_{CO\_night}$ A, 241-295 | n | $F_{CO\_night}$ LA, 296-325 | n |
|---|---|---|---|---|---|---|---|---|---|---|---|---|
| $M_{CO}$ | -0.045 | 380 | -0.043 | 142 | -0.279 ** | 134 | -0.165 ** | 324 | -0.110 ** | 1149 | -0.041 | 700 |
| NEE | 0.069 | 380 | -0.167 * | 142 | -0.118 | 134 | -0.049 | 324 | 0.024 ** | 1149 | 0.025 | 700 |
| RESP | 0.056 | 380 | 0.015 | 142 | -0.006 ** | 134 | 0.125 ** | 324 | 0.062 * | 1149 | 0.072 | 700 |
| $F_{N2O}$ | -0.336 ** | 350 | 0.034 | 120 | -0.607 ** | 126 | -0.197 ** | 307 | 0.009 | 1140 | -0.514 ** | 696 |
| H | 0.315 ** | 380 | 0.170 * | 142 | 0.002 | 134 | 0.051 | 324 | -0.021 ** | 1149 | 0.080 * | 700 |
| LE | -0.241 * | 74 | 0.099 | 72 | 0.459 * | 20 | -0.078 | 62 | 0.135 ** | 453 | 0.161 ** | 279 |
| RH | 0.027 | 380 | -0.016 | 142 | -0.057 | 134 | -0.12 ** | 324 | -0.033 | 1149 | -0.041 ** | 700 |
| $T_{air}$ | 0.107 * | 380 | -0.013 | 142 | 0.092 | 134 | 0.249 ** | 324 | 0.138 ** | 1149 | 0.098 ** | 700 |
| $R_{glob}$ | 0.077 | 380 | 0.118 | 142 | -0.096 | 134 | -0.02 | 324 | -0.001 | 1149 | -0.041 ** | 700 |
| $R_{net}$ | 0.011 | 380 | 0.111 | 142 | 0.026 | 134 | 0.087 | 324 | 0.043 | 1149 | -0.053 ** | 700 |
| G | 0.050 | 380 | 0.029 | 142 | 0.121 | 134 | 0.207 ** | 324 | 0.175 ** | 1149 | 0.162 ** | 700 |
| $T_{soil}$ | 0.075 | 380 | -0.146 | 142 | -0.035 | 134 | 0.167 ** | 324 | 0.038 | 1149 | 0.117 ** | 700 |
| $M_{soil}$ | 0.043 | 380 | 0.308 ** | 142 | 0.212 * | 134 | 0.138 * | 324 | 0.093 ** | 1149 | 0.008 | 700 |

**. Correlation is significant at the 0.01 level (2-tailed).

*. Correlation is significant at the 0.05 level (2-tailed).

Table 5. Reported CO fluxes measured in different ecosystems and climatic regions, using chambers (transparent or dark), micrometeorological flux gradient or eddy covariance methods, and the reported data period, measurement frequency and the moment of the measurements.

| Reference | Ecosystem, climate, country | Measurement method | Data period, measurement frequency, moment of measurement | $F_{CO}$ (nmol m$^{-2}$ s$^{-1}$) |
|---|---|---|---|---|
| Zepp et al., 1997 | Black spruce forest, boreal, Manitoba, Canada | Chambers, transparent | 3 months, weekly, daytime | -1.06 |
| Zepp et al., 1997 | Jack pine forest, boreal, Manitoba, Canada | Chambers, transparent | 3 months, weekly, daytime | -0.58 |
| King, 2000 | Pine forest, Northeast, Walpole, Maine, USA | Chambers, dark | 1.3 years, biweekly, daytime | 1.12 |
| King, 2000 | Mixed hardwood-coniferous forest, Walpole, Maine, USA | Chambers, dark | 1.3 years, biweekly, daytime | 0.62 |
| King, 2000 | Pine forest, Griffin, Georgia, USA | Chambers, dark | 1 year, bimonthly, daytime | -0.21 |
| King, 2000 | Pine forest, Tifton, Georgia, USA | Chambers, dark | 1 year, bimonthly, daytime | -0.95 |
| Kuhlbusch et al., 1998 | Black spruce, boreal, Manitoba, Canada | Chambers, dark | 1 year, bimonthly, daytime | -1.11 |
| Galbally et al. 2010 | Mallee, Eucalyptus sp. Ecosystem, tropical, Australia | Chambers, transparent | 1 year, bimonthly, daytime | 0.61 |
| Kisselle et al., 2002 | Cerrado, campo sujo, tropical, Brazil | Chambers, transparent | 1 year, monthly, daytime | 3.16 |
| Kisselle et al., 2002 | Cerrado, stricto sensu, tropical, Brazil | Chambers, transparent | 1 year, monthly, daytime | 2.66 |
| Varella et al., 2004 | Natural cerrado, tropical, Brazil | Chambers, transparent | 1.5 years, monthly, daytime | 1.91 |
| Varella et al., 2004 | Pasture (*Brachiaria brizantha*), tropical, Brazil | Chambers, transparent | 1.5 years, monthly, daytime | 1.20 |
| King, 2000 | Cropland, corn, Walpole, Maine, USA | Chambers, dark | 1.3 years, biweekly, daytime | 2.19 |
| King, 2000 | Cropland, sorghum/wheat, Griffin, Georgia, USA | Chambers, dark | 1 year, bimonthly, daytime | 1.16 |
| King, 2000 | Cropland, cotton/peanuts/winter wheat, Tifton, Georgia, USA | Chambers, dark | 1 year, bimonthly, daytime | 1.03 |
| Galbally et al. 2010 | Cropland, wheat, tropical, Australia | Chambers, transparent | 1 year, bimonthly, daytime | 0.98 |
| Constant et al., 2008 | Grassland, boreal, Quebec, Canada | Flux gradient | 1 year, diurnal cycle | -2.11 |
| Bruhn et al., 2013 | Grassland, temperate, Denmark | Chambers, dark | 2 months, monthly, daytime | -0.78 |
| Bruhn et al., 2013 | Grassland, temperate, Denmark | Chambers, transparent | 2 months, monthly, daytime | 0.36 |
| van Asperen et al., 2015 | Grassland, Mediterranean, Italy | Chambers, transparent | 5 weeks, summer, diurnal cycle | 0.35 |
| van Asperen et al., 2015 | Grassland, Mediterranean, Italy | Flux gradient | 1 month, 30-min, diurnal cycle | 1.74 |
| this study | Grassland, reed canary grass, boreal, Finland | Eddy covariance | 7 months, 30-min, diurnal cycle | -0.25 |

**Figure captions**

Figure 1. (a) Daily mean air and soil temperatures, (b) global radiation sum ($R_{glob}$), (c) daily precipitation sum ($P_r$) and soil water content (SWC), (d) weekly leaf area index (LAI) (black) and green area index (GAI) (grey), (e) net ecosystem exchange of $CO_2$ (NEE), and (f) cumulative CO fluxes calculated from half-hour mean CO fluxes (cum $F_{CO}$; black lines) and daytime mean CO fluxes ($F_{CO\_day}$; grey) over the 7-month measurement period in a reed canary grass crop. Measurement periods (S = spring, ES = early summer, MS = mid-summer, LS = late summer, A = autumn, LA = late autumn) are separated by solid lines.

Figure 2. Diurnal cycle of half-hour mean CO fluxes ($F_{CO}$, nmol m$^{-2}$ s$^{-1}$) from the reed canary grass crop from six distinct periods during the April to November 2011. Grey areas indicate the moment of sunrise and sunset, and the vertical bars indicate ±1 standard deviation of the fluxes.

Figure 3. Diurnal cycle of half-hour mean global radiation ($R_{glob}$, W m$^{-2}$) (black) and soil temperature at 2.5 cm depth (grey) at the reed canary grass crop from six distinct periods during the April to November 2011. The vertical bars indicate ±1 standard deviation of the fluxes and temperatures.

Figure 4. Diurnal cycle of half-hour mean net ecosystem exchange of $CO_2$ (NEE, µmol $CO_2$ m$^{-2}$ s$^{-1}$) from the reed canary grass crop from six distinct periods during the April to November 2011. Grey areas indicate the moment of sunrise and sunset, and the vertical bars indicate ±1 standard deviation of the fluxes.

Figure 5. Daytime half-hour average CO fluxes ($F_{CO}$) against global radiation ($R_{glob}$), sensible heat flux (H) and net ecosystem exchange of $CO_2$ (NEE) measured over two emission periods (Spring, days 110-145, Early Summer, days 146-160) at the reed canary grass crop in Maaninka. The bin averages with ±1 standard deviation are presented in black line.

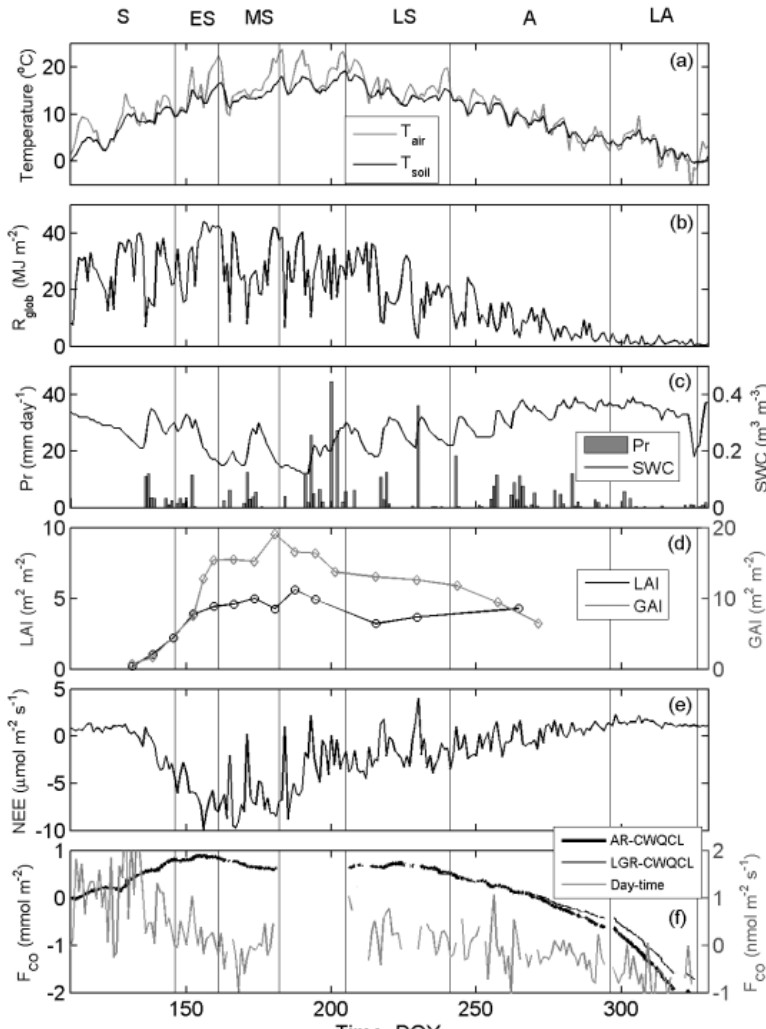

Figure 1. (a) Daily mean air and soil temperatures, (b) global radiation sum ($R_{glob}$), (c) daily precipitation sum ($P_r$) and soil water content (SWC), (d) weekly leaf area index (LAI) (black) and green area index (GAI) (grey), (e) net ecosystem exchange of $CO_2$ (NEE), and (f) cumulative CO fluxes calculated from half-hour mean CO fluxes (cum $F_{CO}$; black lines) and daytime mean CO fluxes ($F_{CO\_day}$; grey) over the 7-month measurement period in a reed canary grass crop. Measurement periods (S = spring, ES = early summer, MS = mid-summer, LS = late summer, A = autumn, LA = late autumn) are separated by solid lines.

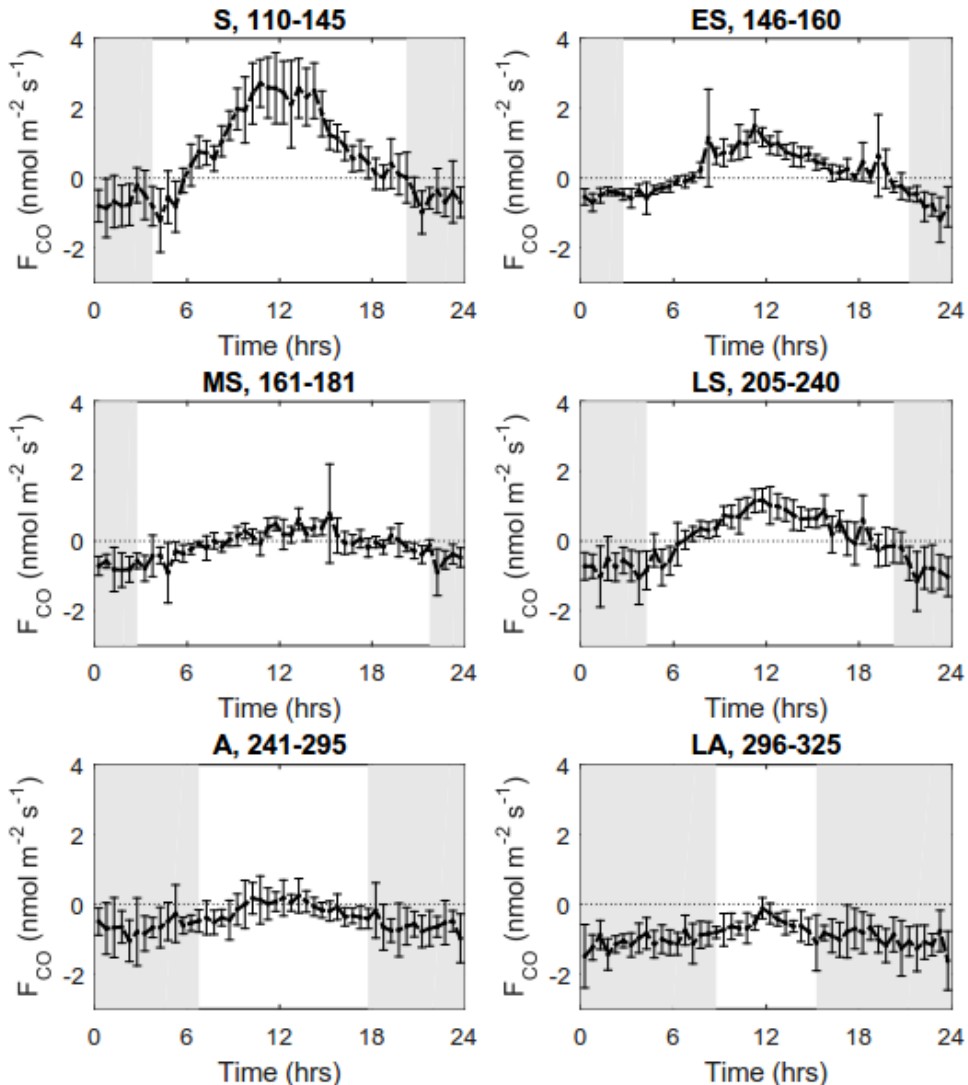

Figure 2. Diurnal cycle of half-hour mean CO fluxes ($F_{CO}$, nmol m$^{-2}$ s$^{-1}$) from the reed canary grass crop from six distinct periods during the April to November 2011. Grey areas indicate the moment of sunrise and sunset, and the vertical bars indicate ±1 standard deviation of the fluxes.

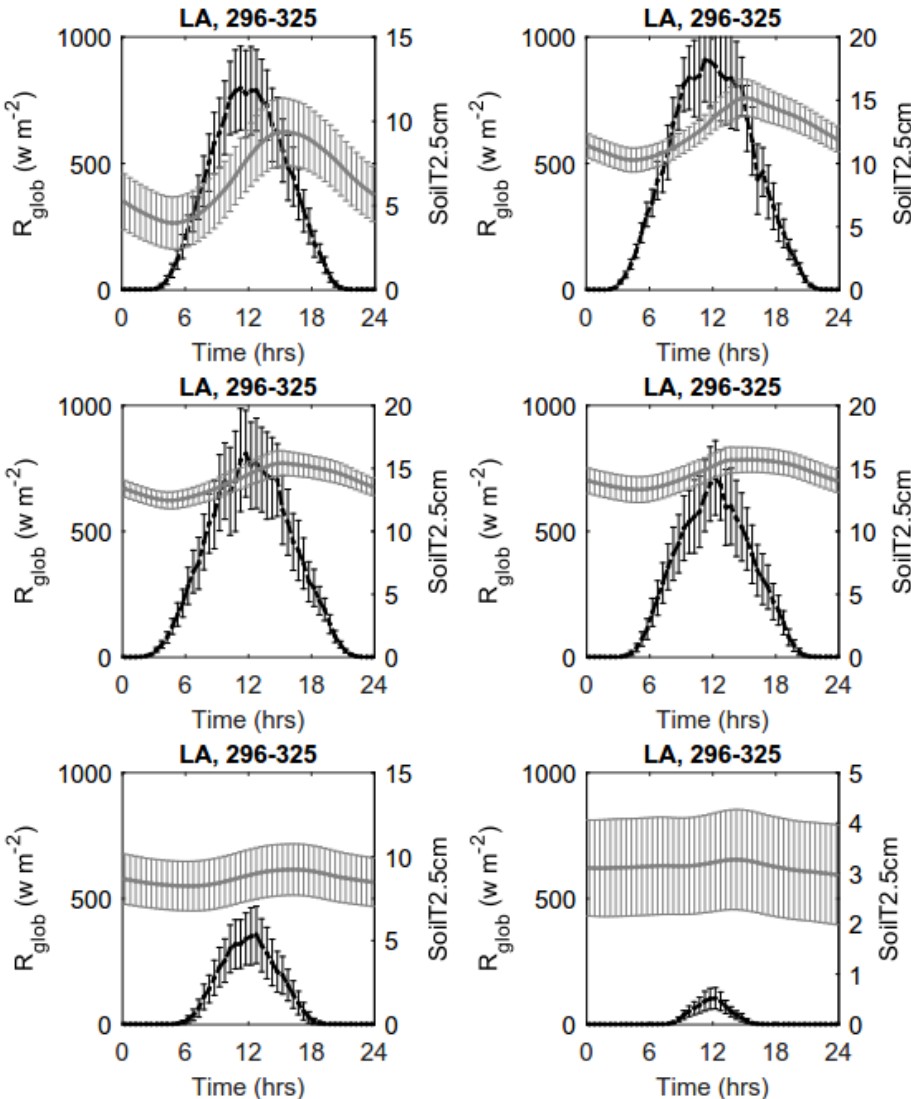

Figure 3. Diurnal cycle of half-hour mean global radiation ($R_{glob}$, W m$^{-2}$) (black) and soil temperature at 2.5 cm depth (grey) at the reed canary grass crop from six distinct periods during the April to November 2011. The vertical bars indicate ±1 standard deviation of the fluxes and temperatures.

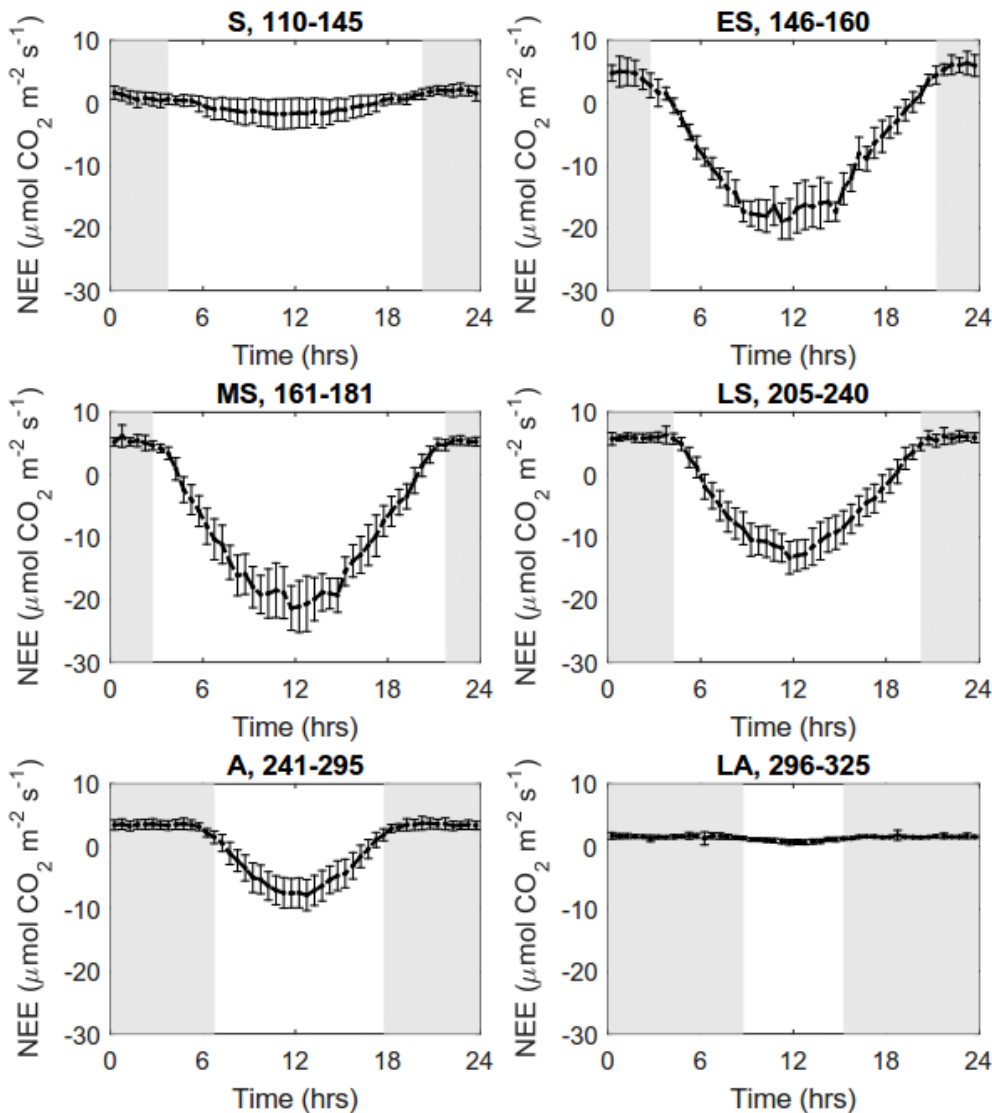

Figure 4. Diurnal cycle of half-hour mean net ecosystem exchange of $CO_2$ (NEE, µmol $CO_2$ m$^{-2}$ s$^{-1}$) from the reed canary grass crop from six distinct periods during the April to November 2011. Grey areas indicate the moment of sunrise and sunset, and the vertical bars indicate ±1 standard deviation of the fluxes.

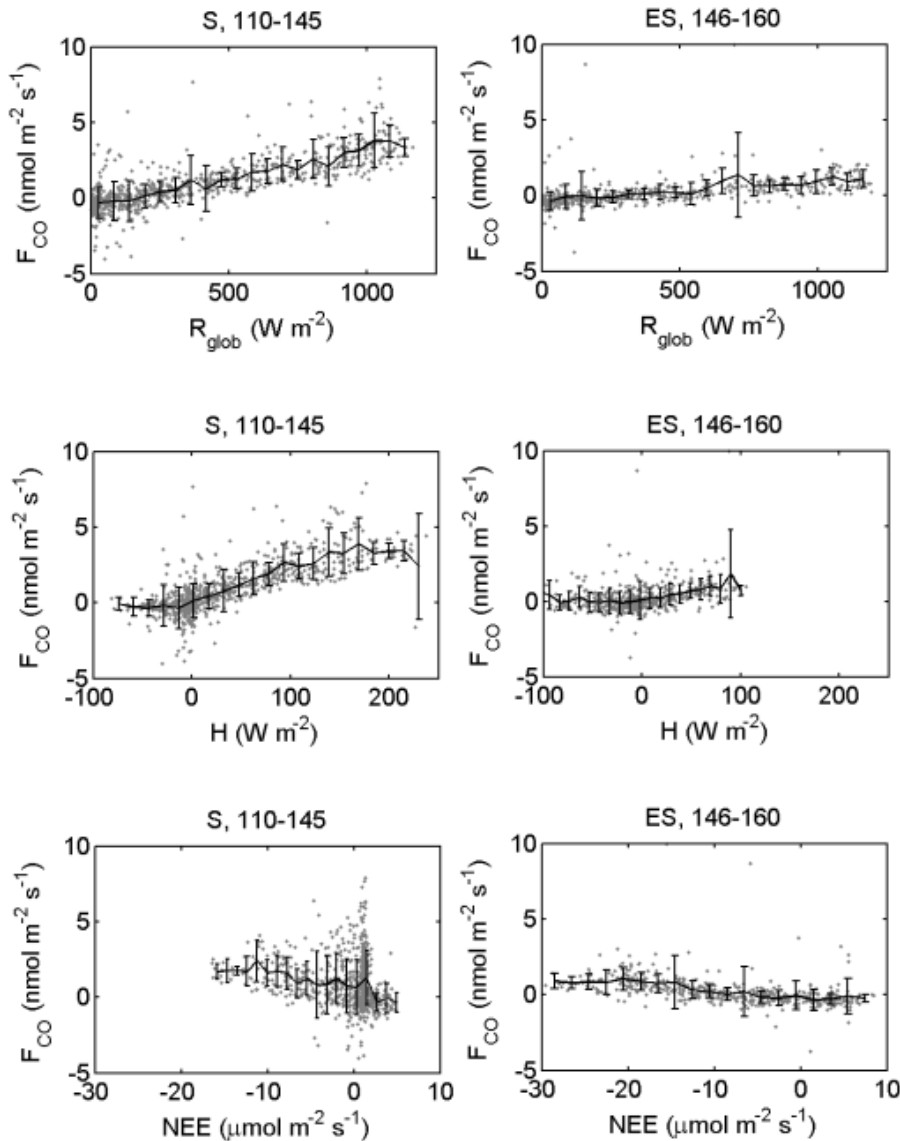

Figure 5. Daytime half-hour average CO fluxes ($F_{CO}$) against global radiation ($R_{glob}$), sensible heat flux (H) and net ecosystem exchange of $CO_2$ (NEE) measured over two emission periods (Spring, days 110-145, Early Summer, days 146-160) at the reed canary grass crop in Maaninka. The bin averages with ±1 standard deviation are presented in black line.