# Peer review of "Seasonal and diurnal variation in CO fluxes from an agricultural bioenergy crop"

_Biogeosciences, 2015_

## Referee Comment (RC1) · Anonymous Referee #1 · 24 Feb 2016

General comments:

The paper is well written and is the first to show the use of the eddy covariance technique for CO flux measurements. It is interesting to see a CO flux dataset with a high temporal resolution over 9 months, which is novel. Quite some studies have shown a diurnal CO cycle before but, as they say, this study is the first one to study the change in diurnal cycle over several months. The paper gives a good overview of how CO fluxes can be predicted or modeled by showing correlation matrices for many different variables, and by showing how the correlation matrices are changing over the season. This is for example very useful information for climate and carbon transport models and therefore a valuable dataset.

However, the interpretation of the dataset is at some points weak. With the Eddy Covariance measurements, they try to answer process level based questions, which their dataset is not fully suitable for. For example, the measured EC CO fluxes are a net result (sum) of several processes (uptake by soil, production by soil, production by dead organic matter and production by livings plants) and each of these uptake and emission processes have their own dependencies on environmental variables. While not being able to separate these sources (and their mechanisms and dependencies), still process level based questions are tried to be answered by use of best fitting correlation matrices, and by use of several assumptions (for example the assumption of stable soil CO uptake). When the paper wants to focus on process level based questions, this approach and its considerations and restrictions should be discussed in more detail. Also some other parts of the dataset interpretation and dataset explanation need some more work. Furthermore, by discussing the limitations of the interpretation part, they can determine interesting (process level) research topics/setups for future CO flux studies, thereby contributing ideas for future CO flux studies.

In general, I would consider this a nice dataset which should be published. However, as said, the interpretation of this dataset is at some points weak and needs to be worked on. The points which should be revised or rewritten are more elaborately described in the 'specific comments' section.

Specific comments:

The terms photodegradation, thermal degradation and abiotic degradation are not always used with care. With the EC method it is hard to separate different uptake and emission processes. Based on correlation coefficients, they conclude that radiation is the main driving factor of CO emission. This conclusion cannot be made based on their data. Radiation has many indirect effects such as on temperature, biological activity, etc. From their data it is hard to conclude whether direct photodegradation, or indirect effects of radiation (such as indirect photodegradation (fragmentation of organic matter) or thermal degradation) are the main cause of the CO production. In some places in the paper, this is well acknowledged (page 11, line 4-8). In other places

this is neglected and the statistical correlation to radiation is given as a proof for direct photodegradation being the main cause. The difference between direct and indirect photodegradation should be explained, and conclusions on this subject should be formulated more carefully.

They make an important (risky) assumption by saying that biological soil CO uptake is constant, based on the paper of Conrad & Seiler (1985). However, other CO flux studies have observed the typical biological temperature response wherein biological activity increases with temperature (for example: Ingersoll 1973, Whalen & Reeburgh (2001), others). Also, especially in cold ecosystems, a small temperature change usually influences biological rates significantly. The assumed stable soil CO uptake assumption in this ecosystem seems unlikely. With the current dataset, this assumption cannot be validated or falsified. So, the authors should reconsider this assumption and think of the consequences if there is a typical temperature response, for example as found by Whalen & Reeburgh (2001), with a Q10 of 2.0. How would this influence their main conclusions? Either this possibility should be discussed, or the 'stable production' assumption should be removed from their manuscript.

Concerning possible biological CO production mechanisms, they hypothesize that CO emissions are not driven by microbial activity. While it is likely that the observed CO emissions are not driven by microbial activity, the used argumentation might be misleading: they base it on the poor correlation between FCO and FN2O, and the poor correlation between FCO and RESP. However, FCO is a net flux (sum) of uptake and production, while RESP and FN2O are solely production fluxes. This makes the validity of the correlation questionable. Also, in case the CO production is caused by biological as well as by abiotic sources, would this not result in the same poor correlation between FCO and FN2O? With the current dataset, it is difficult to determine whether biological fluxes are present, but saying that the poor correlation indicates the absence of biological sources might be misleading. In previous studies, what are the magnitudes of the reported biological fluxes? In this ecosystem, would they have the same magnitude? Are they also expected in autumn when vegetation is less active/dormant/etc? If the authors believe that biological fluxes play a (small) role, it would be good to spend some sentences on the assumed mechanisms and maybe indicate the magnitude of the observed biological fluxes in other studies as a comparison.

In the discussion (page 11, line 21-25), they use the high C to N ratio to confirm their theory that photodegradation is the main cause. However, this argument is not explained. Why does a high C to N ratio confirm the hypothesis?

The comparison to other variables (NEE; heat and energy flux) is stated as a goal in the last paragraph of the introduction, but it is not well explained what is expected. Also, in general the comparison is held very small, especially for the N2O fluxes. The results are only shown in a table but not discussed, and the N2O fluxes are hardly mentioned in the Discussion and forgotten in the Conclusion. For the N2O results, the reader is referred to another paper. If the authors think that the N2O story is an important part, since they state their interest in the introduction, they should show some N2O results, interpret these results, and spend some text on why they expect a correlation. Does a N2O figure maybe fit in Figure 1? Or, if they prefer to refer to the other paper, please then describe the N2O fluxes (magnitude and diurnal variation) briefly in this paper.

Looking at figure 2 and 3, they correctly conclude that not all CO fluxes can be initiated by radiation, since CO fluxes are already increasing when the sun is still down (in autumn), and they refer to the possibility of thermal degradation. With the assumption of stable soil CO uptake during the dark hours, and with the idea of thermal degradation being responsible for the increasing CO production during the morning hours in the dark, is it possible to (roughly) estimate how much thermal degradation is contributing to total CO fluxes? Can this be extrapolated to the day? And does this estimate change when there is no stable soil CO uptake assumed?

The sampling line material is made from PTFE, which is reported to be inert. However, was the whole sampling set up made of PTFE (from inlet to instrument)? Other mate-

rials are known to be possible strong CO emitters, and previous CO flux studies have found CO producing material in setups during blank tests (Schade 1999, van Asperen 2015). Has the setup been tested for internal CO production and have blank tests been performed? If so, please mention this. If there is internal CO emission, it probably wont influence your results largely due to the large sampling flow, but if possible, it would be the best to quantify this.

On Page 6, line 26-27, you estimate that the site is a net sink of CO for the 9 months, which is nicely shown in Table 1. Concerning that you seem to have a good idea of which environmental variables are important per time of the season, and that you are the first one to show a dataset with such high temporal resolution for 9 months, is it possible to give an estimate of the net CO fluxes for the other 3 months, so you can give an annual estimate? Such as done in Table 6 or 7, in the paper of Ingersoll (Soil's potential as a sink for atmospheric carbon monoxide, 1973). That would be an interesting addition.

Technical corrections:

General: - Please check your references, for example, Lee (2012) and Zahniser (2009) are missing in the reference list, but maybe there are more. - For many units in different places in the manuscript (ha-1, m-2), the 'superscript' is not used. - The hyphen is not used consistently throughout the manuscript - Different places in the manuscript: Please use the same term for G (ground heat flux or soil heat flux)

Text: - Page 2, line 2: of a strong greenhouse gas → of the strong greenhouse gas. - Page 2, line 12-17: quite some references are named in the different places in the manuscript, which are not named in this part. It might be nice if the references which are coming back in table 3, also come back here in the right place. - Page 2, line18: Here is stated that CO emissions are thermal or UV-induced. However, it is not only by UV, also by visible radiation, as shown by Lee (2012). Lee (2012) is mentioned on page 9, line 21, but does not appear in reference list. - Page 2, line 22: Most of the reported

[Figure]

CO flux measurements are either short-term field experiments from.... it seems that the author wants to make a point here that no CO measurements are made so far in this cropland boreal ecosystem, or are only made short-term. But neither of the points is clearly made. Is this the first measurement in this ecosystem? Or the first long term? And can there be an indication for which percentage of land use/Finland/boreal zone this ecosystem is representative for? (see paper Ingersoll, 1973, table 6,7) - Page 3, line 20: The footprint length is given, and the size of the field is given. I assume the author wants to implicate that the footprint is homogeneous in all directions, but this is not stated. Clarify for the reader. - Page 3, line 20: A 6.3 ha field is introduced, is this the same field as meant in the rest of §2.1, before this sentence? Not clear formulated. - Page 4, line 16: No white space between '(see Fig.1c)' and 'Considering'. - Page 4, line 15-20: it is stated that in the majority of the measurements is representative for the RCG canopy. What happened to the minority of the data when it is not? Can this be mentioned? - Page 4, line 25: Reference Zahniser is also missing. - Page 4, line 24-26: Unclear and incorrect sentences, please rephrase. - Page 5, line 7: PTFE lines were used, which are under most conditions inert. However, other parts of the used material might not be. Has there been a blank measurement? If so, this should be mentioned. - Page 5, line 11: The measurement position of G and Tsoil etc is not named in the 'Material and Methods', although partly named later in the manuscript. - Page 5, line 21: a verb is missing. Do you mean: LGR-CWQCL measurements were corrected for..... - Page 5, line 22: unclear sentence. Do you mean: the same applied to the AR-CWQCL measurements after software update in July 2011. - Page 6, line 15: while the length of periods were → while the lengths of the periods were. - Page 6, line 25: to the mid-June–> to mid-June. - Page 7, line 15: near constant CO uptake, is the value you found similar to other studies? - Page 7, line 25-26: please mentioned '(days 110-145)' after 'during the spring'. - Page 8, line 1: Here the discussion suddenly jumps from CO to CO2, maybe introduce this a little clearer. - Page 8, line 1: Have LAI and GAI been introduced before? - Page 8, line 1-5: N2O is not discussed at all here. - Page 9, line 2-4: different type of ecosystems are compared here, but they lay

in different climate zone. Does this comparison make sense then? - Page 11, line 1: suggestion: we expect that radiation–> we expect that the effects of radiation. - Page 11, line 3: T soil at a depth of 2.5cm–> this should also be in materials and methods. - Page 11, line 24: mean and stdev are mentioned, however, the mean value is missing. - Page 11, line 26: the early summer emission–> the early summer CO emission.

Tables & Figures: In Table 3, a nice overview is given of previous CO studies. The fluxes which are reported here, are that daily averages? Several of these studies have also measured a daily cycle. Can the magnitude of these results be indicated? That might make the overview more complete.

In Figure 4, NEE is mentioned. I assume this measured by the EC measurements of CO2? Maybe mention the trace gas in the caption, also in other places in the manuscript when mentioning NEE.

---

## Referee Comment (RC2) · Anonymous Referee #2 · 24 Feb 2016

**1   General comments**

The manuscript of Pihlatie et al. on carbon monoxide (CO) flux measurements above an agricultural bioenergy crop (reed canary grass) represents an important study on the biosphere-atmosphere exchange of CO. While previous studies mainly focused on short term measurements of CO fluxes, the authors present the first eddy covariance measurements of CO fluxes over an entire growing season, making it a unique study. Like this, the authors can investigate the dependency of the CO flux on different environmental parameters such as irradiation, temperature, crop cover, fertilization status, etc.

Interestingly, the authors find that the reed canary grass ecosystem acted as a net

source of CO at the beginning and a net sink during of the rest of the growing season. Also, they measured a strong diurnal cycle, as opposed to other previous studies over cropland, with mostly net emission during daytime and net uptake during night. In their study the authors correlate the net CO flux with environmental parameters to obtain an understanding on the controlling processes. As the nature of CO exchange is complex with many possible sinks and sources that have been observed in previous studies, this is challenging. As a consequence, the conclusions made on the underlying processes can often only remain assumptions, and therefore, the study provides only limited new insight into processes of CO exchange. As stated by the authors, further process related studies are necessary for future research.

The authors use state of the art measurement techniques for the quantification of CO fluxes and the fluxes were analyzed according to standard quality control procedures. Furthermore, the manuscript is clearly structured and well written. Due to the unique data set, I suggest the manuscript to be published in BG, after the more specific comments below have been addressed.

**2 Specific comments**

P. 3, L. 16-20: At which day was the crop cultivated? For completeness, I suggest to add this information to this short description of the growing season.

P. 4, L. 21-28: In this paragraph it is not clear that these are the same analyzers as used for the flux $N_2O$ intercomparison in Rannik et al. (2015). It would be good to state this in this manuscript or move the above sentence "The comparison of four laser-based..." to the end of the paragraph.

P. 6, L. 4-7: Here it would be interesting to know, what the magnitude of the CO flux loss was, regarding the given response times of the EC systems. In context of the effect of the inlet lines, it would also be beneficial to mention their inner diameters in

this section. According to Rannik et al. (2015) the reason for the larger response time of the system was caused by laminar flow due to a larger tubing diameter.

P. 6, L. 8-10: As stated here, more data had to be removed during daytime than during night-time. However, especially at night-time flux data has to be often rejected due to insufficiently developed turbulence. For this, a flux quality criterion using e.g. integral turbulence characteristics as suggested by Foken and Wichura (1996) is often applied. Also a test on stationarity, which was not applied for the $N_2O$ fluxes in Rannik et al. (2015) for intercomparison reasons, might be important for CO.

P. 6, L. 22: The results chapter presents the measured CO flux and its correlation with various environmental parameters. In addition, I find it important to also present the CO mixing ratios as they can influence the CO flux significantly. Especially, the amount of CO uptake might be largely dependent on the available atmospheric CO. To rule out the effect of changing atmospheric CO levels on the CO flux when interpreting the results, CO mixing ratios should then also be included in the correlation analysis.

P. 6, L. 22: As it was mentioned in the method section, two different instruments for the CO flux measurements were used. However, in the result section the data from both analyzers is only shown as the cumulative flux in Figure 1f. If two independent analyzer are used, I would expect a paragraph or statement on the comparability of both measurements. This would give a better insight into the associated flux errors and would be also be valuable information for the CO flux community. Looking at the cumulative flux estimates, there seems to be a good agreement between days 205-270, while after that both fluxes seem to differ. Also, it should be stated in the manuscript that the presented fluxes (despite the green cumulative curve) are from the AR-CWQCL instrument while the LGR-CWQCL instrument was only operated from day 205.

P. 7, L. 19-21: As stated, the concept of the gross $F_{CO}$ only holds if the CO uptake can be assumed to be constant over the entire diurnal cycle. However, especially turbulent

transport and transport through the quasi-laminar boundary layer at the surface typically show distinct diurnal cycles with maxima during daytime. Hence, I would expect the CO uptake to increase during the day, unless the CO uptake is limited mainly by soil microbial consumption or transport in soil (then, the CO flux would also mainly be independent from above surface CO-concentrations, which would change during day). Is there more evidence that can support the assumption of a constant CO uptake? The authors note that there is evidence from previous studies that the temperature effect on microbial consumption can be assumed to be small. In my opinion it should also be shown that the CO uptake is mainly limited by soil microbial consumption or transport in soil for the assumption of a constant CO uptake to be valid. Otherwise, the diurnal variation in the aerodynamic and the quasi-laminar boundary layer resistances would have to be taken into account. In general, the use of a bi-directional exchange model would be useful to address the issue of flux partitioning and importance of soil uptake, although I understand that this is challenging given the lack of detailed process studies on CO exchange and might be the scope of future research.

P. 8, L. 8-16: What was the applied definition for daytime and night-time periods? This is valuable information, as the correlation values are often largely dependent on the variation of the used parameters, which are typically larger during daytime. In this context, it might be also valuable to mention if the flux error had an impact on the weak correlations found during night-time.

P. 9, L. 11-13: To correct for this bias, a gap-filling method can be applied for the calculation of cumulative CO fluxes.

P. 9, L. 14-15: As $F_{CO}$ describes the net CO flux, one should differentiate here more explicitly between the emission component and uptake component of the flux. Otherwise the reader may assume you are referring to the net emission/uptake.

**3   Technical comments**

P. 3, L. 4: Write "reed canary grass" instead of "read canary grass". Correct also on P. 13, L. 9 and 19, L. 1.

P. 3, L. 13: Omit space after "27°" or introduce after all units (°, ', "). Use same degree sign as used in L. 15.

P. 3, L. 17: Use superscript for "-1" in "ha-1".

P. 4, L. 10: Shouldn't it be "L=+-100 m" for the definition of the near-neutral range?

P. 4, L. 16: Insert space before "Considering".

P. 4, L. 26: Write "LGR-CWQCL" instead of "LGRCW-QCL" as in the rest of the manuscript.

P. 6, L. 1: Do you intentionally differentiate between "co-variances" (here and L. 5) and "covariance"?

P. 6, L. 10: Write "daytime" instead of "day-time" as in the rest of the manuscript. Correct also on P. 7, L. 15 and on P. 9, L. 24.

P. 6, L. 27: I suggest using "over the 9-month measurement period" instead of "in the end of the 9-month measurement period" as the used expression could be misleading otherwise.

P. 7, L. 17: Use superscript in units.

Figures 2-5: Instead of using the day of year numbers, I suggest to use the introduced classification of S, ES, MS . . . in the subplot titles (or use both, DOY + the classification). This makes it easier to compare with Figure 1 and descriptions in the text.

**4 References**

Foken, T. and Wichura, B.: Tools for quality assessment of surface-based flux measurements, Agric. For. Meteorol., 78(1-2), 83–105, doi:10.1016/0168-1923(95)02248-1, 1996.

Rannik, Haapanala, S., Shurpali, N. J., Mammarella, I., Lind, S., Hyvönen, N., Peltola, O., Zahniser, M., Martikainen, P. J. and Vesala, T.: Intercomparison of fast response commercial gas analysers for nitrous oxide flux measurements under field conditions, Biogeosciences, 12(2), 415–432, doi:10.5194/bg-12-415-2015, 2015.

---

## Author Comment (AC1) · 25 May 2016

Response to the reviewer comments Dear Dr. Ivonne Trebs, and two reviewers,

Referee #1: General comments: The paper is well written and is the first to show the use of the eddy covariance technique for CO flux measurements. It is interesting to see a CO flux dataset with a high temporal resolution over 9 months, which is novel. Quite some studies have shown a diurnal CO cycle before but, as they say, this study is the first one to study the change in diurnal cycle over several months. The paper gives a good overview of how CO fluxes can be predicted or modeled by showing correlation matrices for many different variables, and by showing how the correlation matrices are changing over the season. This is for example very useful information for climate and carbon transport models and therefore a valuable dataset.

[Figure]

However, the interpretation of the dataset is at some points weak. With the Eddy Co-variance measurements, they try to answer process level based questions, which their dataset is not fully suitable for. For example, the measured EC CO fluxes are a net result (sum) of several processes (uptake by soil, production by soil, production by dead organic matter and production by livings plants) and each of these uptake and emission processes have their own dependencies on environmental variables. While not being able to separate these sources (and their mechanisms and dependencies), still process level based questions are tried to be answered by use of best fitting correlation matrices, and by use of several assumptions (for example the assumption of stable soil CO uptake). When the paper wants to focus on process level based questions, this approach and its considerations and restrictions should be discussed in more detail. Also some other parts of the dataset interpretation and dataset explanation need some more work. Furthermore, by discussing the limitations of the interpretation part, they can determine interesting (process level) research topics/setups for future CO flux studies, thereby contributing ideas for future CO flux studies.

In general, I would consider this a nice dataset which should be published. However, as said, the interpretation of this dataset is at some points weak and needs to be worked on. The points which should be revised or rewritten are more elaborately described in the 'specific comments' section.

We want to thank the reviewer for these constructive and important comments. We have now addressed the general concern of how to interpret the results with the current data set. We have added more discussion about challenges in separating the different processes of CO exchange, and restrictions of this study setup in addressing process level issues. We have also estimated the effect of temperature dependency of the CO uptake, as requested, and we have carefully edited the text so that we do not over interpret the results related to CO forming processes. Furthermore, we moved part of the process level discussion to the end of the paper to suggest ideas for future research topics on CO exchange.

Specific comments: The terms photodegradation, thermal degradation and abiotic degradation are not always used with care. With the EC method it is hard to separate different uptake and emission processes. Based on correlation coefiňĄcients, they conclude that radiation is the main driving factor of CO emission. This conclusion cannot be made based on their data. Radiation has many indirect effects such as on temperature, biological activity, etc. From their data it is hard to conclude whether direct photodegradation, or indirect effects of radiation (such as indirect photodegradation (fragmentation of organic matter) or thermal degradation) are the main cause of the CO production. In some places in the paper, this is well acknowledged (page 11, line 4-8). In other places this is neglected and the statistical correlation to radiation is given as a proof for direct photodegradation being the main cause. The difference between direct and indirect photodegradation should be explained, and conclusions on this subject should be formulated more carefully.

Thank you for taking this topic up. Indeed, we agree that we have not been consistent with the use of the terms photodegradation, thermal degradation and abiotic degradation. We have now used more space to explain these different mechanisms, including direct and indirect photodegradation. We also aim at not overstating process level drivers of CO fluxes as our data does not allow us to conclude this. We have now carefully gone through the text and modified it so that we do not make too strong statements or process level conclusions based on our results.

They make an important (risky) assumption by saying that biological soil CO uptake is constant, based on the paper of Conrad & Seiler (1985). However, other CO flux studies have observed the typical biological temperature response wherein biological activity increases with temperature (for example: Ingersoll 1973, Whalen & Reeburgh (2001), others). Also, especially in cold ecosystems, a small temperature change usually influences biological rates significantly. The assumed stable soil CO uptake assumption in this ecosystem seems unlikely. With the current dataset, this assumption cannot be validated or falsified. So, the authors should reconsider this assumption

and think of the consequences if there is a typical temperature response, for example as found by Whalen & Reeburgh (2001), with a Q10 of 2.0. How would this influence their main conclusions? Either this possibility should be discussed, or the 'stable production' assumption should be removed from their manuscript.

We agree that the use of an assumption of a stable soil CO uptake was too simplistic, and as reported by e.g. Ingersoll (1973) and Whalen and Reeburgh (2001) not correct. Assuming a temperature response of Q10 of 1.8 for the CO uptake reported by Whalen & Reeburgh (2001), we estimated the daytime CO uptake from the nighttime net CO fluxes and air temperatures. We used soil temperature at 2.5 cm depth in the calculation as we considered that this is closest to the location where microbial CO consumption takes place. Hence, we assumed that the night-time CO fluxes (near constant negative fluxes) result from microbial CO consumption, which has a temperature response. The resulting daytime CO uptake estimated for each measurement period (Summer, Early Summer, Mid-Summer, Late Summer, Autumn, Late Autumn) allowed us to estimate the gross CO production during daytime, which is the difference between net daytime CO flux and daytime CO uptake. These results are now reported in two tables: Table 1. The mean, median and 25-75th percentiles of the net CO fluxes, net daytime CO fluxes and net night-time CO fluxes for each measurement period. Table 2. The daytime and night-time air temperatures, daytime CO uptake (using Q10 of 1.8), and gross daytime CO emission.

Based on these calculations, we find that the daytime CO uptake is almost twice as high as the night-time CO uptake. This is further reflected in significantly higher gross daytime CO emissions. We have added discussion on the diurnal variation in CO uptake as well as the effect of this to the gross CO emissions, which is now largely overcome by the high CO uptake.

Concerning possible biological CO production mechanisms, they hypothesize that CO emissions are not driven by microbial activity. While it is likely that the observed CO emissions are not driven by microbial activity, the used argumentation might be misleading: they base it on the poor correlation between FCO and FN2O, and the poor correlation between FCO and RESP. However, FCO is a net ïñĆux (sum) of uptake and production, while RESP and FN2O are solely production ïñĆuxes. This makes the validity of the correlation questionable. Also, in case the CO production is caused by biological as well as by abiotic sources, would this not result in the same poor correlation between FCO and FN2O? With the current dataset, it is difïñĄcult to determine whether biological ïñĆuxes are present, but saying that the poor correlation indicates the absence of biological sources might be misleading. In previous studies, what are the magnitudes of the reported biological ïñĆuxes? In this ecosystem, would they have the same magnitude? Are they also expected in autumn when vegetation is less active/dormant/etc? If the authors believe that biological ïñĆuxes play a (small) role, it would be good to spend some sentences on the assumed mechanisms and maybe indicate the magnitude of the observed biological ïñĆuxes in other studies as a comparison.

We agree with the reviewer that the used argumentation in explaining the CO production mechanisms was not sufficient. We also agree that the poor correlation between FCO and FN2O, or FCO and RESP does not prove that the CO emissions are not driven by microbial activity. We have modified this chapter to include discussion on the reasons why we do not expect the CO formation to be of microbial origin (Conrad and Seiler, 1980), and to give a better understanding of the abiotic and biotic CO production mechanisms. We also discuss the connections between FOC and FN2O, and FCO and RESP, and possible reasons why we did not find significant correlations. We also added more discussion on the current understanding of biological CO production, the seasonality in biological CO fluxes, and whether biological CO production could significantly contribute to CO fluxes in our agricultural ecosystem.

In the discussion (page 11, line 21-25), they use the high C to N ratio to conïñĄrm their theory that photodegradation is the main cause. However, this argument is not explained. Why does a high C to N ratio conïñĄrm the hypothesis? After going through

the text in the manuscript and reconsidering the reviewer suggestions, we do understand how naïve it was to suggest photodegradation as the main process leading to CO emissions. We understand that abiotic CO formation is a result of both photodegradation and thermal degradation, which cannot be fully separated in field experiments due to e.g. indirect effects radiation. Currently, there are studies supporting for higher contribution of photodegradation to the CO formation (e.g. Lee et al., 2012), but also studies suggesting thermal degradation as the dominant CO forming process (e.g. van Asperen et al., 2015). Related to the C to N ratio of the plant material, a meta-analysis shows that CO formation via photodegradation increases with C to N ratio of the plant material (King et al., 2012). Also, As the plant material in our measurement site has a high C to N ratio, and as this dry plant material was well exposed to radiation in the spring, we expect that the conditions were suitable for CO formation via photodegradation. However, this does not confirm that photodegradation was the dominant process at our site, nor does it exclude thermal degradation to take place.

We have now carefully gone through the text, including this paragraph, to discuss more generally the combined effect of photodegradation and thermal degradation. Instead of searching for one specific process, we consider that it is more important to discuss the combined effect of abiotic processes (photodegradation and thermal degradation).

The comparison to other variables (NEE; heat and energy flux) is stated as a goal in the last paragraph of the introduction, but it is not well explained what is expected. Also, in general the comparison is held very small, especially for the N2O fluxes. The results are only shown in a table but not discussed, and the N2O fluxes are hardly mentioned in the Discussion and forgotten in the Conclusion. For the N2O results, the reader is referred to another paper. If the authors think that the N2O story is an important part, since they state their interest in the introduction, they should show some N2O results, interpret these results, and spend some text on why they expect a correlation. Does a N2O figure maybe fit in Figure 1? Or, if they prefer to refer to the other paper, please then describe the N2O fluxes (magnitude and diurnal

variation) briefly in this paper.

We appreciate this concern. We have added text to clarify why we would expect correlations between FCO and other measured variables. With some of them (heat flux, radiation, energy flux) it is clear based on process understanding and previous studies stating that CO emissions are driven by radiation and temperature, however, with some of the measured scalars (e.g. NEE, RESP) we did not know what to expect as there is very limited information available on the links between them and FCO. Based on understanding of biological CO formation, a positive correlation between FCO and NEE would indicate involvement of a biological component in the FCO, hinting towards biological CO production. With respect to FN2O, we would not expect a strong relationship with FCO measured in the field due to the difficulties in separating between overlapping abiotic CO production, microbial CO uptake (Conrad and Seiler, 1980; Moxley and Smith 1998), and microbial N2O production/uptake in the soil. Nitrifiers are among a diverse microbial community oxidizing CO in soils (Jones and Morita, 1983; King and Weber, 2007). Hence high nitrification activity may be reflected in higher CO oxidation in the soil. However, in the field, this is difficult to distinguish as the CO uptake and emission processes take place simultaneously and may cancel each other out. In our study site, no microbial community structure analysis was conducted, however, denitrification was suggested as the dominant N2O forming process especially during high-flux period in the spring and early summer (Shurpali et al., 2016). During the background flux period (days 206-280) the N2O fluxes are small due to low N availability indicating also low nitrification and denitrification activities. In order to distinguish between nitrifier driven CO consumption, microbial community analysis should have been conducted parallel to laboratory studies focusing on CO uptake in controlled conditions. Based on this, we did not want to add a new figure of FN2O, and we considered Figure 1 to be already very tight and have no space for additional scalars. We added more description of the FN2O dynamics, and referred to the recently accepted paper reporting the diurnal variability in FN2O (Shurpali et al., 2016).

Jones, R.D., Morita, R.Y.: Carbon monoxide oxidation by chemolithotrophic ammonium oxidisers. Can. J. Microbiol. 29, 1545-1551, 1983.

King, G. M., and Weber, C. F.: Distribution, diversity and ecology of aerobic CO-oxidizing bacteria. Nature Reviews, Microbiology, 5, 107-118, 2007.

Moxley, J.M., and Smith, K.A.: Carbon monoxide production and emission by some Scottish soils. Tellus, 50B, 151-162, 1998.

Shurpali, N. J. et al. Neglecting diurnal variations leads to uncertainties in terrestrial nitrous oxide emissions. Sci. Rep. 6, 25739; doi: 10.1038/srep25739 (2016).

Looking at figure 2 and 3, they correctly conclude that not all CO fluxes can be initiated by radiation, since CO fluxes are already increasing when the sun is still down (in autumn), and they refer to the possibility of thermal degradation. With the assumption of stable soil CO uptake during the dark hours, and with the idea of thermal degradation being responsible for the increasing CO production during the morning hours in the dark, is it possible to (roughly) estimate how much thermal degradation is contributing to total CO fluxes? Can this be extrapolated to the day? And does this estimate change when there is no stable soil CO uptake assumed?

In order to estimate how much a temperature increase in the morning hours would contribute to the CO production via thermal degradation, we used a Q10-value of 2.1 (van Asperen et al., 2015) to estimate. At first we calculated the temperature differences at 2.5 cm depth in the soil between mid-night and morning hours just before the sunrise, when sun elevation angle became positive, during the six measurement periods. We found that the soil temperature decreased (0.1 to 1 ⁰C) from mid-night to the morning hours. Similar trend was observed also in air temperature. This phenomena was consistent throughout the whole 7-month measurement campaign despite the fact that the time of sunrise changed markedly between the seasons (very short nights during the summer). The attached figure 1 illustrates the diurnal cycle in mean soil temperature for each of the six measurement periods together with sun elevation angle data.

The graphs shows how the soil temperature still continues to decrease even after the sunset, when sun elev is above zero.

See pdf-file: Figure 1. Mean sun elevation angle (h<0 night-time, h>0 daytime) and mean soil temperature at 2.5 cm depth at the reed canary grass crop over the six distinct measurement periods (S = Spring, ES = Early Summer, MS = Mid-Summer, LS = Late Summer, A = Autumn, LA = Late Autumn).

As a result, we do not expect temperature driven CO production via thermal degradation to take place in the early morning hours. As seen in the Fig. 2 and 3. and as stated by the referee, the CO fluxes are already increasing when the sun is still down. Hence, the increase in net CO fluxes during the morning hours indicates that CO production from an unspecified process increases, or CO uptake decreases during the morning hours, or that both of these take place. As explained above, we do not expect thermal degradation to be responsible for increased CO production, however, we can speculate that the CO uptake was affected by the decreasing temperature, as it was earlier estimated the CO uptake is temperature driven (see above, Q10 of 1.8). To conclude, based on this analysis, we cannot estimate the role of thermal degradation to the CO production at this site. We have added discussion concerning this, and the challenges to separate thermal degradation from the CO production. We also added discussion on the possible effect of night-time temperature variation on CO uptake, partly explaining the increasing net CO flux during the early morning hours.

The sampling line material is made from PTFE, which is reported to be inert. However, was the whole sampling set up made of PTFE (from inlet to instrument)? Other materials are known to be possible strong CO emitters, and previous CO flux studies have found CO producing material in setups during blank tests (Schade 1999, van Asperen 2015). Has the setup been tested for internal CO production and have blank tests been performed? If so, please mention this. If there is internal CO emission, it probably wont influence your results largely due to the large sampling flow, but if possible, it would be the best to quantify this.

All the sampling lines were made of PTFE, and due to the high flow rates in the sampling line, we did not expect internal CO production in the tubing, inlet or the whole measurement setup. We did not conduct material tests for the measurement setup we present in this paper, however, we have conducted extensive material tests with one of the analyzer (LGR-CW-QCL) and we have found that most of the common tube materials made of Teflon (FEP, PFA), or Nylon and Polyurethane release CO, and that this rate of release depends on temperature and radiation (unpublished data). We consider this is critically important in systems when a sample is accumulated within a system and when there is no constant flow through the system, such as static soil chambers. We added a sentence in the Materials & Methods (page 5) stating that PTFE tubing was found inert with respect to CO under constant-flow setup with the LGR-CW-QCL analyzer (unpublished data).

On Page 6, line 26-27, you estimate that the site is a net sink of CO for the 9 months, which is nicely shown in Table 1. Concerning that you seem to have a good idea of which environmental variables are important per time of the season, and that you are the fi̧rst one to show a dataset with such high temporal resolution for 9 months, is it possible to give an estimate of the net CO fluxes for the other 3 months, so you can give an annual estimate? Such as done in Table 6 or 7, in the paper of Ingersoll (Soil's potential as a sink for atmospheric carbon monoxide, 1973). That would be an interesting addition.

We also think a full annual balance of FCO would be very interesting. Our measurements cover the snow-free period of little more than 7 months (not full 9 months as stated earlier). Based on our measurements, the FCO was rather constant during the autumn and late autumn, but very variable during the spring right after the snow melt when the measurements started. It is very probable that the FCO are minimal during the snow-cover period in December-February, as temperatures and radiation are low and we can expect rather small microbial CO consumption activity in the soil. However, for the spring period during the snow-melt in March-April,

the assumption of small FCO does not necessarily hold as the amount of radiation and temperature increase and the soil surface is freed from the snow allowing the old previous year's crop residues to decompose. Hence, we expect that the use of the mean FCO from the measurement period probably underestimates the FCO during the early spring period. Nevertheless, we performed a back-of-envelope calculation assuming a mean FCO over the whole measurement campaign of -0.25 nmol m2- s-1 to apply for the missing period of day 326 – day 109 (22 November 2011 - 18 April 2012). This results in an annual net cumulative FCO of -111 mg CO m-2 yr-1. When we further extrapolated this to the grassland area in Finland (in total 14891 km2 (Eurostat, statistics, http://ec.europa.eu/eurostat/statistics-explained/index.php/File:Land_cover,_2012_LUCAS2012.png), we obtained a CO sink of -1649 tons CO yr-1. This estimate is slightly less but similar in magnitude as that produced by the model by Potter et al. (1996) for tundra, boreal and temperate zone soils, which we consider more realistic than the estimate by Ingersoll (1974), which is based on data from laboratory experiments with above ambient CO concentrations. We have added our annual FCO estimates, discussed their uncertainties and compared them to literature values in the discussion part of the manuscript.

Having done this exercise, it is easy to say that more high-resolution measurements are needed to cover the whole seasonal cycle in CO exchange, and to obtain a reliable estimate for annual CO balance from boreal ecosystems.

Technical corrections: General: - Please check your references, for example, Lee (2012) and Zahniser (2009) are missing in the reference list, but maybe there are more. - For many units in different places in the manuscript (ha-1, m-2), the 'superscript' is not used. - The hyphen is not used consistently throughout the manuscript - Different places in the manuscript: Please use the same term for G (ground heat flux or soil heat flux) Thank you for spotting these. We will go through them carefully.

Text: - Page 2, line 2: of a strong greenhouse gas → of the strong greenhouse gas. Corrected.

- Page 2, line 12-17: quite some references are named in the different places in the manuscript, which are not named in this part. It might be nice if the references which are coming back in table 3, also come back here in the right place. We agree. We will check that all relevant references are added here.

- Page 2, line18: Here is stated that CO emissions are thermal or UV-induced. However, it is not only by UV, also by visible radiation, as shown by Lee (2012). Lee (2012) is mentioned on page 9, line 21, but does not appear in reference list. Thank you for spotting this. We have added Lee (2012) to the reference list, and also we elaborated a little more that also visible radiation may induce CO emissions.

- Page 2, line 22: Most of the reported CO flux measurements are either short-term field experiments from.... it seems that the author wants to make a point here that no CO measurements are made so far in this cropland boreal ecosystem, or are only made short-term. But neither of the point is clearly made. Is this the first measurement in this ecosystem? Or the first long term? We wanted to in the first place point out that this is the first long-term study reporting FCO from any ecosystem. We clarified this in the text.

And can there be an indication for which percentage of land use/Finland/boreal zone this ecosystem is representative for? (see paper Ingersoll, 1973, table 6,7) We have added information that the site can be classified as a grassland and as follows, FCO can be estimated for the grassland area of Finland. We also added information concerning the cultivation area of RCG crops as comparison to the area of grasslands generally. Based on the mean annual FCO estimated for our RCG crop, we estimated the FCO for grasslands in Finland as was done in Ingersoll 1974.

- Page 3, line 20: The footprint length is given, and the size of the field is given. I assume the author wants to implicate that the footprint is homogeneous in all directions, but this is not stated. Clarify for the reader. We clarified that the footprint is homogenous in all directions.

- Page 3, line 20: A 6.3 ha field is introduced, is this the same field as meant in the rest of §2.1, before this sentence? Not clear formulated. This sentence was rewritten to clarify that the study field was 6.3 ha in size.

- Page 4, line 16: No white space between '(see Fig.1c)' and 'Considering'. Corrected.

- Page 4, line 15-20: it is stated that in the majority of the measurements is representative for the RCG canopy. What happened to the minority of the data when it is not? Can this be mentioned? We assumed homogeneous canopy in all directions i.e. the estimated footprint extent is applicable to all directions. We estimated that the upwind distance contributing 80% of flux under stable conditions (L = +10 m), in case of low canopy, was 166 m. We use this as a very conservative estimate because low canopy existed only a short time period in the beginning of the campaign. For high canopy the respective distance was estimated to be 60 m. Therefore, considering minimum fetch in South direction 135 m, we concluded that fetch was sufficient under majority of observation conditions. No data rejection according to footprint estimation was done. We modified the sentence to be more clear.

- Page 4, line 25: Reference Zahniser is also missing. Corrected.

- Page 4, line 24-26: Unclear and incorrect sentences, please rephrase. This sentence was clarified, and more information was given of the two analyzers used, the time periods they were used, and justification why data from one of them only was used when analyzing the seasonal and diurnal variability in the FCO. Reviewer #2 was also asking for more information on the instrument comparison. See more details of that in the response to Reviewer #2.

- Page 5, line 7: PTFE lines were used, which are under most conditions inert. However, other parts of the used material might not be. Has there been a blank measurement? If so, this should be mentioned. We did not perform blank tests during this measurement campaign as we assumed that the PTFE lines were not a significant source of CO, and because potential CO emissions from the materials can be largely

avoided in the high-flow setup. We acknowledge that many materials, including FEP, PFA and Nylon may emit large quantities of CO, especially in static systems with minimal flow speeds (unpublished data in laboratory experiments using the LGR-CW-QCL analyzer (model N2O/CO-23d, Los Gatos Research Inc.). However, in a laboratory experiment testing PTFE tubing at flow rates of 2.5 L/min, we did not find significant CO production in the tubing system.

- Page 5, line 11: The measurement position of G and Tsoil etc is not named in the 'Material and Methods', although partly named later in the manuscript. This information is now added to the materials and methods section.

- Page 5, line 21: a verb is missing. Do you mean: LGR-CWQCL measurements were corrected for..... We mean that the LGR-CW-QCL analyzer itself corrects for the water vapor effect (an inbuilt correction algorithm). This was clarified.

– Page 5, line 22: unclear sentence. Do you mean: the same applied to the AR-CWQCL measurements after software update in July 2011. Yes, the same spectroscopic correction was applied to the AR-CW-QCL measurements after the software update. We clarified this part in the text.

- Page 6, line 15: while the length of periods were → while the lengths of the periods were. Corrected.

- Page 6, line 25: to the mid-June–> to mid-June. Corrected.

- Page 7, line 15: near constant CO uptake, is the value you found similar to other studies? We have added comparison of night-time CO fluxes measured in this study to those observed in other studies.

- Page 7, line 25-26: please mentioned '(days 110-145)' after 'during the spring'. Corrected.

- Page 8, line 1: Here the discussion suddenly jumps from CO to CO2, maybe introduce this a little clearer. We added introducing sentence to the start of the chapter

stating that compared to the FCO, the net CO2 exchange, expressed as net ecosystem exchange (NEE) was very small during the spring.

- Page 8, line 1: Have LAI and GAI been introduced before? Indeed, these variables were not introduced before. As also many other supporting measurements were not described earlier, we introduced a new chapter (2.3) describing "meteorological, soil and crop variables" and the methods used in measuring them.

- Page 8, line 1-5: N2O is not discussed at all here. We added a section describing the diurnal variation in N2O fluxes during the measurement period.

- Page 9, line 2-4: different type of ecosystems are compared here, but they lay in different climate zone. Does this comparison make sense then? We reconsidered this comparison and modified the text to explain similarities between measurement from the same ecosystem types, and on the other hand, similarities between measurements in the same climatic zone. The variation in CO fluxes seems to be so high, and continuous or long-term measurements so rare that it is difficult to see trends between different ecosystem types measured in the same climatic zone, or trends between different climatic zones with respect to CO fluxes from the same ecosystem type.

- Page 11, line 1: suggestion: we expect that radiation–> we expect that the effects of radiation. Corrected.

– Page 11, line 3: T soil at a depth of 2.5cm–> this should also be in materials and methods. We added a description in the materials and methods (section 2.3)

- Page 11, line 24: mean and stdev are mentioned, however, the mean value is missing. Maybe there was a printing error as in our version the mean was visible. Anyhow, this was checked to make sure that the numbers appear there.

- Page 11, line 26: the early summer emission–> the early summer CO emission. Corrected.

Tables & Figures: In Table 3, a nice overview is given of previous CO studies. The

fluxes which are reported here, are that daily averages? Several of these studies have also measured a daily cycle. Can the magnitude of these results be indicated? That might make the overview more complete. Indeed, this information would be very valuable to the scientific community. We modified the Table 3 to include information of daytime and night-time CO fluxes whenever this information was available.

In Figure 4, NEE is mentioned. I assume this measured by the EC measurements of $CO_2$? Maybe mention the trace gas in the caption, also in other places in the manuscript when mentioning NEE. We added a description of the NEE (net ecosystem exchange of $CO_2$) in the Figure 4 legend, and also in the text where NEE was mentioned (e.g. chapter 3.2 Diurnal variation).

Please also note the supplement to this comment:
http://www.biogeosciences-discuss.net/bg-2015-622/bg-2015-622-AC1-supplement.pdf

---

## Author Comment (AC2) · 25 May 2016

Response to the reviewer comments Dear Dr. Ivonne Trebs, and two reviewers, Referee #2: 1 General comments

The manuscript of Pihlatie et al. on carbon monoxide (CO) flux measurements above an agricultural bioenergy crop (reed canary grass) represents an important study on the biosphere-atmosphere exchange of CO. While previous studies mainly focused on short term measurements of CO fluxes, the authors present the first eddy covariance measurements of CO fluxes over an entire growing season, making it a unique study. Like this, the authors can investigate the dependency of the CO flux on different environmental parameters such as irradiation, temperature, crop cover, fertilization status, etc. Interestingly, the authors find that the reed canary grass ecosystem acted

as a net source of CO at the beginning and a net sink during of the rest of the growing season.

Also, they measured a strong diurnal cycle, as opposed to other previous studies over cropland, with mostly net emission during daytime and net uptake during night. In their study the authors correlate the net CO flux with environmental parameters to obtain an understanding on the controlling processes. As the nature of CO exchange is complex with many possible sinks and sources that have been observed in previous studies, this is challenging. As a consequence, the conclusions made on the underlying processes can often only remain assumptions, and therefore, the study provides only limited new insight into processes of CO exchange. As stated by the authors, further process related studies are necessary for future research.

The authors use state of the art measurement techniques for the quantification of CO fluxes and the fluxes were analyzed according to standard quality control procedures. Furthermore, the manuscript is clearly structured and well written. Due to the unique data set, I suggest the manuscript to be published in BG, after the more specific comments below have been addressed.

We want to sincerely thank the reviewer for constructive comments that help to improve the quality of the manuscript. We have carefully addressed all the comments and responded to them as follows. We acknowledge the concern of risks in process-based interpretations of the results. We hope the corrections will satisfy these concerns and underline the future research needs and gaps in knowledge in this field of research.

2 Specific comments

P. 3, L. 16-20: At which day was the crop cultivated? For completeness, I suggest to add this information to this short description of the growing season. The crop cultivation date was added to the Materials and Methods section.

P. 4, L. 21-28: In this paragraph it is not clear that these are the same analyzers as

used for the flux N2O intercomparison in Rannik et al. (2015). It would be good to state this in this manuscript or move the above sentence "The comparison of four laser-based. . ." to the end of the paragraph. We modified this chapter so that it better states the same analyzers were used for the flux N2O intercomparison in Rannik et al. (2015). We also give more information of the data collection periods for the two analyzers used in this manuscript, and give reasoning why data from only one of them is used in correlation analysis of this paper. Please, see also our response to the comment concerning the results section at P. 6, L. 22. In response to this comment, we show the intercomparison of the two analyzers with respect to FCO, and we give this information shortly in the corrected manuscript.

P. 6, L. 4-7: Here it would be interesting to know, what the magnitude of the CO flux loss was, regarding the given response times of the EC systems. In context of the effect of the inlet lines, it would also be beneficial to mention their inner diameters in this section. According to Rannik et al. (2015) the reason for the larger response time of the system was caused by laminar flow due to a larger tubing diameter. For AR-CW-QCL the 5 and 95 percentile values of flux underestimation were 2.1 and 12.2% and for LGR-CW-QCL 5.7 and 21.4%, respectively. We added the information of the inlet lines (inner diameter and lag time from tube flow).

P. 6, L. 8-10: As stated here, more data had to be removed during daytime than during night-time. However, especially at night-time flux data has to be often rejected due to insufficiently developed turbulence. For this, a flux quality criterion using e.g. integral turbulence characteristics as suggested by Foken and Wichura (1996) is often applied. Also a test on stationarity, which was not applied for the N2O fluxes in Rannik et al. (2015) for intercomparison reasons, might be important for CO. We did not perform flux stationarity test. First, a range of tests was applied according to Vickers and Mahrt (1997), which ensure data screening for system malfunctioning as well as physical but unusual behavior, including the non-stationary conditions. Therefore we did not perform an additional test for stationarity according to Foken and Wichura (1996), and we

relied on the tests performed. It is the choice of the researcher to choose the test, however, statistically different tests tend to identify the same occasions of measurements, whereas the result depends also on the threshold criteria applied. E.g. Rannik et al. (2003) analysed performance of different tests and concluded that flux tests based on relative errors such as the stationarity test by Foken and Wichura (1996) are not feasible when the fluxes are small and therefore the relative errors becomes large. Therefore, we chose to perform tests on single time series to ensure quality of measurements used in the analysis and not using the flux stationarity test because the CO fluxes are frequently small and respectively with large relative random errors. Rannik, Ü., Aalto, P., Keronen, P., Vesala, T. and Kulmala, M., 2003. Interpretation of aerosol particle fluxes over a pine forest: Dry deposition and random errors. J. Geophys Res., 108 (D17), pp. AAC 3-1—3-11. DOI: 10.1029/2003JD003542.

P. 6, L. 22: The results chapter presents the measured CO flux and its correlation with various environmental parameters. In addition, I find it important to also present the CO mixing ratios as they can influence the CO flux signficantly. Especially, the amount of CO uptake might be largely dependent on the available atmospheric CO. To rule out the effect of changing atmospheric CO levels on the CO flux when interpreting the results, CO mixing ratios should then also be included in the correlation analysis. We had the atmospheric CO mixing ratio data [CO] in the original correlation analysis, however, as the correlations between daytime [CO] and FCO were very poor (r<0.2) and mostly not significant, we did not include [CO] in the table 2. Now we have added [CO] in the Table 2. To assess whether diurnal FCO depends on the diurnal trend in [CO], we performed additional correlation analysis using the half-hourly mean values of FCO and [CO] for each of the six measurement periods (Spring, Early Summer, Mid Summer, Late Summer, Autumn, Late Autumn). This analysis showed strong significant negative correlation between FCO and [CO] during all other periods except in the spring, also seen in Figure 1 (below). This analysis indicates that CO uptake increases with increasing [CO], and that CO uptake may be limited by [CO] at this site.

It is noteworthy, however, that the variation in [CO] are very small and hence we doubt that this is the sole factor controlling CO uptake. At the same time, the same correlation analysis using the mean diurnal FCO and other measured variables (radiation, sensible heat flux, latent heat flux, soil heat flux, NEE, RESP, soil temperature) produce highly significant and strong correlations between FCO and the measured scalars. This shows that the diurnal FCO is strongly driven by these variables, and that the correlation between FCO and [CO] may result from the dependencies of FCO and its driving variables, not solely on substrate limitation of CO consumption.

At this point, we added the results from this correlation analysis to the text, and we included new discussion on the effect of variation in [CO] on FCO. We also consider adding a new table presenting the correlation analysis between the mean diurnal FCO and all other measured variables.

see pdf-file: Figure 1. Dependency of FCO on the mixing ratio of CO [CO] during mean diurnal cycle from the six measurement periods from April to November 2011 at the reed canary grass crop. Values marked in red denote for daytime and values in blue denote for night-time data.

P. 6, L. 22: As it was mentioned in the method section, two different instruments for the CO flux measurements were used. However, in the result section the data from both analyzers is only shown as the cumulative flux in Figure 1f. If two independent analyzer are used, I would expect a paragraph or statement on the comparability of both measurements. This would give a better insight into the associated flux errors and would be also be valuable information for the CO flux community. Looking at the cumulative flux estimates, there seems to be a good agreement between days 205-270, while after that both fluxes seem to differ. Also, it should be stated in the manuscript that the presented fluxes (despite the green cumulative curve) are from the AR-CWQCL instrument while the LGR-CWQCL instrument was only operated from day 205. We agree that it would benefit the scientific community to show the intercomparison data of these two gas analyzers. For the period when both AR-CWQCL and LGR-CWQCL

were measuring FCO, we made plots showing the FCO measured by LGR-CWQCL against the FCO measured by AR-CWQCL (Figure 2). Also, we plotted the time series of half-hourly mean FCO and the daily mean FCO from both analyzers (Figure 3). This comparison shows considerable agreement between the analyzers with a slope of 0.96 and correlation coefficient of 0.95. The comparison shows that LGR-CWQCL shows slightly (4%) smaller fluxes compared to AR-CWQCL. The difference between the analyzers, however, is very small, giving us confidence in the use of either of the analyzer in further analysis. We have added a chapter in the results section describing the intercomparison of the two analyzers, however, we think it is unnecessary to show a figure from this comparison.

See pdf-file: Figure 2. Comparison of FCO measured by LGR-CWQCL (LGR) against the FCO measured by AR-CWQCL (AR-QCL) over the period days 206-330 at the reed canary grass crop.

See pdf-file: Figure 3. Half-hourly mean FCO and daily mean FCO measured by AR-CWQCL (ARI-QCL) and LGR-CWQCL (LGR-QCL) during the period of days 206-330.

P. 9, L. 11-13: To correct for this bias, a gap-filling method can be applied for the calculation of cumulative CO fluxes. We used a simple statistical gap-filling method to test how this would affect the cumulative FCO over the whole measurement season (Figure 4). The gap-filling was performed by choosing randomly the unique missing values from within time-window +- 5 days, by differentiating days and nights (according to elevation of sun). This simple gap-filling was performed for days excluding those which had no single measurements available. Hence, the gap-filling method removes possible bias due to different fraction of missing during day- and night-time. However, it does not guarantee correct cumulative sum because days with no data were not gap-filled including the measurement break. We hesitated to gap-fill the periods when no data was available due to the relatively poor correlations between the measured variables and FCO, especially during summer period (days 181-205). The gap-filling exercise in Figure 4 shows that the emission period in the spring and in late summer

is strengthened due to the even contribution of daytime and night-time data, which in this case includes a higher number of positive FCO. Similarly, the gap-filling leads to strengthened CO uptake in the autumn indicating that a higher number of night-time data was missing from that period. Overall, the cumulative curve of the original data and the gap-filled FCO result in very similar CO uptake rate after the 7-months of measurements. At this point, we hesitate to include the gap-filled data in the manuscript as it does not change the interpretation of the results. Still, we are happy to include the data if the reviewers/Editor see this as an informative and important part of the manuscript.

Figure 4. Cumulative FCO calculated from the measured data (bold lines) and gap-filled data (thin lines).

P. 7, L. 19-21: As stated, the concept of the gross FCO only holds if the CO uptake can be assumed to be constant over the entire diurnal cycle. However, especially turbulent transport and transport through the quasi-laminar boundary layer at the surface typi-cally show distinct diurnal cycles with maxima during daytime. Hence, I would expect the CO uptake to increase during the day, unless the CO uptake is limited mainly by soil microbial consumption or transport in soil (then, the CO flux would also mainly be independent from above surface CO-concentrations, which would change during day). Is there more evidence that can support the assumption of a constant CO uptake? The authors note that there is evidence from previous studies that the temperature effect on microbial consumption can be assumed to be small. In my opinion it should also be shown that the CO uptake is mainly limited by soil microbial consumption or transport in soil for the assumption of a constant CO uptake to be valid. Otherwise, the diurnal variation in the aerodynamic and the quasi-laminar boundary layer resistances would have to be taken into account. In general, the use of a bi-directional exchange model would be useful to address the issue of flux partitioning and importance of soil up-take, although I understand that this is challenging given the lack of detailed process studies on CO exchange and might be the scope of future research.

We agree that the use of the assumption of constant CO uptake may have been wrong. This was pointed out also by the referee #1, who suggested to use reported temperature dependencies of CO uptake from e.g. Whalen and Reeburgh (2001). As suggested, we used a Q10 value of 1.8 (Whalen and Reeburgh, 2001) to calculate the daytime CO uptake from the night-time CO fluxes over the six distinct measurement period. This allowed us to recalculate the gross CO emissions during daytime. Assuming this temperature dependency, the CO uptake was up to 2 times higher during day than during night. As the net daytime FCO remained positive during the spring, early summer and late summer, we expect that also CO emissions must have increased during the day. In a new table (Table 2), we report the daytime and night-time soil temperatures, the temperature difference between day and night, which is used for calculating the temperature dependent CO uptake during daytime, and the consequent gross CO emissions.

Based on the correlation analysis between FCO and [CO] we also found that the CO uptake seem to increase with increasing [CO] (see comment above). This indicates that the microbial CO consumption may be substrate limited during daytime, when the [CO] is slightly lower than in the night. This furthermore, may decrease the CO uptake during daytime, possibly and partly eliminating the increase in CO consumption due to increased daytime temperatures. It is not possible to differentiate between the microbial CO consumption and the physical substrate limitation, however, we acknowledge these mechanisms, and we added discussion of them to the text.

P. 8, L. 8-16: What was the applied definition for daytime and night-time periods? This is valuable information, as the correlation values are often largely dependent on the variation of the used parameters, which are typically larger during daytime. In this context, it might be also valuable to mention if the flux error had an impact on the weak correlations found during night-time. Since random uncertainty of flux estimates is inherent property of the eddy covariance method, the correlations can be affected by these errors. Day- and night-time fluxes differed significantly in magnitude only during

the first sub-period of the campaign, doy 110-145, see Fig. 2, therefore we can expect that night-time correlation values were affected by the random flux errors more than the day-time values only during the first period. We added the definition of daytime and night-time periods in the text by stating that we used sun elevation angle (h<0 for night-time, h>0 for daytime) to separate between daytime and night-time data.

P. 9, L. 11-13: To correct for this bias, a gap-filling method can be applied for the calculation of cumulative CO fluxes. As explained above, we tested a use of gap-filling for missing data to estimate the effect of uneven data removal during daytime and night-time. This gap-filling indicates that the real FCO are more positive during the spring and summer compared to the actual quality screened data, which removes more data during daytime than during night-time. The cumulative gap-filled FCO curve (above) shows that both the emission period in the spring and the uptake period in the late summer and autumn may be more pronounced than that of the data without gap-filling. The resulting net cumulative FCO over the whole measurement period, however, seems to be very similar with or without gap-filling (see above).

P. 9, L. 14-15: As FCO describes the net CO flux, one should differentiate here more explicitly between the emission component and uptake component of the flux. Otherwise the reader may assume you are referring to the net emission/uptake.

3 Technical comments

P. 3, L. 4: Write "reed canary grass" instead of "read canary grass". Correct also on P. 13, L. 9 and 19, L. 1. Corrected.

P. 3, L. 13: Omit space after "27âŮ̧ę" or introduce after all units (âŮ̧ę, ', "). Use same degree sign as used in L. 15. Corrected.

P. 3, L. 17: Use superscript for "-1" in "ha-1". Corrected.

P. 4, L. 10: Shouldn't it be "L=+-100 m" for the definition of the near-neutral range? We used L = -100 m as the simulation case for neutral stratification. Since the absolute value of this L is much larger than the measurement height, the neutral stability assumption for this case is well justified.

P. 4, L. 16: Insert space before "Considering". Corrected.

P. 4, L. 26: Write "LGR-CWQCL" instead of "LGRCW-QCL" as in the rest of the manuscript. Corrected. And in fact, we chose to use the abbreviation LGR-CW-QCL as in Rannik et al. (2015).

P. 6, L. 1: Do you intentionally differentiate between "co-variances" (here and L. 5) and "covariance"? We did not intend to use "co-variances" but rather "covariance". This is now corrected.

P. 6, L. 10: Write "daytime" instead of "day-time" as in the rest of the manuscript. Correct also on P. 7, L. 15 and on P. 9, L. 24. Corrected.

P. 6, L. 27: I suggest using "over the 9-month measurement period" instead of "in the end of the 9-month measurement period" as the used expression could be misleading otherwise. Corrected.

P. 7, L. 17: Use superscript in units. Corrected.

Figures 2-5: Instead of using the day of year numbers, I suggest to use the introduced classification of S, ES, MS . . . in the subplot titles (or use both, DOY + the classifi-cation). This makes it easier to compare with Figure 1 and descriptions in the text. We modified the figures 2-5, and a new figure 6 to include the classification of S, ES, MS. . . + DOY (e.g. S, 110-145), similar to that presented in Table 1.

4 References

Foken, T. and Wichura, B.: Tools for quality assessment of surface-based flux measurements, Agric. For. Meteorol., 78(1-2), 83–105, doi:10.1016/0168-1923(95)02248-1, 1996. Rannik, Haapanala, S., Shurpali, N. J., Mammarella, I., Lind, S., Hyvönen, N., Peltola, O., Zahniser, M., Martikainen, P. J. and Vesala, T.: Intercomparison of

fast response commercial gas analysers for nitrous oxide flux measurements under field conditions, Biogeosciences, 12(2), 415–432, doi:10.5194/bg-12-415-2015, 2015.

Please also note the supplement to this comment:
http://www.biogeosciences-discuss.net/bg-2015-622/bg-2015-622-AC2-supplement.pdf

———————————————

---

## Author Response (AR1)

**Response to the reviewer comments**

text written in blue = corrections made/planned at the time of submitting the response letter
text written in red = corrections made to the manuscript together with the submission to BG

Referee #1:
General comments:
The paper is well written and is the first to show the use of the eddy covariance technique for CO flux measurements. It is interesting to see a CO flux dataset with a high temporal resolution over 9 months, which is novel. Quite some studies have shown a diurnal CO cycle before but, as they say, this study is the first one to study the change in diurnal cycle over several months. The paper gives a good overview of how CO fluxes can be predicted or modeled by showing correlation matrices for many different variables, and by showing how the correlation matrices are changing over the season. This is for example very useful information for climate and carbon transport models and therefore a valuable dataset.

However, the interpretation of the dataset is at some points weak. With the Eddy Co-variance measurements, they try to answer process level based questions, which their dataset is not fully suitable for. For example, the measured EC CO fluxes are a net result (sum) of several processes (uptake by soil, production by soil, production by dead organic matter and production by livings plants) and each of these uptake and emission processes have their own dependencies on environmental variables. While not being able to separate these sources (and their mechanisms and dependencies), still process level based questions are tried to be answered by use of best fitting correlation matrices, and by use of several assumptions (for example the assumption of stable soil CO uptake). When the paper wants to focus on process level based questions, this approach and its considerations and restrictions should be discussed in more detail. Also some other parts of the dataset interpretation and dataset explanation need some more work. Furthermore, by discussing the limitations of the interpretation part, they can determine interesting (process level) research topics/setups for future CO flux studies, thereby contributing ideas for future CO flux studies.

In general, I would consider this a nice dataset which should be published. However, as said, the interpretation of this dataset is at some points weak and needs to be worked on. The points which should be revised or rewritten are more elaborately described in the 'specific comments' section.

We want to thank the reviewer for these constructive and important comments. We have now addressed the general concern of how to interpret the results with the current data set. We have added more discussion about challenges in separating the different processes of CO exchange, and restrictions of this study setup in addressing process level issues. We have also estimated the effect of temperature dependency of the CO uptake, as requested, and we have carefully edited the text so that we do not over interpret the results related to CO forming processes. Furthermore, we moved part of the process level discussion to the end of the paper to suggest ideas for future research topics on CO exchange. In addition to the specific comments below, we have extensively modified the discussion, and part of the introduction to better introduce the processes involved in CO exchange, discuss the interpretation of the results, and also to compare the results from this study to other published work. These changes are marked in red, as everywhere in the manuscript.

Specific comments:

The terms photodegradation, thermal degradation and abiotic degradation are not always used with care. With the EC method it is hard to separate different uptake and emission processes. Based on correlation coefficients, they conclude that radiation is the main driving factor of CO emission. This conclusion cannot be made based on their data. Radiation has many indirect effects such as on temperature, biological activity, etc. From their data it is hard to conclude whether direct photodegradation, or indirect effects of radiation (such as indirect photodegradation (fragmentation of organic matter) or thermal degradation) are the main cause of the CO production. In some places in the paper, this is well acknowledged (page 11, line 4-8). In other places this is neglected and the statistical correlation to radiation is given as a proof for direct photodegradation being the main cause. The difference between direct and indirect photodegradation should be explained, and conclusions on this subject should be formulated more carefully.

Thank you for taking this topic up. Indeed, we agree that we have not been consistent with the use of the terms photodegradation, thermal degradation and abiotic degradation. We have now used more space to explain these different mechanisms, including direct and indirect photodegradation (see e.g. Page 2, lines 20-29, Page 3, lines 1-3). We also aim at not overstating process level drivers of CO fluxes as our data does not allow us to conclude this (see e.g. Page 13, lines 21-28, Page 14, lines 10-13). We have now carefully gone through the text and modified it so that we do not make too strong statements or process level conclusions based on our results.

They make an important (risky) assumption by saying that biological soil CO uptake is constant, based on the paper of Conrad & Seiler (1985). However, other CO flux studies have observed the typical biological temperature response wherein biological activity increases with temperature (for example: Ingersoll 1973, Whalen & Reeburgh (2001), others). Also, especially in cold ecosystems, a small temperature change usually influences biological rates significantly. The assumed stable soil CO uptake assumption in this ecosystem seems unlikely. With the current dataset, this assumption cannot be validated or falsified. So, the authors should reconsider this assumption and think of the consequences if there is a typical temperature response, for example as found by Whalen & Reeburgh (2001), with a Q10 of 2.0. How would this influence their main conclusions? Either this possibility should be discussed, or the 'stable production' assumption should be removed from their manuscript.

We agree that the use of an assumption of a stable soil CO uptake was too simplistic, and as reported by e.g. Ingersoll (1973) and Whalen and Reeburgh (2001) not correct. Assuming a temperature response of Q10 of 1.8 for the CO uptake reported by Whalen & Reeburgh (2001), we estimated the daytime CO uptake from the night-time net CO fluxes and soil temperatures. We used soil temperature at 2.5 cm depth in the calculation as we considered that this is closest to the location where microbial CO consumption takes place. Hence, we assumed that the night-time CO fluxes (near constant negative fluxes) result from microbial CO consumption, which has a temperature response. The resulting daytime CO uptake estimated for each measurement period (Summer, Early Summer, Mid-Summer, Late Summer, Autumn, Late Autumn) allowed us to estimate the gross CO production during daytime, which is the difference between net daytime net CO flux and calculated daytime CO uptake. These results are now reported in two tables: Table 1. The mean, median and 25-75th percentiles of the net CO fluxes, net daytime CO fluxes and net night-time CO fluxes for each measurement period. Table 2. The gross daytime CO emission (gross FCO), difference between daytime and night-time soil temperatures, daytime CO uptake (using Q10 of 1.8), and gross daytime CO emission (gross FCO (Q10, 1.8)).

Based on these calculations, we find that the daytime CO uptake is almost twice as high as the night-time CO uptake. This is further reflected in significantly higher gross daytime CO emissions. In Page 13, lines 5-13) we have added discussion on the diurnal variation in CO uptake as well as the effect of this to the gross CO emissions, which is now largely overcome by the high CO uptake.

Concerning possible biological CO production mechanisms, they hypothesize that CO emissions are not driven by microbial activity. While it is likely that the observed CO emissions are not driven by microbial activity, the used argumentation might be misleading: they base it on the poor correlation between FCO and FN2O, and the poor correlation between FCO and RESP. However, FCO is a net flux (sum) of uptake and production, while RESP and FN2O are solely production fluxes. This makes the validity of the correlation questionable. Also, in case the CO production is caused by biological as well as by abiotic sources, would this not result in the same poor correlation between FCO and FN2O? With the current dataset, it is difficult to determine whether biological fluxes are present, but saying that the poor correlation indicates the absence of biological sources might be misleading. In previous studies, what are the magnitudes of the reported biological fluxes? In this ecosystem, would they have the same magnitude? Are they also expected in autumn when vegetation is less active/dormant/etc? If the authors believe that biological fluxes play a (small) role, it would be good to spend some sentences on the assumed mechanisms and maybe indicate the magnitude of the observed biological fluxes in other studies as a comparison.

We agree with the reviewer that the used argumentation in explaining the CO production mechanisms was not sufficient. We also agree that the poor correlation between daytime FCO and FN2O, or FCO and RESP does not prove that the CO emissions are not driven by microbial activity. We have modified this chapter to include discussion on the reasons why we do not expect the CO formation to be of microbial origin (Conrad and Seiler, 1980), and to give a better understanding of the abiotic and biotic CO production mechanisms. We also discuss the connections between FCO and FN2O as we found significant negative correlation between night-time FCO and FN2O, indicating involvement of N2O forming microbes in the CO uptake (see later our answer focusing on FN2O, and manuscript at page 15, lines 8-18). We also added more discussion on the current understanding of biological CO production, however, as there is very little information available on this topic, we stated also that biological CO formation and its importance is currently still poorly understood.

In the discussion (page 11, line 21-25), they use the high C to N ratio to confirm their theory that photodegradation is the main cause. However, this argument is not explained. Why does a high C to N ratio confirm the hypothesis?
After going through the text in the manuscript and reconsidering the reviewer suggestions, we do understand how naïve it was to suggest photodegradation as the main process leading to CO emissions. We understand that abiotic CO formation is a result of both photodegradation and thermal degradation, which cannot be fully separated in field experiments due to e.g. indirect effects radiation. Currently, there are studies supporting for higher contribution of photodegradation to the CO formation (e.g. Lee et al., 2012), but also studies suggesting thermal degradation as the dominant CO forming process (e.g. van Asperen et al., 2015). Related to the C to N ratio of the plant material, a meta-analysis shows that CO formation via photodegradation increases with C to N ratio of the plant material (King et al., 2012). Also, as the plant material in our measurement site has a high C to N ratio, and as this dry plant material was well exposed to radiation in the spring, we expect that the conditions were suitable for CO formation via photodegradation. However, this does not confirm that photodegradation was the dominant process at our site, nor does it exclude thermal degradation to take place.

We have now carefully gone through the text, including this paragraph, to discuss more generally the combined effect of photodegradation and thermal degradation. Instead of searching for one specific process, we consider that it is more important to discuss the combined effect of abiotic processes (photodegradation and thermal degradation).

The comparison to other variables (NEE; heat and energy flux) is stated as a goal in the last paragraph of the introduction, but it is not well explained what is expected. Also, in general the comparison is held very small, especially for the N2O fluxes. The results are only shown in a table but not discussed, and the N2O fluxes are hardly mentioned in the Discussion and forgotten in the Conclusion. For the N2O results, the reader is referred to another paper. If the authors think that the N2O story is an important part, since they state their interest in the introduction, they should show some N2O results, interpret these results, and spend some text on why they expect a correlation. Does a N2O figure maybe fit in Figure 1? Or, if they prefer to refer to the other paper, please then describe the N2O fluxes (magnitude and diurnal variation) briefly in this paper.

We appreciate this concern. We have added text to clarify why we would expect correlations between FCO and other measured variables. With some of them (heat flux, radiation, energy flux) it is clear based on process understanding and previous studies stating that CO emissions are driven by radiation and temperature, however, with some of the measured scalars (e.g. NEE, RESP) we did not know what to expect as there is very limited information available on the links between them and FCO. Based on understanding of biological CO formation, a negative correlation between FCO and NEE would indicate involvement of a biological component in the CO production. Indeed, the FCO and NEE correlated negatively (r=-0.469) during early summer (days 146-160), which gives support to the CO formation from living and actively photosynthesizing plants. With respect to FN2O and FCO, we do not expect a strong relationship due to the difficulties in separating between overlapping abiotic CO production, microbial CO consumption (Conrad and Seiler, 1980; Moxley and Smith 1998), and microbial N2O production/uptake in the soil. Nitrifiers are among a diverse microbial community oxidizing CO in soils (Jones and Morita, 1983; King and Weber, 2007). Hence a high nitrification activity may be reflected in higher CO consumption in the soil. In the field, this could be visible during night-time when the CO consumption is expected to dominate the net CO fluxes, while in most of the year during daytime the CO production overrides the consumption. If a large fraction of the CO uptake was due to nitrification activity, we should be able to see this in negative correlation between night-time FN2O and FCO. In fact, we found significant negative correlations between FN2O and FCO during night-time in the spring (r=-0.336), mid-summer (r=-0.607) and late autumn (r=-0.514). These correlations were significant but much weaker during the daytime (Table 3). These findings hint towards a marked role of nitrifiers in CO consumption at the reed canary grass site, however, we cannot confirm this as no microbial community structure analysis was conducted.

Jones, R.D., Morita, R.Y.: Carbon monoxide oxidation by chemolithotrophic ammonium oxidisers. Can. J. Microbiol. 29, 1545-1551, 1983.

King, G. M., and Weber, C. F.: Distribution, diversity and ecology of aerobic CO-oxidizing bacteria. Nature Reviews, Microbiology, 5, 107-118, 2007.

Moxley, J.M., and Smith, K.A.: Carbon monoxide production and emission by some Scottish soils. Tellus, 50B, 151-162, 1998.

Shurpali, N. J. et al. Neglecting diurnal variations leads to uncertainties in terrestrial nitrous oxide emissions. Sci. Rep. 6, 25739; doi: 10.1038/srep25739 (2016).

Looking at figure 2 and 3, they correctly conclude that not all CO fluxes can be initiated by radiation, since CO fluxes are already increasing when the sun is still down (in autumn), and they refer to the possibility of thermal degradation. With the assumption of stable soil CO uptake during the dark hours, and with the idea of thermal degradation being responsible for the increasing CO production during the morning hours in the dark, is it possible to (roughly) estimate how much thermal degradation is contributing to total CO fluxes? Can this be extrapolated to the day? And does this estimate change when there is no stable soil CO uptake assumed?

In order to estimate how much a temperature increase in the morning hours would contribute to the CO production via thermal degradation, we used a Q10-value of 2.1 (van Asperen et al., 2015) to estimate. At first we calculated the temperature differences at 2.5 cm depth in the soil between mid-night and morning hours just before the sunrise, when sun elevation angle became positive, during the six measurement periods. We found that the soil temperature decreased (0.1 to 1 ºC) from mid-night to the morning hours. Similar trend was observed also in air temperature. This phenomena was consistent throughout the whole 7-month measurement campaign despite the fact that the time of sunrise changed markedly between the seasons (very short nights during the summer). The attached figure 1 illustrates the diurnal cycle in mean soil temperature for each of the six measurement periods together with sun elevation angle data. The graphs shows how the soil temperature still continues to decrease even after the sunset, when sun elev is above zero.

[Figure]

Figure 1. Mean sun elevation angle (h<0 night-time, h>0 daytime) and mean soil temperature at 2.5 cm depth at the reed canary grass crop over the six distinct measurement periods (S = Spring, ES = Early Summer, MS = Mid-Summer, LS = Late Summer, A = Autumn, LA = Late Autumn).

As a result, we do not expect temperature driven CO production via thermal degradation to take place in the early morning hours. As seen in the Fig. 2 and 3. and as stated by the referee, the CO fluxes are already increasing when the sun is still down. Hence, the increase in net CO fluxes during the morning hours indicates that CO production from an unspecified process increases, or CO uptake decreases during the morning hours, or that both of these take place. As explained above, we do not expect thermal degradation to be responsible for increased CO production, however, we can speculate that the CO uptake was affected by the decreasing temperature, as it was earlier estimated the CO uptake is temperature driven (see above, Q10 of 1.8). To conclude, based on this analysis, we cannot estimate the role of thermal degradation to the CO production at this site. In Page 14, lines 1-16, we have added discussion concerning this, and the challenges to separate thermal degradation from the CO production. We also added discussion on the possible effect of night-time temperature variation on CO uptake, partly explaining the increasing net CO flux during the early morning hours.

The sampling line material is made from PTFE, which is reported to be inert. However, was the whole sampling set up made of PTFE (from inlet to instrument)? Other materials are known to be possible strong CO emitters, and previous CO flux studies have found CO producing material in setups during blank tests (Schade 1999, van Asperen 2015). Has the setup been

tested for internal CO production and have blank tests been performed? If so, please mention this. If there is internal CO emission, it probably wont influence your results largely due to the large sampling flow, but if possible, it would be the best to quantify this.

All the sampling lines were made of PTFE, and due to the high flow rates in the sampling line, we did not expect internal CO production in the tubing, inlet or the whole measurement setup. We did not conduct material tests for the measurement setup we present in this paper, however, we have conducted extensive material tests with one of the analyzer (LGR-CW-QCL) and we have found that most of the common tube materials made of Teflon (FEP, PFA), or Nylon and Polyurethane release CO, and that this rate of release depends on temperature and radiation (unpublished data). We consider this is critically important in systems when a sample is accumulated within a system and when there is no constant flow through the system, such as static soil chambers. We added a sentence in the Materials & Methods (Page 6, lines 2-3) stating that PTFE tubing was found inert with respect to CO under constant-flow setup with the LGR-CW-QCL analyzer (unpublished data).

On Page 6, line 26-27, you estimate that the site is a net sink of CO for the 9 months, which is nicely shown in Table 1. Concerning that you seem to have a good idea of which environmental variables are important per time of the season, and that you are the first one to show a dataset with such high temporal resolution for 9 months, is it possible to give an estimate of the net CO fluxes for the other 3 months, so you can give an annual estimate? Such as done in Table 6 or 7, in the paper of Ingersoll (Soil's potential as a sink for atmospheric carbon monoxide, 1973). That would be an interesting addition.

We also think a full annual balance of FCO would be very interesting. Our measurements cover the snow-free period of little more than 7 months (not full 9 months as stated earlier). Based on our measurements, the FCO was rather constant during the autumn and late autumn, but very variable during the spring right after the snow melt when the measurements started. It is very probable that the FCO are minimal during the snow-cover period in December-February, as temperatures and radiation are low and we can expect rather small microbial CO consumption activity in the soil. However, for the spring period during the snow-melt in March-April, the assumption of small FCO does not necessarily hold as the amount of radiation and temperature increase and the soil surface is freed from the snow allowing the old previous year's crop residues to decompose. Hence, we expect that the use of the mean FCO from the measurement period probably underestimates the FCO during the early spring period. Nevertheless, we performed a back-of-envelope calculation assuming a mean FCO over the whole measurement campaign of -0.25 nmol m2- s-1 to apply for the missing period of day 326 – day 109 (22 November 2011 - 18 April 2012). This results in an annual net cumulative FCO of -111 mg CO m-2 yr-1. Based on only 7 months of measurements at one measurement site, and due to the lack of a process-based model for the CO exchange, we decided not to present the simple extrapolation of the annual FCO to grasslands in Finland. In Pages 12-13, lines 26- we have added our annual FCO estimates, and discussed their uncertainties.

Having done this exercise, it is easy to say that more high-resolution measurements are needed to cover the whole seasonal cycle in CO exchange, and to obtain a reliable estimate for annual CO balance from boreal ecosystems.

Technical corrections:
General:

- Please check your references, for example, Lee (2012) and Zahniser (2009) are missing in the reference list, but maybe there are more. - For many units in different places in the manuscript (ha-1, m-2), the 'superscript' is not used.
- The hyphen is not used consistently throughout the manuscript - Different places in the manuscript: Please use the same term for G (ground heat flux or soil heat flux)
Thank you for spotting these. We will go through them carefully.

Text:
- Page 2, line 2: of a strong greenhouse gas → of the strong greenhouse gas.
Corrected.

- Page 2, line 12-17: quite some references are named in the different places in the manuscript, which are not named in this part. It might be nice if the references which are coming back in table 3, also come back here in the right place.
We agree. We will check that all relevant references are added here.

- Page 2, line18: Here is stated that CO emissions are thermal or UV-induced. However, it is not only by UV, also by visible radiation, as shown by Lee (2012). Lee (2012) is mentioned on page 9, line 21, but does not appear in reference list.
Thank you for spotting this. We have added Lee (2012) to the reference list, and changed the sentence as follows: "…however, most often the CO production has been related to abiotic processes such as thermal or UV- or visible light-induced degradation of organic matter or plant material (references)" (Page 2, lines 17-19).

- Page 2, line 22: Most of the reported CO flux measurements are either short-term field experiments from.... it seems that the author wants to make a point here that no CO measurements are made so far in this cropland boreal ecosystem, or are only made short-term. But neither of the point is clearly made. Is this the first measurement in this ecosystem? Or the first long term?
We wanted to in the first place point out that this is the first long-term study reporting FCO from any ecosystem. We clarified this in the text.

And can there be an indication for which percentage of land use/Finland/boreal zone this ecosystem is representative for? (see paper Ingersoll, 1973, table 6,7)
Despite of what we had written in the response letter of the BGD paper (that we would do the extrapolation according to Ingersoll (1974), we hesitated to continue in this direction. This is justified by the fact that we eventually had only 7 months of measurements instead of the 9 months reported in the BGD paper. Also, we consider that making such extrapolations from one measurement site in Finland, and from measurement which do not cover a full year, is too uncertain. Especially as we do not yet have a ready process-based model to gap-fill the missing data and make a reliable annual estimate of the site.

- Page 3, line 20: The footprint length is given, and the size of the field is given. I assume the author wants to implicate that the footprint is homogeneous in all directions, but this is not stated. Clarify for the reader.
Page 4, line 15: We clarified that the footprint is homogenous in all directions.

- Page 3, line 20: A 6.3 ha field is introduced, is this the same field as meant in the rest of §2.1, before this sentence? Not clear formulated.
Page 4, line 14: This sentence was rewritten to clarify that the study field was 6.3 ha in size.

- Page 4, line 16: No white space between '(see Fig.1c)' and 'Considering'.
Corrected.

- Page 4, line 15-20: it is stated that in the majority of the measurements is representative for the RCG canopy. What happened to the minority of the data when it is not? Can this be mentioned?
We assumed homogeneous canopy in all directions i.e. the estimated footprint extent is applicable to all directions. We estimated that the upwind distance contributing 80% of flux under stable conditions (L = +10 m), in case of low canopy, was 166 m. We use this as a very conservative estimate because low canopy existed only a short time period in the beginning of the campaign. For high canopy the respective distance was estimated to be 60 m. Therefore, considering minimum fetch in South direction 135 m, we concluded that fetch was sufficient under majority of observation conditions. No data rejection according to footprint estimation was done. We modified the sentence to be more clear.

- Page 4, line 25: Reference Zahniser is also missing.
Corrected.

- Page 4, line 24-26: Unclear and incorrect sentences, please rephrase.
Page 5, lines 16-21: This sentence was clarified, and more information was given of the two analyzers used, the time periods they were used, and justification why data from one of them only was used when analyzing the seasonal and diurnal variability in the FCO. Reviewer #2 was also asking for more information on the instrument comparison. See more details of that in the response to Reviewer #2.

- Page 5, line 7: PTFE lines were used, which are under most conditions inert. However, other parts of the used material might not be. Has there been a blank measurement? If so, this should be mentioned.
We did not perform blank tests during this measurement campaign as we assumed that the PTFE lines were not a significant source of CO, and because potential CO emissions from the materials can be largely avoided in the high-flow setup. We acknowledge that many materials, including FEP, PFA and Nylon may emit large quantities of CO, especially in static systems with minimal flow speeds (unpublished data in laboratory experiments using the LGR-CW-QCL analyzer (model N2O/CO-23d, Los Gatos Research Inc.). However, in a laboratory experiment testing PTFE tubing at flow rates of 2.5 L/min, we did not find significant CO production in the tubing system (see Page 6, lines 2-3).

- Page 5, line 11: The measurement position of G and Tsoil etc is not named in the 'Material and Methods', although partly named later in the manuscript.
This information in addition to the information of how other supporting material was collected is now added in a new chapter (2.3 Supporting measurements) to the materials and methods section.

- Page 5, line 21: a verb is missing. Do you mean: LGR-CWQCL measurements were corrected for.....
We mean that the LGR-CW-QCL analyzer itself corrects for the water vapor effect (an inbuilt correction algorithm). This was clarified.

– Page 5, line 22: unclear sentence. Do you mean: the same applied to the AR-CWQCL measurements after software update in July 2011.

Yes, the same spectroscopic correction was applied to the AR-CW-QCL measurements after the software update. We clarified this part in the text.

- Page 6, line 15: while the length of periods were → while the lengths of the periods were.
Corrected.

- Page 6, line 25: to the mid-June–> to mid-June.
Corrected.

- Page 7, line 15: near constant CO uptake, is the value you found similar to other studies?
We have added comparison of night-time CO fluxes measured in this study to those observed in other studies (Page 13, lines 21-26, see also Tables 4 and 5).

- Page 7, line 25-26: please mentioned '(days 110-145)' after 'during the spring'.
Corrected.

- Page 8, line 1: Here the discussion suddenly jumps from CO to CO2, maybe introduce this a little clearer.
Page 10, line 7: We added introducing sentence to the start of the chapter stating that compared to the FCO, the net CO2 exchange, expressed as net ecosystem exchange (NEE) was very small during the spring.

- Page 8, line 1: Have LAI and GAI been introduced before?
Indeed, these variables were not introduced before. As also many other supporting measurements were not described earlier, we introduced a new chapter (2.3) describing "supporting measurements" and the methods used in measuring them.

- Page 8, line 1-5: N2O is not discussed at all here.
Page 10, lines 12-16: We added a section describing the diurnal variation in N2O fluxes during the measurement period.

- Page 9, line 2-4: different type of ecosystems are compared here, but they lay in different climate zone. Does this comparison make sense then?
Page 12, lines 12-25: We reconsidered this comparison and modified the text to explain similarities between measurement from the same ecosystem types, and on the other hand, similarities between measurements in the same climatic zone. The variation in CO fluxes seems to be so high, and continuous or long-term measurements so rare that it is difficult to see trends between different ecosystem types measured in the same climatic zone, or trends between different climatic zones with respect to CO fluxes from the same ecosystem type.

- Page 11, line 1: suggestion: we expect that radiation–> we expect that the effects of radiation.
Corrected.

– Page 11, line 3: T soil at a depth of 2.5cm–> this should also be in materials and methods.
We added a description in the materials and methods (section 2.3)

- Page 11, line 24: mean and stdev are mentioned, however, the mean value is missing.
Maybe there was a printing error as in our version the mean was visible. Anyhow, this was checked to make sure that the numbers appear there.

- Page 11, line 26: the early summer emission–> the early summer CO emission.

Corrected.

Tables & Figures: In Table 3, a nice overview is given of previous CO studies. The fluxes which are reported here, are that daily averages? Several of these studies have also measured a daily cycle. Can the magnitude of these results be indicated? That might make the overview more complete.

Indeed, this information would be very valuable to the scientific community. However, we found that only very few studies report daytime and night-time CO fluxes, and that those reported fluxes may be based only on 24 hours' measurements (e.g. Zepp et al., 1997). For instance, Constant et al. (2008) measured CO fluxes by a flux gradient method over a grassland ecosystem for one year, but they do not specifically report night-time fluxes. With our knowledge, our study and the one by van Asperen et al. (2015) are the only studies available reporting daytime and night-time fluxes of CO. In page 13, lines 21-26, we have included these results in the discussion section, but not in the Table 3 (now renumbered as Table 4).

In Figure 4, NEE is mentioned. I assume this measured by the EC measurements of CO2? Maybe mention the trace gas in the caption, also in other places in the manuscript when mentioning NEE.

We added a description of the NEE (net ecosystem exchange of CO2) in the Figure 4 legend, and also in the text where NEE was mentioned (e.g. chapter 3.2 Diurnal variation).

**Response to the reviewer comments**

text written in blue = corrections made/planned at the time of submitting the response letter
text written in red = corrections made to the manuscript together with the submission to BG

Referee #2:
1 General comments

The manuscript of Pihlatie et al. on carbon monoxide (CO) flux measurements above an agricultural bioenergy crop (reed canary grass) represents an important study on the biosphere-atmosphere exchange of CO. While previous studies mainly focused on short term measurements of CO fluxes, the authors present the first eddy covariance measurements of CO fluxes over an entire growing season, making it a unique study. Like this, the authors can investigate the dependency of the CO flux on different environmental parameters such as irradiation, temperature, crop cover, fertilization status, etc. Interestingly, the authors find that the reed canary grass ecosystem acted as a net source of CO at the beginning and a net sink during of the rest of the growing season.

Also, they measured a strong diurnal cycle, as opposed to other previous studies over cropland, with mostly net emission during daytime and net uptake during night. In their study the authors correlate the net CO flux with environmental parameters to obtain an understanding on the controlling processes. As the nature of CO exchange is complex with many possible sinks and sources that have been observed in previous studies, this is challenging. As a consequence, the conclusions made on the underlying processes can often only remain assumptions, and therefore, the study provides only limited new insight into processes of CO exchange. As stated by the authors, further process related studies are necessary for future research.

The authors use state of the art measurement techniques for the quantification of CO fluxes and the fluxes were analyzed according to standard quality control procedures. Furthermore, the manuscript is clearly structured and well written. Due to the unique data set, I suggest the manuscript to be published in BG, after the more specific comments below have been addressed.

We want to sincerely thank the reviewer for constructive comments that help to improve the quality of the manuscript. We have carefully addressed all the comments and responded to them as follows. We acknowledge the concern of risks in process-based interpretations of the results. We hope the corrections will satisfy these concerns and underline the future research needs and gaps in knowledge in this field of research. In addition to the specific comments below, we have extensively modified the discussion, and part of the introduction to better introduce the processes involved in CO exchange, and also compare the results from this study to other published work. These changes are marked in red, as everywhere in the manuscript.

2 Specific comments

P. 3, L. 16-20: At which day was the crop cultivated? For completeness, I suggest to add this information to this short description of the growing season.
Page 3, line 20: The crop cultivation date was added to the Materials and Methods section.

P. 4, L. 21-28: In this paragraph it is not clear that these are the same analyzers as used for the flux N2O intercomparison in Rannik et al. (2015). It would be good to state this in this

manuscript or move the above sentence "The comparison of four laser-based. . ." to the end of the paragraph.

Page 4, line 28: We modified this chapter so that it better states the same analyzers were used for the flux N2O intercomparison in Rannik et al. (2015). We also give more information of the data collection periods for the two analyzers used in this manuscript, and give reasoning why data from only one of them is used in correlation analysis of this paper. Please, see also our response to the comment concerning the results section at P. 6, L. 22. In response to this comment, we show the intercomparison of the two analyzers with respect to FCO, and we give this information shortly in the corrected manuscript.

P. 6, L. 4-7: Here it would be interesting to know, what the magnitude of the CO flux loss was, regarding the given response times of the EC systems. In context of the effect of the inlet lines, it would also be beneficial to mention their inner diameters in this section. According to Rannik et al. (2015) the reason for the larger response time of the system was caused by laminar flow due to a larger tubing diameter.

Page 6, lines 23-24, and lines 26-28: For AR-CW-QCL the 5 and 95 percentile values of flux underestimation were 2.1 and 12.2% and for LGR-CW-QCL 5.7 and 21.4%, respectively. We added the information of the inlet lines (inner diameter and lag time from tube flow).

P. 6, L. 8-10: As stated here, more data had to be removed during daytime than during night-time. However, especially at night-time flux data has to be often rejected due to insufficiently developed turbulence. For this, a flux quality criterion using e.g. integral turbulence characteristics as suggested by Foken and Wichura (1996) is often applied. Also a test on stationarity, which was not applied for the N2O fluxes in Rannik et al. (2015) for intercomparison reasons, might be important for CO.

We did not perform flux stationarity test. First, a range of tests was applied according to Vickers and Mahrt (1997), which ensure data screening for system malfunctioning as well as physical but unusual behavior, including the non-stationary conditions. Therefore we did not perform an additional test for stationarity according to Foken and Wichura (1996), and we relied on the tests performed. It is the choice of the researcher to choose the test, however, statistically different tests tend to identify the same occasions of measurements, whereas the result depends also on the threshold criteria applied. E.g. Rannik et al. (2003) analysed performance of different tests and concluded that flux tests based on relative errors such as the stationarity test by Foken and Wichura (1996) are not feasible when the fluxes are small and therefore the relative errors becomes large. Therefore, we chose to perform tests on single time series to ensure quality of measurements used in the analysis and not using the flux stationarity test because the CO fluxes are frequently small and respectively with large relative random errors. This is discussed on Page 7, lines 2-5.

Rannik, Ü., Aalto, P., Keronen, P., Vesala, T. and Kulmala, M., 2003. Interpretation of aerosol particle fluxes over a pine forest: Dry deposition and random errors. J. Geophys Res., 108 (D17), pp. AAC 3-1—3-11. DOI: 10.1029/2003JD003542.

P. 6, L. 22: The results chapter presents the measured CO flux and its correlation with various environmental parameters. In addition, I find it important to also present the CO mixing ratios as they can influence the CO flux significantly. Especially, the amount of CO uptake might be largely dependent on the available atmospheric CO. To rule out the effect of changing atmospheric CO levels on the CO flux when interpreting the results, CO mixing ratios should then also be included in the correlation analysis.

We had the atmospheric CO mixing ratio data (MCO) in the original correlation analysis, however, as the correlations between daytime MCO and FCO were very poor (r<0.2), we did not include MCO in the table 2. Now we have added MCO in the revised version Table 3. In Page 12, lines 15: We also added a short discussion on the potential effect of MCO on CO uptake at our site, as suggested by the referee.

In the response letter published in BGD we discussed the correlation between CO flux and concentration obtained through averaging for certain time of day. Diurnal variation in environmental variables is natural due to variation in solar radiation and resulting boundary layer development processes. Also, the concentrations and fluxes have very different source areas. The fluxes represent the local source area whereas the concentrations with long atmospheric life time (such as CO) can be affected by very distant sources and modulated by diurnal cycle in atmospheric mixing. Therefore, correlation in diurnal variation between CO fluxes and concentrations does not necessarily stem from the causal relationship between these variables. Due to these reasons we have omitted the discussion from the revised manuscript.

P. 6, L. 22: As it was mentioned in the method section, two different instruments for the CO flux measurements were used. However, in the result section the data from both analyzers is only shown as the cumulative flux in Figure 1f. If two independent analyzer are used, I would expect a paragraph or statement on the comparability of both measurements. This would give a better insight into the associated flux errors and would be also be valuable information for the CO flux community. Looking at the cumulative flux estimates, there seems to be a good agreement between days 205-270, while after that both fluxes seem to differ. Also, it should be stated in the manuscript that the presented fluxes (despite the green cumulative curve) are from the AR-CWQCL instrument while the LGR-CWQCL instrument was only operated from day 205.

We agree that it would benefit the scientific community to show the intercomparison data of these two gas analyzers. For the period when both AR-CWQCL and LGR-CWQCL were measuring FCO, we made plots showing the FCO measured by LGR-CWQCL against the FCO measured by AR-CWQCL (Figure 1). Also, we plotted the time series of half-hourly mean FCO and the daily mean FCO from both analyzers (Figure 2). This comparison shows considerable agreement between the analyzers with a slope of 0.96 and correlation coefficient of 0.95. The comparison shows that LGR-CWQCL shows slightly (4%) smaller fluxes compared to AR-CWQCL. The difference between the analyzers, however, is very small, giving us confidence in the use of either of the analyzer in further analysis. In page 8, lines 18-23 we have added a chapter in the results section describing the intercomparison of the two analyzers, however, we think it is unnecessary to show a figure from this comparison.

[Figure]

Figure 1. Comparison of FCO measured by LGR-CWQCL (LGR) against the FCO measured by AR-CWQCL (AR-QCL) over the period days 206-330 at the reed canary grass crop.

[Figure]

Figure 2. Half-hourly mean FCO and daily mean FCO measured by AR-CWQCL (ARI-QCL) and LGR-CWQCL (LGR-QCL) during the period of days 206-330.

P. 9, L. 11-13: To correct for this bias, a gap-filling method can be applied for the calculation of cumulative CO fluxes.

We used a simple statistical gap-filling method to test how this would affect the cumulative FCO over the whole measurement season (Figure 4). The gap-filling was performed by choosing randomly the unique missing values from within time-window +- 5 days, by differentiating days and nights (according to elevation of sun). This simple gap-filling was performed for days excluding those which had no single measurements available. Hence, the gap-filling method removes possible bias due to different fraction of missing during day- and night-time. However, it does not guarantee correct cumulative sum because days with no data were not gap-filled including the measurement break. We hesitated to gap-fill the periods when no data was available due to the relatively poor correlations between the measured variables and FCO, especially during summer period (days 181-205). In page 11, lines 25-27 we mentioned that we tested this simple statistical gap-filling method, and that we decided not to present these results.

The gap-filling exercise in Figure 4 shows that the emission period in the spring and in late summer is strengthened due to the even contribution of daytime and night-time data, which in this case includes a higher number of positive FCO. Similarly, the gap-filling leads to strengthened CO uptake in the autumn indicating that a higher number of night-time data was missing from that period. Overall, the cumulative curve of the original data and the gap-filled FCO result in very similar CO uptake rate after the 7-months of measurements. At this point, we hesitate to include the gap-filled data in the manuscript as it does not change the interpretation of the results. Still, we are happy to include the data if the reviewers/Editor see this as an informative and important part of the manuscript.

[Figure]

Figure 4. Cumulative FCO calculated from the measured data (bold lines) and gap-filled data (thin lines).

P. 7, L. 19-21: As stated, the concept of the gross FCO only holds if the CO uptake can be assumed to be constant over the entire diurnal cycle. However, especially turbulent transport and transport through the quasi-laminar boundary layer at the surface typically show distinct

diurnal cycles with maxima during daytime. Hence, I would expect the CO uptake to increase during the day, unless the CO uptake is limited mainly by soil microbial consumption or transport in soil (then, the CO flux would also mainly be independent from above surface CO-concentrations, which would change during day). Is there more evidence that can support the assumption of a constant CO uptake? The authors note that there is evidence from previous studies that the temperature effect on microbial consumption can be assumed to be small. In my opinion it should also be shown that the CO uptake is mainly limited by soil microbial consumption or transport in soil for the assumption of a constant CO uptake to be valid. Otherwise, the diurnal variation in the aerodynamic and the quasi-laminar boundary layer resistances would have to be taken into account. In general, the use of a bi-directional exchange model would be useful to address the issue of flux partitioning and importance of soil uptake, although I understand that this is challenging given the lack of detailed process studies on CO exchange and might be the scope of future research.

We agree that the use of the assumption of constant CO uptake may have been wrong. This was pointed out also by the referee #1, who suggested to use reported temperature dependencies of CO uptake from e.g. Whalen and Reeburgh (2001). As suggested, we used a $Q_{10}$ value of 1.8 (Whalen and Reeburgh, 2001) to calculate the daytime CO uptake from the night-time CO fluxes over the six distinct measurement period. This allowed us to recalculate the gross CO emissions during daytime. Assuming this temperature dependency, the CO uptake was approximately 2 times higher during day than during night. As the net daytime FCO remained positive during the spring, early summer and late summer, we expect that also CO emissions must have increased during the day. In a new table (Table 2), we report the soil temperature difference between day and night, which is used for calculating the temperature dependent CO uptake during daytime, and the consequent gross CO emissions.

P. 8, L. 8-16: What was the applied definition for daytime and night-time periods? This is valuable information, as the correlation values are often largely dependent on the variation of the used parameters, which are typically larger during daytime. In this context, it might be also valuable to mention if the flux error had an impact on the weak correlations found during night-time.
Since random uncertainty of flux estimates is inherent property of the eddy covariance method, the correlations can be affected by these errors. Day- and night-time fluxes differed significantly in magnitude only during the first sub-period of the campaign, doy 110-145, see Fig. 2, therefore we can expect that night-time correlation values were affected by the random flux errors more than the day-time values only during the first period. In page 7, lines 14-15, we added the definition of daytime and night-time periods by stating that we used sun elevation angle (h<0 for night-time, h>0 for daytime) to separate between daytime and night-time data.

P. 9, L. 11-13: To correct for this bias, a gap-filling method can be applied for the calculation of cumulative CO fluxes.
As explained above, we tested a use of gap-filling for missing data to estimate the effect of uneven data removal during daytime and night-time. This gap-filling indicates that the real FCO are more positive during the spring and summer compared to the actual quality screened data, which removes more data during daytime than during night-time. The cumulative gap-filled FCO curve (above) shows that both the emission period in the spring and the uptake period in the late summer and autumn may be more pronounced than that of the data without gap-filling. The resulting net cumulative FCO over the whole measurement period, however, seems to be very similar with or without gap-filling (see above). Hence, we did not include gap-filled data in the manuscript. However, in Page 11, lines 25-27, we commented the use of gap-filling in order to justify not using it.

P. 9, L. 14-15: As FCO describes the net CO flux, one should differentiate here more explicitly between the emission component and uptake component of the flux. Otherwise the reader may assume you are referring to the net emission/uptake.
Corrected.

3 Technical comments

P. 3, L. 4: Write "reed canary grass" instead of "read canary grass". Correct also on P. 13, L. 9 and 19, L. 1.
Corrected.

P. 3, L. 13: Omit space after "27∘" or introduce after all units (∘, ', "). Use same degree sign as used in L. 15.
Corrected.

P. 3, L. 17: Use superscript for "-1" in "ha-1".
Corrected.

P. 4, L. 10: Shouldn't it be "L=+-100 m" for the definition of the near-neutral range?
We used L = -100 m as the simulation case for neutral stratification. Since the absolute value of this L is much larger than the measurement height, the neutral stability assumption for this case is well justified.

P. 4, L. 16: Insert space before "Considering".
Corrected.

P. 4, L. 26: Write "LGR-CWQCL" instead of "LGRCW-QCL" as in the rest of the manuscript.
Corrected. And in fact, throughout the text, we chose to use the abbreviation LGR-CW-QCL as in Rannik et al. (2015).

P. 6, L. 1: Do you intentionally differentiate between "co-variances" (here and L. 5) and "covariance"?
We did not intend to use "co-variances" but rather "covariance". This is now corrected.

P. 6, L. 10: Write "daytime" instead of "day-time" as in the rest of the manuscript. Correct also on P. 7, L. 15 and on P. 9, L. 24.
Corrected.

P. 6, L. 27: I suggest using "over the 9-month measurement period" instead of "in the end of the 9-month measurement period" as the used expression could be misleading otherwise.
Corrected.

P. 7, L. 17: Use superscript in units.
Corrected.

Figures 2-5: Instead of using the day of year numbers, I suggest to use the introduced classification of S, ES, MS . . . in the subplot titles (or use both, DOY + the classification). This makes it easier to compare with Figure 1 and descriptions in the text.
We modified the figures 2-5 to include the classification of S, ES, MS… + DOY (e.g. S, 110-145), similar to that presented in Tables 1 and 2.

**4 References**

[revised manuscript text omitted]

---

## Editor Decision (ED1)

**Second review by referee #1**

**General comments**

Literature review should go partly from Discussion to Introduction.
The N2O flux data is still a little bit misplaced in the manuscript. In the results some N2O flux patterns are described, but only after reading the discussion it becomes clear to the reader WHY the authors want to study N2O fluxes in combination with CO fluxes, and suddenly in the Conclusions, it becomes one of the main findings. I agree with the authors that some process understanding and previous studies should be known by the readers, but since the link between CO and N2O is not so well known, I would add a few lines in the Introduction as well. So, part of what is written in Discussion should also/only appear in the Introduction.

The same remark counts for the NEE/CO2 literature review. In the last sentence of the Introduction, the authors state that they will focus on CO2, and don't mention NEE. It should be clear to the reader earlier why this is of interest. So, part of the NEE-CO relationship literature review should appear earlier in the manuscript.

Concerning the NEE-CO discussion, it would be good if the authors are a little more careful with their conclusions. They state that a negative correlation between FCO and NEE indicates biological CO formation (page 14, line 21-22). Here no reference is given, is this a conclusion of the authors themselves? While I agree that the negative correlation CAN be an indicator for the biological CO formation, it can also be caused by other indirect effects (some environmental factors are closely related to NEE, and could also be a driving factor for FCO). It would be good if the authors give a reference here, or shortly elaborate why they are sure about this statement.

Footprint analyses
Footprint analyses are now well explained, just a minor comment. In the introduction it is made clear that CO fluxes are pretty dependent on environmental factors such as ground water. Considering this, are there any elevation changes in your footprint? Wet or dry spots? Or does this not play any role? A sentence can be added to clarify this for the reader.

Figure 2, 3 and 4.
The figure which was added to the Authors response (so not in manuscript) is very useful. Is it possible to plot the soil temperature inside the manuscript as well? I see that the authors decided to write 'not shown' but I think there is no need for a new figure, it could be easily added to Figure 3. This would help for the statement that thermal degradation probably doesnt place a role in the early morning hours (page 14, line 10-15). A small remark on this point. It seems unlikely that thermal degradation is ever fully absent. However, considering the previous studies who found exponential curves with temperature, and that the fieldsite is located in a cold climate, the fluxes are probably very small. Therefore, I would rephrase your sentence like 'we expect it to be negligible'.

Figure 2: It would be good to draw a black line on the 0-line, to clearly divide uptake and emission periods. Also, in this figure, could you indicate the moment of sunset and sunrise?

Figure 3: As said, the addition of soil temperature here would be very nice and insightful.
Figure 4: Maybe also here indicate the moment of sunrise and sunset.

Tables 1 and 2 (and explanation in text)
Table 1 and 2, and the explanation in the text, need quite some improvement. In the current form, it takes a lot of effort from the reader to interpret each column, and especially the values in Table 2 are quite a puzzle without a good explanation. This could be easily improved by adding a few lines either in the text, or besides the column.

Table 1
Table 1, please check the description and maybe clarify methods. As I understand, first 'mean' column takes average of FCO when hsun>0, second 'mean 'column takes average of FCO when h<0. But the last 'mean' column is not explained. The reader probably assumes it is the mean over all FCO data, but it is better to clarify this.

Table 2
Table 2: I think the use of FCO is confusing here. In Table 1, FCO is meant for the actual measured flux (right?). In Table 2, you state 'gross FCO', but I think you actually mean 'estimated production during daytime', right? In the text, it is more clear because you define it as emissions (page 9, line 24). However, using 'gross FCO' is confusing, since F stands for flux, and flux is usually the net result of uptake and emission. So, if the authors indeed mean production, please rename this term and call it 'gross CO production during the day' or something similar. Gross FCO will confuse the readers.

The first columns of Table 2 'Gross FCO' are explained in the Table-text in the last sentence (Gross CO fluxes refer to the difference between.... presented in Table 1). Please add such a sentence for all 3 'mean' values (for 'gross FCO-day', for 'uptake FCOday(Q10,1.8)', for 'gross FCO_day(Q10, 1.8)'), and elaborate. For example:

Gross CO fluxes (gross FCO_day) refer to the difference between daytime fluxes (FCO_day) and nighttime fluxes (FCO_night) presented in Table 1. With other words, this is the estimated net production of CO with an assumed constant CO uptake, based on measured uptake rates at night.

Uptake CO fluxes (uptake FCO_day(Q10, 1.8)) refers to the estimated CO uptake taking place during the day, based on measured CO uptake values at night. The value is extrapolated from averaged measured night time CO uptake (Table 1), and extrapolated with a Q10 of 1.8 to day time temperatures (Whalen and Rheeburg).

Estimated CO production/emission fluxes (gross F_CO_day) values are based on column 1 from Table 1, minus column 6 from Table 2. Etc.

You could also refer to page 9, line 24 here, where you describe the 2 'ways' of estimation. You could cleary state you refer to the first 'way' here. So link the text (at page 9, line 24) better with your values in Table 2.

Again, even if the information is probably findable in the text, it should be more clear since in this

form, it takes too much effort of the reader to interpret this table.

**Specific comments**

P1, line 15: 'However'.. I have the feeling this sentence does not contradict the previous one, so better not use 'however'. Maybe choose another word. 'In general, soils are considered as.....'

P1, line 16, micrometeorological eddy--> the micrometeorological eddy

P1, line 18: as well as relevant--> as well as to relevant

P1, line 20-21, you mention that mid-April to mid June the field is a net source, the rest of the measurement period (July-Nov) was a net sink, but you exclude the end of June in this sentence. This is not the maintenance period, right? I would rephrase.

P1, line 22: and an emission--> and a net CO emissions

P2, Line 17: reference to Funk 1994 is not in bibliography. Please check all your references once more

P2, Line 17: Emissions of CO from water logged soils have often been attributed to anaerobic production of CH4.

The paper of Funk only says that the occurrence of CO fluxes correspond with the occurrence of CH4 fluxes. This paper mostly underlines the UTILIZATION of CO for producing CH4. Furthermore, the paper of Varella doesnt measure or mention any CH4. Please remove or correct this statement, and refer to the correct papers.

P2, Line 18: 'such as thermal or UV- or visible light'
change to
'such as thermal, UV- or visible light'

P2-3, Line 26- Line 1:In the current form, the sentence is incorrect gramatically. Either divide into two sentences (split before 'while' and check commas), or rephrase.

P3, line 1-3: What is described here is also sometimes refered to as indirect photodegradation. Can you merge this with page 2, line 23?

P3, line 9: 'are needed for CO is formation'--> remove 'is'

P3, line 15: remove extra bracket

P 3, line 17, add white space before 'with a tendency'

P3, line 16-20: Here the statement is made that higher CO uptake is reported from natural and dry soils, followed by many references. Do these references all support this statement, or you state this

fact yourself after reading these articles? Or do these articles only support the first part of the sentence (the reported -2 to 2 nmol m2 s)? Please clarify

P3, line 19-22. Same statement here. Now it seems that all these papers support this statement. I assume that you observed this gradient yourself after reading these papers? Maybe clarify this.

P3, line 25: 'and in North'--> change to 'and in the North'

P3, line 28: 'using micrometeorological'--> change to 'using the micrometeorological

P4, line 1, 'as well as relevant' change to 'as well as with relevant'

P4, line 6, sentence has unlogical order. Change to something like:
The measurements were conducted on a mineral agrictultural field located in Eastern Finland (63..., 27...), cultivated with a perennial reed canary grass (RCG, Ph.....)

P4, line 10, sentence has unlogical order. Change to something like:
In 2011 in the beginning of the growing season (23 May), the crop was fertilized with an N-P-K-S fertilizer.....

P4, line 11: Be consistent with dates. Sometimes you write 23 May, other places 28 OF april. Furthermore, in line 12, you add the day number (day 118). That is quite useful, since you continue using that the rest of the manuscript. Maybe also do that for page 4, line 11.

P4, line 20-22, 'within the ploughing layer from the surface to about 30 cm'--> does this count for as well the soil pH as the soil organic matter? Unclear from this sentence. Also the last part of the sentence seems to lack a verb. Please check.

P5, line 7: reduces footprint extent--> reduces the footprint extent

P5, line 9. Why is refered to figure 1 c. Do you mean 1d?

P5, line 17, please add day numbers after April to November 2011 (Like you did in line 18)

P 5, lines 20-21. This sentence seems to assume that the reader knows about the Rannik paper. Please rephrase, something like:
The AR-CW-QCL and LGR-CQ-QCL were the same as used in the study of Rannik (2015) wherein four laser based fast response gas analyzers to measure N2O were compared (or something similar).

P 5, line 22, add day number

P6, line 21. 'Sa' is not defined in text.

P6, line 23-26. The despiking process is well described. However, which percentage of your data was replaced? Can you state this in the text?

P7, line 27, add coma after 'the fluxes', makes reading more smooth

P7, line 28. Groups of days well described. Just a suggestion, is it possible to add real dates between brackets? Easer for reader to interpret the groups.

P8, line 22: the term 'cumulative CO flux' is introduced as cum FCO. The text says it shows that the site is a net sink of CO. Where is that shown? I assume that cumulative stands for the total measurement period of 7 months? Is this the same term as 'net FCO' for days 110-325 in table 1? If so, you can refer to Table 1, and clarify that cumulative is the same as net FCO for the period 'all' in Table 1.

P9, line 9, The autumn was characterized by decreasing FCO..... Statement too vague. By 'the autumn' do you mean A+LA (so days 241 to 325)? And, which FCO is meant here? Net FCO during the day or night or net? Or estimated production in Table 2? Please clarify

Page 10, line 2, add white space

Page 11, lines 19: over the whole measurement period → over the whole 7 month measurement period.

Page 12, line 5-7. sentence unclear. Maybe: were rather low, crop was not yet → were rather low and the crops were not yet....

Page 12, line 10. Decreasing amount of--> decreasing amounts of

page 12, line 25: to calculate and annual--> to calculate an annual

Page 12, line 25: when stating the number -0.25, please refer to Table 1, so reader knows where the number comes from

Page 13, line 7-11. I would refer here to the same papers as in Table 5, to be consistent to the reader

Page 13, line 14. You introduce Mco here, and introduce the abbreviation. However, if you dont use this term anymore afterwards, I think there is no need to introduce an abbreviation.

Page 13, line 15: In line 12 you state that you expect CO emission also exists during the day. In line 15, you state 'if existing'. I would phrase your doubt less strong, more like

In our site the estimated/assumed daytime CO consumption is overruled by.....

Page 13, line 26: drives → drive

Page 14, line 29. You state that a supporting factor includes the high C to N ratio. However, since it is an important point, I would add the accompaying reference right after this point. Now it is at the

end of the sentence and unclear for the reader which reference belongs to which supporting factor. Or you could take this point out of this sentence and merge it with the next sentence, since you elaborate there anyway.

Page 14, line 15-16: Is thermal degradation not by definition temperature dependent? No need for reference here.

Page 14, line 21: Based on understanding of biological CO formation, a negative correlation between FCO and NEE....

This is nice information, but it would be good if the reader is aware of this assumed relationship before. Could this expected relationship be stated and explained in Introduction? This might help the reader understand the flow and content of the paper better. Also, as mentioned in general comments, please elaborate on this negative correlation, can this only mean biological formation, or can this correlation also be caused by something else.

Page 14, line 25-26: at the RCG crop--> at the RCG field site/arable land/....

Page 15, line 6-7: verb missing. Maybe: net CO emission also--> net CO emissions occuring also

Page 15,line 14: verb too much, remove 'remain'

Page 15, line 14-16: incorrect/unclear sentence. Rephrase to something like:
A study by K&C (2002) demonstrated the lack of understanding in sink-source dynamics of CO, and showed that plant roots are capable of producing CO, which rate/source can be as high as the current.....

Page 15, line 17. Also stated in general comments. This expected strong relationship should already be clear in Introduction.

Page 16, line 6: the smaller emissions of CO...... Do you refer to literature here or to your own data? Rephrase/clarify

**Figures**

Figure 1. If the manuscript is printed in black/white, the lines are hard to differentiate. Could the lines have different patterns?

Very minor comments, but why are there different types of blue used in figure 5, in comparison to previous figures?

**Tables**

Table 2, please add white space before (Q10, 1.8) (two times)

Table 5, The authors have explained why they keep the table in this form, and have elaborated in the text about which study measured at daytime, and which diurnal. This is fine, but it would help if the header and table would be more self-explanatory. Elaborate column 4 by for example: 'Data Period, measurement frequency, and moment of measurement'. In the current form the header 'diurnal cycle' doesn't really cover the content of the column

---

## Author Response (AR2)

Author response to the referee #1 comments

We want to sincerely thank the reviewer for detailed and constructive comments and suggestions that certainly have improved the manuscript. We acknowledge the expert input and deep understanding of the topic, and we feel privileged to have had him/her as a reviewer. We have now carefully addressed all the comments.

You can find our response below (in blue) to each of the reviewer comments. In the corrected manuscript version attached below, we have colored the changes in red. We have not marked each small change (e.g. missing commas, changed words etc), but mostly highlighted the significant changes.

on behalf of all co-authors,
Mari Pihlatie

--
Second review by referee #1

General comments

Literature review should go partly from Discussion to Introduction.
The N2O flux data is still a little bit misplaced in the manuscript. In the results some N2O flux patterns are described, but only after reading the discussion it becomes clear to the reader WHY the authors want to study N2O fluxes in combination with CO fluxes, and suddenly in the Conclusions, it becomes one of the main findings. I agree with the authors that some process understanding andprevious studies should be known by the readers, but since the link between CO and N2O is not so well known, I would add a few lines in the
introduction as well. So, part of what is written in Discussion should also/only appear in the Introduction.

We have added short description of N2O process understanding and potential links between CO and N2O fluxes to the introduction (P 2, lines 14-16). In the end of the introduction (P 4, lines 7-13), we also added a sentence to describe what we expect based on the current understanding of the links between N2O and CO fluxes. We have also modified the discussion accordingly, not to repeat what was written in the introduction.

The same remark counts for the NEE/CO2 literature review. In the last sentence of the Introduction, the authors state that they will focus on CO2, and don't mention NEE. It should be clear to the reader earlier why this is of interest. So, part of the NEE-CO relationship literature
review should appear earlier in the manuscript.

In the introduction (P 2, lines 10-17) we write that the "Understanding of the biological processes leading to CO release and the importance of these sources in terrestrial ecosystems are poorly understood (Moxley and Smith, 1998; King and Crosby, 2002; Vreman et al., 2011; He and He, 2014)." We list biological processes, which have been found to release CO, and give references therein, but we also state that the importance of biological CO forming processes in the net CO exchange and, in general, to the global CO budget still remain largely unknown.

In the end of the introduction (P 4, lines 7-13), we now write: "Based on previous studies, we expect that the diurnal and seasonal variations in $F_{CO}$ are strongly dependent on radiation and temperature. On the other hand, we do not expect strong relationships between $F_{CO}$ and NEE, or $F_{CO}$ and $N_2O$ fluxes due to the limited information available on the involvement of biological processes in $F_{CO}$, and challenges in separating between parallel abiotic and biotic drivers of $F_{CO}$. We hypothesize that a negative correlation between $F_{CO}$

and NEE can indicate an involvement of a biological component in CO production, and that a positive correlation between night-time $F_{CO}$ and $N_2O$ flux may indicate an involvement of nitrifiers in CO consumption."

Concerning the NEE-CO discussion, it would be good if the authors are a little more careful with their conclusions. They state that a negative correlation between FCO and NEE indicates biological CO formation (page 14, line 21-22). Here no reference is given, is this a conclusion of the authors themselves? While I agree that the negative correlation CAN be an indicator for the biological CO formation, it can also be caused by other indirect effects (some environmental factors are closely related to NEE, and could also be a driving factor for FCO). It would be good if the authors give a reference here, or shortly elaborate why they are sure about this statement.

We have now rewritten this part of the text in the Discussion (P 15, lines 16-28) to avoid drawing too strong conclusions on the issue. Now it states: "Although we cannot separate between biotic and abiotic CO formation at the RCG field site, our findings of the negative correlation between daytime $F_{CO}$ and NEE (r=-0.469) during early summer (days 146-160), the period of maximum NEE, indicate that some CO may also be formed via plant physiological processes. This early summer CO emission period (days 146-160) coincides with the steepest slope in $CO_2$ uptake (more negative NEE), supporting the findings of Wilks (1959), Bruhn et al. (2013) and Fraser et al. (2015) that CO can be emitted not only from dead plant matter but also from living green leaves. The observed daytime CO emissions during early summer can have also been formed through abiotic processes, which also occur in living plants (Tarr et al., 1995; Erickson et al., 2015). King et al. (2012) suggested that the CO emissions from photodegradation generally decrease with increasing leaf area index, and Tarr et al. (1995) and Erickson et al. (2015) found that the CO photoproduction efficiency is lower for living plants compared to senescent or dead vegetation. These studies support our findings of lower daytime CO emissions from fully developed crop during the summer (days 205-240) compared to CO emissions during the spring (days 110-145), when the ground was covered by the dead plant litter. Still the role of biological CO formation in living green plants and the forming processes remain unresolved and call for further process-studies."

Footprint analyses
Footprint analyses are now well explained, just a minor comment. In the introduction it is made clear that CO fluxes are pretty dependent on environmental factors such as ground water. Considering this, are there any elevation changes in your footprint? Wet or dry spots? Or does this not play any role? A sentence can be added to clarify this for the reader.

P 4, lines 26-27: We added a sentence specifying that there is a slight slope in the footprint and the wettest area lies in the northern corner of the footprint. During the snow melt there is always standing water in the northern corner of the field, however, there is no standing water during the growing season. We did not perform any measurements on the moisture gradient at the field.

Figure 2, 3 and 4.
The figure which was added to the Authors response (so not in manuscript) is very useful. Is it possible to plot the soil temperature inside the manuscript as well? I see that the authors decided to write 'not shown' but I think there is no need for a new figure, it could be easily added to Figure 3. This would help for the statement that thermal degradation probably doesnt place a role in the early morning hours (page 14, line 10-15). A small remark on this point. It seems unlikely that thermal degradation is ever fully absent. However, considering the previous studies who found exponential curves with temperature, and that the fieldsite is located in a cold climate, the fluxes are probably very small. Therefore, I would rephrase your sentence like 'we expect it to be negligible'.

We made all the suggested changes to the Figures and to the text concerning the role of thermal degradation. We also changed all the figures to black and white improve readability when printed black and white.

Figure 2: It would be good to draw a black line on the 0-line, to clearly divide uptake and emission periods. Also, in this figure, could you indicate the moment of sunset and sunrise?

Figure 2. We added 0-line, and the moment of sunrise and sunset.

Figure 3: As said, the addition of soil temperature here would be very nice and insightful.
Figure 4: Maybe also here indicate the moment of sunrise and sunset.

Figure 3. We added soil temperature
Figure 4. We added the moment of sunrise and sunset

Tables 1 and 2 (and explanation in text)
Table 1 and 2, and the explanation in the text, need quite some improvement. In the current form, it takes a lot of effort from the reader to interpret each column, and especially the values in Table 2 are quite a puzzle without a good explanation. This could be easily improved by adding a few lines either in the text, or besides the column.

We clarified the Tables 1 and 2, and explanation in the text. We followed the suggestions below to make the reading easier. See in detail below.

Table 1
Table 1, please check the description and maybe clarify methods. As I understand, first 'mean' column takes average of FCO when hsun>0, second 'mean 'column takes average of FCO when h<0. But the last 'mean' column is not explained. The reader probably assumes it is the mean over all FCO data, but it is better to clarify this.

In Table 1, the last column (net FCO) is now explained as "a net flux over all $F_{CO}$ data (net $F_{CO}$) for the six measurement periods".

Table 2
Table 2: I think the use of FCO is confusing here. In Table 1, FCO is meant for the actual measured flux (right?). In Table 2, you state 'gross FCO', but I think you actually mean 'estimated production during daytime', right? In the text, it is more clear because you define it as emissions (page 9, line 24). However, using 'gross FCO' is confusing, since F stands for flux, and flux is usually the net result of uptake and emission. So, if the authors indeed mean production, please rename this term and call it 'gross CO production during the day' or something similar. Gross FCO will confuse the readers.

We agree that the use of FCO was misleading here. We renamed the 'gross FCO' to 'gross daytime CO emission'. We wanted to use the word 'emission' instead of production as the word 'emission' better explains that we measure net emission, whereas 'production' refers to the production process, while there can be simultaneous consumption of CO.

The first columns of Table 2 'Gross FCO' are explained in the Table-text in the last sentence (Gross CO fluxes refer to the difference between.... presented in Table 1). Please add such a sentence for all 3 'mean' values (for 'gross FCO-day', for 'uptake FCOday(Q10,1.8)', for 'gross FCO_day(Q10, 1.8)'), and elaborate. For example:

Gross CO fluxes (gross FCO_day) refer to the difference between daytime fluxes (FCO_day) and

nighttime fluxes (FCO_night) presented in Table 1. With other words, this is the estimated net production of CO with an assumed constant CO uptake, based on measured uptake rates at night.

Uptake CO fluxes (uptake FCO_day(Q10, 1.8)) refers to the estimated CO uptake taking place during the day, based on measured CO uptake values at night. The value is extrapolated from averaged measured night time CO uptake (Table 1), and extrapolated with a Q10 of 1.8 to day time temperatures (Whalen and Rheeburg).

Estimated CO production/emission fluxes (gross F_CO_day) values are based on column 1 from Table 1, minus column 6 from Table 2. Etc.

In the Table-text of Table 2, we write now: The estimated gross daytime CO emission is calculated in two ways: 1) assuming a constant CO uptake, and 2) assuming temperature dependent CO uptake. Gross daytime CO emission based on a constant CO uptake (way 1, Chapter 2.4) refers to the difference between daytime fluxes (FCO_day) and night-time fluxes (FCO_night) presented in Table 1. The temperature corrected gross daytime CO emission (Gross daytime CO emission (Q10, 1.8)) refers to the difference between daytime fluxes (FCO_day) (Table 1.) and daytime CO uptake (Q10, 1.8). The daytime CO uptake (Daytime CO uptake (Q10, 1.8)) is calculated by extrapolating the night-time CO fluxes (FCO_night) to daytime using the difference between day and night soil temperatures (2.5 cm depth) ($\Delta$tsoil) and the Q10-value of 1.8 (Whalen and Reeburgh, 2001), as described in Chapter 2.4.

You could also refer to page 9, line 24 here, where you describe the 2 'ways' of estimation. You could cleary state you refer to the first 'way' here. So link the text (at page 9, line 24) better with your values in Table 2.

In the Table-text, we added a reference to the Chapter 2.4 (instead of giving page and line numbers), as it is difficult to give specific page and line numbers, when (and if) this manuscript is accepted as a publication.

Again, even if the information is probably findable in the text, it should be more clear since in this form, it takes too much effort of the reader to interpret this table.

We agree and we hope the description is now clarified. We have also modified the description of the calculations in Chapter 2.4 (Pages 8-9, lines 24-), and therein we refer to the Table 2.

Specific comments
P1, line 15: 'However'.. I have the feeling this sentence does not contradict the previous one, so better not use 'however'. Maybe choose another word. 'In general, soils are considered as.....'

P1, lines 14-15: We changed the order of two sentences, stating now that "Soils are generally considered as a sink of CO due to microbial oxidation processes, while emissions of CO have been reported from a wide range of soil-plant systems."

P1, line 16, micrometeorological eddy--> the micrometeorological eddy

Corrected.

P1, line 18: as well as relevant--> as well as to relevant

Corrected.

P1, line 20-21, you mention that mid-April to mid June the field is a net source, the rest of the measurement period (July-Nov) was a net sink, but you exclude the end of June in this sentence. This

is not the maintenance period, right? I would rephrase.

We rewrote this sentence as follows: "The reed canary grass crop was a net source of CO from mid-April to mid-June, and a net sink throughout the rest of the measurement period from mid-June to November 2011, excluding a measurement break in July."

P1, line 22: and an emission--> and a net CO emissions

Corrected.

P2, Line 17: reference to Funk 1994 is not in bibliography. Please check all your references once more

All the references were checked, and Funk 1994 was added to the reference list.

P2, Line 17: Emissions of CO from water logged soils have often been attributed to anaerobic production of CH4.

The paper of Funk only says that the occurrence of CO fluxes correspond with the occurrence of CH4 fluxes. This paper mostly underlines the UTILIZATION of CO for producing CH4. Furthermore, the paper of Varella doesnt measure or mention any CH4. Please remove or correct this statement, and refer to the correct papers.

P2, line 19-23: We modified the sentence accordingly: "Although microbial CO formation may occur in anaerobic conditions (Funk et al., 1994; Rich and King, 1999), most often the CO production has been related to abiotic processes such as thermal, UV- or visible light-induced degradation of organic matter or plant material…"

P2, Line 18: 'such as thermal or UV- or visible light' change to 'such as thermal, UV- or visible light'

Corrected.

P2-3, Line 26- Line 1:In the current form, the sentence is incorrect gramatically. Either divide into two sentences (split before 'while' and check commas), or rephrase.

P3, lines 5-9: We split the sentence, and it is now written: "Thermal degradation is identified as the temperature-dependent degradation of carbon in the absence of radiation and possibly oxygen (Derendorp et al., 2011; Lee et al., 2012; van Asperen et al., 2015).  The separation between CO formation through thermal degradation and photodegradation is very challenging because they both can take place simultaneously and the indirect photodegradation may occur even in the absence of solar radiation if adequate thermal energy is present (Lee et al., 2012)."

P3, line 1-3: What is described here is also sometimes refered to as indirect photodegradation. Can you merge this with page 2, line 23?

The description of indirect photodegradation was now merged in one place, and can be found in P2-3, line 25 onwards.

P3, line 9: 'are needed for CO is formation'--> remove 'is'

P3, lines 11-17: this chapter dealing with biological CO formation was rewritten.

P3, line 15: remove extra bracket

Corrected.

P 3, line 17, add white space before 'with a tendency'

Corrected.

P3, line 16-20: Here the statement is made that higher CO uptake is reported from natural and dry soils, followed by many references. Do these references all support this statement, or you state this fact yourself after reading these articles? Or do these articles only support the first part of the sentence (the reported -2 to 2 nmol m2 s)? Please clarify

P3, lines 23-26: we modified this part so that it more clearly states what is supported by the literature. We also removed part of the text to focus on the relevant information related to our study, and to give a general view of the reported CO flux rates. More details of the differences between ecosystem types is discusses in the Discussion (P13, lines 7-12).

P3, line 19-22. Same statement here. Now it seems that all these papers support this statement. I assume that you observed this gradient yourself after reading these papers? Maybe clarify this.

P3, lines 24-26: now we write that: "Based on the available literature, there is a tendency of south to north gradient with higher CO emissions from tropical and Mediterranean environments compared to boreal and temperate ecosystems."

P3, line 25: 'and in North'--> change to 'and in the North'

Corrected.

P3, line 28: 'using micrometeorological'--> change to 'using the micrometeorological

Corrected.

P4, line 1, 'as well as relevant' change to 'as well as with relevant'

Corrected.

P4, line 6, sentence has unlogical order. Change to something like:
The measurements were conducted on a mineral agrictultural field located in Eastern Finland (63...,
27...), cultivated with a perennial reed canary grass (RCG, Ph.....)

P4, lines 16-17: Corrected as follows: "The measurements were conducted on a mineral agricultural field located in Eastern Finland (63°9'48.69'' N, 27°14'3.29'' E), cultivated with a perennial reed canary grass (RCG, *Phalaris arundinaceae*, L. cv. Palaton)."

P4, line 10, sentence has unlogical order. Change to something like:
In 2011 in the beginning of the growing season (23 May), the crop was fertilized with an N-P-K-S
fertilizer.....

P4, line 20: Corrected as follows: "In 2011 in the beginning of the growing season (23 May, day 143), …"

P4, line 11: Be consistent with dates. Sometimes you write 23 May, other places 28 OF april.

Furthermore, in line 12, you add the day number (day 118). That is quite useful, since you continue using that the rest of the manuscript. Maybe also do that for page 4, line 11.

We modified the date formats to be consistent, and added the day number when it was suitable.

P4, line 20-22, 'within the ploughing layer from the surface to about 30 cm'--> does this count for as well the soil pH as the soil organic matter? Unclear from this sentence. Also the last part of the sentence seems to lack a verb. Please check.

P 5, lines 3-4: We modified this text, and now it reads: "Within the ploughing layer from the surface to about 30 cm, soil pH varies from 5.4 to 6.1, and soil organic matter content varied between 3 and 11%, respectively."

P5, line 7: reduces footprint extent--> reduces the footprint extent

Corrected.

P5, line 9. Why is refered to figure 1 c. Do you mean 1d?

Corrected.

P5, line 17, please add day numbers after April to November 2011 (Like you did in line 18)

Corrected.

P 5, lines 20-21. This sentence seems to assume that the reader knows about the Rannik paper. Please rephrase, something like:
The AR-CW-QCL and LGR-CQ-QCL were the same as used in the study of Rannik (2015) wherein four laser based fast response gas analyzers to measure N2O were compared (or something similar).

P 5, lines 2-4: the sentence was corrected as suggested.

P 5, line 22, add day number

Corrected.

P6, line 21. 'Sa' is not defined in text.

P 7, line 3: now Sa is defined.

P6, line 23-26. The despiking process is well described. However, which percentage of your data was replaced? Can you state this in the text?

We consider that this information is not relevant for the reader. The despiking is performed to the high frequency rawdata (10 Hz timeseries). The number of spikes for each half-hour is saved in the output, and if there are more than 300 sec of spikes in one half-hour, that flux value is marked as a missing value.

P7, line 27, add coma after 'the fluxes', makes reading more smooth

P7, line 10: A comma was added.

P7, line 28. Groups of days well described. Just a suggestion, is it possible to add real dates between

brackets? Easer for reader to interpret the groups.

P7, lines 11-13: real dates were added.

P8, line 22: the term 'cumulative CO flux' is introduced as cum FCO. The text says it shows that the site is a net sink of CO. Where is that shown? I assume that cumulative stands for the total measurement period of 7 months? Is this the same term as 'net FCO' for days 110-325 in table 1? If so, you can refer to Table 1, and clarify that cumulative is the same as net FCO for the period 'all' in Table 1.

P9, lines 12-14: We modified the text to better explain how the cumulative flux was obtained, as follows "Cumulative CO flux (cum FCO) curves, calculated by cumulating the half-hourly fluxes, show that the site was a net sink of CO over the 7-month measurement period (Fig. 1f)." Hence, the cumulative flux is not literally the same as net FCO for days 110-325, which is a mean of all the half-hourly CO fluxes over the period of days 110-325.

P9, line 9, The autumn was characterized by decreasing FCO..... Statement too vague. By 'the autumn' do you mean A+LA (so days 241 to 325)? And, which FCO is meant here? Net FCO during the day or night or net? Or estimated production in Table 2? Please clarify

P9, lines 24-27: This sentence was clarified and states now: "The autumn (A, LA) was characterized by decreasing daytime $F_{CO}$ ($F_{CO\_day}$) and slowly dropping air and soil temperatures, decreasing radiation intensity, and decreasing photosynthetic activity of the crop (less negative NEE) (Fig. 1)."

Page 10, line 2, add white space

Added.

Page 11, lines 19: over the whole measurement period → over the whole 7 month measurement period.

Corrected.

Page 12, line 5-7. sentence unclear. Maybe: were rather low, crop was not yet → were rather low and the crops were not yet....

P 12, line 29: Corrected as suggested.

Page 12, line 10. Decreasing amount of--> decreasing amounts of

Corrected as suggested.

page 12, line 25: to calculate and annual--> to calculate an annual

Corrected as suggested.

Page 12, line 25: when stating the number -0.25, please refer to Table 1, so reader knows where the number comes from

 P 13, line 20: this reference to Table 1 was added.

Page 13, line 7-11. I would refer here to the same papers as in Table 5, to be consistent to the reader

P 14, lines 2-5: Many of the referred papers here are process studies, and as this chapter / sentence refers to the processes, and to observations of the processes, we preferred to keep these references. In Table 5, most of the studies are field studies reporting net CO fluxes, and many of these papers do not focus on processes.

Page 13, line 14. You introduce Mco here, and introduce the abbreviation. However, if you dont use this term anymore afterwards, I think there is no need to introduce an abbreviation.

P 14, lines 7-8: We agree with this comment, however, as MCO is presented now in the correlation tables 3 and 4, we left also the abbreviation in the text, as it is shown in the tables. Now the text reads: "We did not find correlation between daytime or night-time CO concentration (MCO) and FCO (Tables 3 and 4),…"

Page 13, line 15: In line 12 you state that you expect CO emission also exists during the day. In line 15, you state 'if existing'. I would phrase your doubt less strong, more like
In our site the estimated/assumed daytime CO consumption is overruled by.....

P 14, line 9-10: Corrected as suggested. "In our site the estimated daytime CO consumption is overruled by a simultaneous strong CO production…"

Page 13, line 26: drives → drive

Corrected.

Page 13, line 29. You state that a supporting factor includes the high C to N ratio. However, since it is an important point, I would add the accompaying reference right after this point. Now it is at the end of the sentence and unclear for the reader which reference belongs to which supporting factor. Or you could take this point out of this sentence and merge it with the next sentence, since you elaborate there anyway.

P 14, lines 24-29: We merged the text with the text where we elaborated with factors supporting CO formation via abiotic degradation processes. Now the text reads: "Factors supporting the CO production through abiotic photodegradation and thermal degradation processes include high C to N ratio of the plant material (King et al., 2012), presence of oxygen (Tarr et al., 1995; Lee et al., 2012), greater solar radiation exposure (no shading) (King et al., 2012), and litter area to mass ratio (King et al., 2012; Lee et al., 2012). As the dead plant material in our measurement site has a high C to N ratio (mean ±stdev: 66±6.3), and as this dry plant material was well exposed to radiation in the spring, we expect that the conditions were suitable for CO formation through abiotic degradation processes."

Page 14, line 15-16: Is thermal degradation not by definition temperature dependent? No need for reference here.

References removed.

Page 14, line 21: Based on understanding of biological CO formation, a negative correlation between FCO and NEE....

This is nice information, but it would be good if the reader is aware of this assumed relationship before. Could this expected relationship be stated and explained in Introduction? This might help the reader understand the flow and content of the paper better. Also, as mentioned in general comments, please elaborate on this negative correlation, can this only mean biological formation, or

can this correlation also be caused by something else.

We have addressed this topic shortly in the Introduction (P 3, lines 10-17) and in the Discussion (P 15, lines 16-28). We agree that it is good to write open our assumptions and expectations as early as possible in the manuscript. As there is very little information available on the connections between FCO and NEE, and in general, on the biological CO formation processes, we have now minimized speculations based on our data. We do acknowledge that some CO may be formed in plant physiological processes, however, we also state that our data does not allow drawing conclusions on the involvement of biological (or abiotic) processes.

Page 14, line 25-26: at the RCG crop--> at the RCG field site/arable land/....

This sentence was deleted as the chapter was reorganized and compressed.

Page 15, line 6-7: verb missing. Maybe: net CO emission also--> net CO emissions occuring also

This sentence was deleted as the chapter was reorganized and compressed.

Page 15, line 14: verb too much, remove 'remain'

Corrected.

Page 15, line 14-16: incorrect/unclear sentence. Rephrase to something like:
A study by K&C (2002) demonstrated the lack of understanding in sink-source dynamics of CO, and showed that plant roots are capable of producing CO, which rate/source can be as high as the current.....

This sentence was deleted due to the efforts in minimizing speculations of biological CO formation at our site.

Page 15, line 17. Also stated in general comments. This expected strong relationship should already be clear in Introduction.

We elaborate the relationship between night-time FCO and N2O fluxes in the Discussion at Page 16, lines 3-13. We also added a sentence in the beginning of the Introduction (P 2, lines 14-16) stating that "A diverse group of soil microbes are capable of oxidizing CO. They include carboxydotrophs, methanotrophs, and nitrifiers (Ferenci et al., 1975; Jones and Morita, 1983; Bender and Conrad, 1994; King and Weber, 2007), hence potentially linking CO fluxes to the exchange of $CH_4$ and $N_2O$." Additionally, in the end of the Introduction (P4, lines 7-13) we state that "Based on previous studies, we expect that the diurnal and seasonal variations in $F_{CO}$ are strongly dependent on radiation and temperature. On the other hand, we do not expect strong relationships between $F_{CO}$ and NEE, or $F_{CO}$ and $N_2O$ fluxes due to the limited information available on the involvement of biological processes in $F_{CO}$, and challenges in separating between parallel abiotic and biotic drivers of $F_{CO}$. We hypothesize that a negative correlation between $F_{CO}$ and NEE can indicate an involvement of a biological component in CO production, and that a positive correlation between night-time $F_{CO}$ and $N_2O$ flux may indicate an involvement of nitrifiers in CO consumption."

Page 16, line 6: the smaller emissions of CO...... Do you refer to literature here or to your own data? Rephrase/clarify

This sentence was deleted to make the Discussion more concise and avoid overlapping. The role of biotic vs. abiotic processes in CO formation are discussed now on Page 15, lines 16-28.

Figures

Figure 1. If the manuscript is printed in black/white, the lines are hard to differentiate. Could the lines have different patterns?

We changed all the figures to black and white to avoid problems in differentiating the lines.

Very minor comments, but why are there different types of blue used in figure 5, in comparison to previous figures?

Now and the figures are presented in black and white.

Tables

Table 2, please add white space before (Q10, 1.8) (two times)

White space added.

Table 5, The authors have explained why they keep the table in this form, and have elaborated in the text about which study measured at daytime, and which diurnal. This is fine, but it would help if the header and table would be more self-explanatory. Elaborate column 4 by for example: 'Data Period, measurement frequency, and moment of measurement'. In the current form the header 'diurnal cycle' doesn't really cover the content of the column

Table 5. We changed the column 4 to 'Data Period, measurement frequency, and moment of measurement' as suggested.

[revised manuscript text omitted]